# Distributionally Robust Federated Averaging

**Yuyang Deng**     **Mohammad Mahdi Kamani**     **Mehrdad Mahdavi**
The Pennsylvania State University
`{yzd82,mqk5591,mzm616}@psu.edu`

## Abstract

In this paper, we study communication efficient distributed algorithms for distributionally robust federated learning via periodic averaging with adaptive sampling. In contrast to standard empirical risk minimization, due to the minimax structure of the underlying optimization problem, a key difficulty arises from the fact that the global parameter that controls the mixture of local losses can only be updated infrequently on the global stage. To compensate for this, we propose a Distributionally Robust Federated Averaging (DRFA) algorithm that employs a novel snapshotting scheme to approximate the accumulation of history gradients of the mixing parameter. We analyze the convergence rate of DRFA in both convex-linear and nonconvex-linear settings. We also generalize the proposed idea to objectives with regularization on the mixture parameter and propose a proximal variant, dubbed as DRFA-Prox, with provable convergence rates. We also analyze an alternative optimization method for regularized case in strongly-convex-strongly-concave and non-convex (under PL condition)-strongly-concave settings. To the best of our knowledge, this paper is the first to solve distributionally robust federated learning with reduced communication, and to analyze the efficiency of local descent methods on distributed minimax problems. We give corroborating experimental evidence for our theoretical results in federated learning settings.

## 1   Introduction

Federated learning (FL) has been a key learning paradigm to train a centralized model from an extremely large number of devices/users without accessing their local data [21]. A commonly used approach is to aggregate the individual loss functions usually weighted proportionally to their sample sizes and solve the following optimization problem in a distributed manner:

$$\min_{\boldsymbol{w} \in \mathcal{W}} F(\boldsymbol{w}) := \sum_{i=1}^{N} \frac{n_i}{n} \left\{ f_i(\boldsymbol{w}) := \mathbb{E}_{\xi \sim \mathcal{P}_i}[\ell(\boldsymbol{w}; \xi)] \right\}, \tag{1}$$

where $N$ is number of clients each with $n_i$ training samples drawn from some unknown distribution $\mathcal{P}_i$ (possibly different from other clients), $f_i(\boldsymbol{w})$ is the local objective at device $i$ for a given loss function $\ell$, $\mathcal{W}$ is a closed convex set, and $n$ is total number of samples.

In a federated setting, in contrast to classical distributed optimization, in solving the optimization problem in Eq. 1, three key challenges need to be tackled including i) communication efficiency, ii) the low participation of devices, and iii) heterogeneity of local data shards. To circumvent the communication bottleneck, an elegant idea is to periodically average locally evolving models as employed in FedAvg algorithm [34]. Specifically, each local device optimizes its own model for $\tau$ local iterations using SGD, and then a subset of devices is selected by the server to communicate their models for averaging. This approach, which can be considered as a variant of local SGD [44, 13, 14] but with partial participation of devices, can significantly reduce the number of communication rounds, as demonstrated both empirically and theoretically in various studies [26, 20, 12, 15, 46].

While being compelling from the communication standpoint, FedAvg does not necessarily tackle the *data heterogeneity* concern in FL. In fact, it has been shown that the generalization capability of the central model learned by FedAvg, or any model obtained by solving Eq. 1 in general, is inevitably plagued by increasing the diversity among local data distributions [24, 18, 12]. This is mainly due to the fact the objective in Eq. 1 assumes that all local data are sampled from the same distribution, but in a federated setting, local data distributions can significantly vary from the average distribution. Hence, while the global model enjoys a good *average performance*, its performance often degrades significantly on local data when the distributions drift dramatically.

To mitigate the data heterogeneity issue, one solution is to personalize the global model to local distributions. A few notable studies [8, 32] pursued this idea and proposed to learn a mixture of the global and local models. While it is empirically observed that the per-device mixture model can reduce the generalization error on local distributions compared to the global model, however, the learned global model still suffers from the same issues as in FedAvg, which limits its adaptation to newly joined devices. An alternative solution is to learn a model that has uniformly good performance over almost all devices by minimizing the agnostic (distributionally robust) empirical loss:

$$\min_{\boldsymbol{w} \in \mathcal{W}} \max_{\boldsymbol{\lambda} \in \Lambda} F(\boldsymbol{w}, \boldsymbol{\lambda}) := \sum_{i=1}^{N} \lambda_i f_i(\boldsymbol{w}), \tag{2}$$

where $\boldsymbol{\lambda} \in \Lambda \doteq \{\boldsymbol{\lambda} \in \mathbb{R}_+^N : \sum_{i=1}^{N} \lambda_i = 1\}$ is the global weight for each local loss function.

The main premise is that by minimizing the robust empirical loss, the learned model is guaranteed to perform well over the worst-case combination of empirical local distributions, i.e., limiting the reliance to only a fixed combination of local objectives[1]. Mohri et al. [35] was among the first to introduce the agnostic loss into federated learning, and provided convergence rates for convex-linear and strongly-convex-strongly-concave functions. However, in their setting, the server has to communicate with local user(s) at each iteration to update the global mixing parameter $\boldsymbol{\lambda}$, which hinders its scalability due to communication cost.

The aforementioned issues, naturally leads to the following question: *Can we propose a provably communication efficient algorithm that is also distributionally robust?* The purpose of this paper is to give an affirmative answer to this question by proposing a **Distributionally Robust Federated Averaging** (DRFA) algorithm that is distributionally robust, while being communication-efficient via periodic averaging, and partial node participation, as we show both theoretically and empirically. From a high-level algorithmic perspective, we develop an approach to analyze minimax optimization methods where model parameter $\boldsymbol{w}$ is trained distributedly at local devices, and mixing parameter $\boldsymbol{\lambda}$ is only updated at server periodically. Specifically, each device optimizes its model locally, and a subset of them are adaptively sampled based on $\boldsymbol{\lambda}$ to perform model averaging. We note that since $\boldsymbol{\lambda}$ is updated only at synchronization rounds, it will inevitably hurt the convergence rate. Our key technical contribution is the introduction and analysis of a *randomized snapshotting schema* to approximate the accumulation of history of local gradients to update $\boldsymbol{\lambda}$ as to entail good convergence.

**Contributions.** We summarize the main contributions of our work as follows:

- To the best of our knowledge, the proposed DRFA algorithm is the first to solve distributionally robust optimization in a communicationally efficient manner for federated learning, and to give theoretical analysis on heterogeneous (non-IID) data distributions. The proposed idea of decoupling the updating of $\boldsymbol{w}$ from $\boldsymbol{\lambda}$ can be integrated as a building block into other federated optimization methods, e.g. [18, 23] to yield a distributionally robust solution.

- We derive the convergence rate of our algorithm when loss function is convex in $\boldsymbol{w}$ and linear in $\boldsymbol{\lambda}$, and establish an $O(1/T^{3/8})$ convergence rate with only $O\left(T^{3/4}\right)$ communication rounds. For nonconvex loss, we establish convergence rate of $O(1/T^{1/8})$ with only $O\left(T^{3/4}\right)$ communication rounds. Compared to [35], we significantly reduce the communication rounds.

- For the regularized objectives, we propose a variant algorithm, dubbed as DRFA-Prox, and prove that it enjoys the same convergence rate as DRFA. We also analyze an alternative method for optimizing regularized objective and derive the convergence rate in strongly-convex-strongly-concave and non-convex (under PL condition)-strongly-concave settings.

- We demonstrate the practical efficacy of the proposed algorithm over competitive baselines through experiments on federated datasets.

## 2 Related Work

**Federated Averaging.** Recently, many federated methods have been considered in the literature. FedAvg, as a variant of local GD/SGD, is firstly proposed in [34] to alleviate the communication bottleneck in FL. The first convergence analysis of local SGD on strongly-convex smooth loss functions has established in [44] by showing an $O\left(1/T\right)$ rate with only $O(\sqrt{T})$ communication rounds. The analysis of the convergence of local SGD for nonconvex functions and its adaptive variant is proposed in [13]. The extension to heterogeneous data allocation and general convex functions, with a tighter bound, is carried out in [19]. [12] analyzed local GD and SGD on nonconvex loss functions as well as networked setting in a fully decentralized setting. The recent work [26] analyzes the convergence of FedAvg under non-iid data for strongly convex functions. In [47, 46], Woodworth et al compare the convergence rate of local SGD and mini-batch SGD, under homogeneous and heterogeneous settings respectively.

**Distributionally Robust Optimization.** There is a rich body of literature on Distributionally Robust Optimization (DRO), and here, we try to list the most closely related work. DRO is an effective approach to deal with the imbalanced or non-iid data [37, 38, 9, 50, 9, 35], which is usually formulated as a minimax problem. A bandit mirror descent algorithm to solve the DRO minimax problem is proposed in [37] . Another approach is to minimize top-k losses in the finite sum to achieves the distributional robustness [9]. The first proposal of the DRO in federated learning is [35], where they advocate minimizing the maximum combination of empirical losses to mitigate data heterogeneity.

**Smooth Minimax Optimization.** Another related line of work to this paper is the minimax optimization. One popular primal-dual optimization method is *(stochastic) gradient descent ascent* or (S)GDA for short. The first work to prove that (S)GDA can converge efficiently on nonconvex-concave objectives is [29]. Other classic algorithms for the minimax problem are extra gradient descent (EGD) [22] and optimistic gradient descent (OGD), which are widely studied and applied in machine learning (e.g., GAN training [11, 6, 31, 28]). The algorithm proposed in [45] combines the ideas of mirror descent and Nesterov's accelerated gradient descent (AGD) [40], to achieve $\tilde{O}\left(1/T^2\right)$ rate on strongly-convex-concave functions, and $\tilde{O}\left(1/T^{1/3}\right)$ rate on nonconvex-concave functions. A proximally guided stochastic mirror descent and variance reduction gradient method (PGSMD/PGSVRG) for nonconvex-concave optimization is proposed in [42]. Recently, an algorithm using AGD as a building block is designed in [30], showing a linear convergence rate on strongly-convex-strongly-concave objective, which matches with the theoretical lower bound [49]. The decentralized minimax problem is studied in [43, 33, 31], however, none of these works study the case where one variable is distributed and trained locally, and the other variable is updated periodically, similar to our proposal.

## 3 Distributionally Robust Federated Averaging

We consider a federated setting where $N$ users aim to learn a global model in a collaborative manner without exchanging their data with each other. However, users can exchange information via a server that is connected to all users. Recall that the distributionally robust optimization problem can be formulated as $\min_{\boldsymbol{w}\in\mathcal{W}} \max_{\boldsymbol{\lambda}\in\Lambda} F(\boldsymbol{w},\boldsymbol{\lambda}) := \sum_{i=1}^{N} \lambda_i f_i(\boldsymbol{w})$, where $f_i(\boldsymbol{w})$ is the local objective function corresponding to user $i$, which is often defined as the empirical or true risk over its local data. As mentioned earlier, we address this problem in a federated setting where we assume that $i$th local data shard is sampled from a local distribution $\mathcal{P}_i$– possibly different from the distribution of other data shards. Our goal is to train a central model $\boldsymbol{w}$ with limited communication rounds. We will start with this simple setting where the global objective is linear in the mixing parameter $\boldsymbol{\lambda}$, and will show in Section 5 that our algorithm can also provably optimize regularized objectives where a functional constraint is imposed on the mixing parameter, with a slight difference in the scheme to update $\boldsymbol{\lambda}$.

### 3.1 The proposed algorithm

To solve the aforementioned problem, we propose DRFA algorithm as summarized in Algorithm 1, which consists of two main modules: local model updating and periodic mixture parameter synchronization. The local model updating is similar to the common local SGD [44] or FedAvg [34], however, there is a subtle difference in selecting the clients as we employ an adaptive sampling schema. To formally present the steps of DRFA, let us define $S$ as the rounds of communication between server and users and $\tau$ as the number of local updates that each user runs between two consecutive rounds of communication. We use $T = S\tau$ to denote the total number of iterations the optimization proceeds.

**Algorithm 1:** Distributionally Robust Federated Averaging (DRFA)

---

**Input:** $N$ clients , synchronization gap $\tau$, total number of iterations $T$, $S = T/\tau$, learning rates $\eta, \gamma$, sampling size $m$, initial model $\bar{w}^{(0)}$ and initial $\boldsymbol{\lambda}^{(0)}$.

**Output:** Final solutions $\hat{w} = \frac{1}{mT} \sum_{t=1}^{T} \sum_{i \in \mathcal{D}^{(\lfloor \frac{t}{\tau} \rfloor)}} w_i^{(t)}$, $\hat{\boldsymbol{\lambda}} = \frac{1}{S} \sum_{s=0}^{S-1} \boldsymbol{\lambda}^{(s)}$, or (2) $w^T, \boldsymbol{\lambda}^S$.

  1: **for** $s = 0$ to $S - 1$ **do**
  2:     Server **samples** $\mathcal{D}^{(s)} \subset [N]$ according to $\boldsymbol{\lambda}^{(s)}$ with size of $m$
  3:     Server **samples** $t'$ from $s\tau + 1, \dots, (s+1)\tau$ uniformly at random
  4:     Server **broadcasts** $\bar{w}^{(s)}$ and $t'$ to all clients $i \in \mathcal{D}^{(s)}$

  5:     **for** clients $i \in \mathcal{D}^{(s)}$ **parallel do**
  6:         Client **sets** $w_i^{(s\tau)} = \bar{w}^{(s)}$
  7:         **for** $t = s\tau, \dots, (s+1)\tau - 1$ **do**
  8:             $w_i^{(t+1)} = \prod_{\mathcal{W}} \left( w_i^{(t)} - \eta \nabla f_i(w_i^{(t)}; \xi_i^{(t)}) \right)$
  9:         **end for**
 10:     **end for**
 11:     Client $i \in \mathcal{D}^{(s)}$ **sends** $w_i^{((s+1)\tau)}$ and $w_i^{(t')}$ back to the server

 12:     Server **computes** $\bar{w}^{(s+1)} = \frac{1}{m} \sum_{i \in \mathcal{D}^{(s)}} w_i^{((s+1)\tau)}$
 13:     Server **computes** $w^{(t')} = \frac{1}{m} \sum_{i \in \mathcal{D}^{(s)}} w_i^{(t')}$
 14:     Server uniformly samples a subset $\mathcal{U} \subset [N]$ of clients with size $m$        // Update $\boldsymbol{\lambda}$
 15:     Server **broadcasts** $w^{(t')}$ to each client $i \in \mathcal{U}$, compute $f_i(w^{(t')}; \xi_i)$ over a local minibatch
 16:     Make $N$-dimensional vector $v$: $v_i = \frac{N}{m} f_i(w^{(t')}; \xi_i)$ if $i \in \mathcal{U}$, otherwise $v_i = 0$
 17:     Server updates $\boldsymbol{\lambda}^{(s+1)} = \prod_{\Lambda} \left( \boldsymbol{\lambda}^{(s)} + \tau\gamma v \right)$

 18: **end for**

---

**Periodic model averaging via adaptive sampling.** Let $\bar{w}^{(s)}$ and $\boldsymbol{\lambda}^{(s)}$ denote the global primal and dual parameters at server after synchronization stage $s - 1$, respectively. At the beginning of the $s$th communication stage, server selects $m$ clients $\mathcal{D}^{(s)} \subset [N]$ randomly based on the probability vector $\boldsymbol{\lambda}^{(s)}$ and broadcasts its current model $\bar{w}^{(s)}$ to all the clients $i \in \mathcal{D}^{(s)}$. Each client $i$, after receiving the global model, updates it using local SGD on its own data for $\tau$ iterations. To be more specific, let $w_i^{(t+1)}$ denote the model at client $i$ at iteration $t$ within stage $s$. At each local iteration $t = s\tau, \dots, (s+1)\tau$, client $i$ updates its local model according to the following rule

$$w_i^{(t+1)} = \prod_{\mathcal{W}} \left( w_i^{(t)} - \eta \nabla f_i(w_i^{(t)}; \xi_i^{(t)}) \right),$$

where $\prod_{\mathcal{W}}(\cdot)$ is the projection onto $\mathcal{W}$ and the stochastic gradient is computed on a random sample $\xi_i^{(t)}$ picked from the $i$th local dataset. After $\tau$ local steps, each client sends its current model $w_i^{((s+1)\tau)}$ to the server to compute the next global average primal model $\bar{w}^{(s+1)} = (1/m) \sum_{i \in \mathcal{D}^{(s)}} w_i^{((s+1)\tau)}$. This procedure is repeated for $S$ stages. We note that adaptive sampling not only addresses the scalability issue, but also leads to smaller communication load compared to full participation case.

**Periodic mixture parameter updating.** The global mixture parameter $\boldsymbol{\lambda}$ controls the mixture of different local losses, and can only be updated by server at synchronization stages. The updating scheme for $\boldsymbol{\lambda}$ will be different when the objective function is equipped with or without the regularization on $\boldsymbol{\lambda}$. In the absence of regularization on $\boldsymbol{\lambda}$, the problem is simply linear in $\boldsymbol{\lambda}$. A key observation is that in linear case, the gradient of $\boldsymbol{\lambda}$ only depends on $w$, so we approximate the sum of history gradients over the previous local period (which does not show up in the real dynamic). Indeed, between two synchronization stages, from iterations $s\tau + 1$ to $(s+1)\tau$, in the *fully synchronized* setting [35], we can update $\boldsymbol{\lambda}$ according to

$$\boldsymbol{\lambda}^{(s+1)} = \prod_{\Lambda} \left( \boldsymbol{\lambda}^{(s)} + \gamma \sum_{t=s\tau+1}^{(s+1)\tau} \nabla_{\boldsymbol{\lambda}} F(w^{(t)}, \boldsymbol{\lambda}^{(s)}) \right)$$

where $w^{(t)} = \frac{1}{m} \sum_{i \in \mathcal{D}^{(s)}} w_i^{(t)}$ is the average model at iteration $t$.

To approximate this update, we propose a *random snapshotting* schema as follows. At the beginning of the $s$th communication stage, server samples a random iteration $t'$ (snapshot index) from the range of $s\tau + 1$ to $(s+1)\tau$ and sends it to sampled devices $\mathcal{D}^{(s)}$ along with the global model. After the local updating stage is over, every selected device sends its local model at index $t'$, i.e., $\boldsymbol{w}_i^{(t')}$, back to the server. Then, server computes the average model $\boldsymbol{w}^{(t')} = \frac{1}{|\mathcal{D}^{(s)}|} \sum_{i \in \mathcal{D}^{(s)}} \boldsymbol{w}_i^{(t')}$, that will be used for updating the mixture parameter $\boldsymbol{\lambda}^{(s)}$ to $\boldsymbol{\lambda}^{(s+1)}$ ($\boldsymbol{\lambda}^{(s+1)}$ will be used at stage $s+1$ for sampling another subset of users $\mathcal{D}^{(s+1)}$). To simulate the update we were supposed to do in the fully synchronized setting, server broadcasts $\boldsymbol{w}^{(t')}$ to a set $\mathcal{U}$ of $m$ clients, selected uniformly at random, to stochastically evaluate their local losses $f_i(\cdot), i \in \mathcal{U}$ at $\boldsymbol{w}^{(t')}$ using a random minibatch $\xi_i$ of their local data. After receiving evaluated losses, server will construct the vector $\boldsymbol{v}$ as in Algorithm 1, where $v_i = \frac{N}{m} f_i(\boldsymbol{w}^{(t')}; \xi_i), i \in \mathcal{U}$ to compute a stochastic gradient at dual parameter. We claim that this is an *unbiased estimation* by noting the following identity:

$$\mathbb{E}_{t', \mathcal{U}, \xi_i} \left[ \tau \boldsymbol{v} \right] = \mathbb{E}_{t'} \left[ \tau \nabla_{\boldsymbol{\lambda}} F \left( \boldsymbol{w}^{(t')}, \boldsymbol{\lambda}^{(s)} \right) \right] = \sum_{t=s\tau+1}^{(s+1)\tau} \nabla_{\boldsymbol{\lambda}} F \left( \boldsymbol{w}^{(t)}, \boldsymbol{\lambda}^{(s)} \right). \tag{3}$$

However, the above estimation has a high variance in the order of $O(\tau^2)$, so a crucial question that we need to address is finding the proper choice of $\tau$ to guarantee convergence, while minimizing the overall communication cost. We also highlight that unlike local SGD, the proposed algorithm requires two rounds of communication at each synchronization step for decoupled updating of parameters.

## 4 Convergence Analysis

In this section, we present our theoretical results on the guarantees of the DRFA algorithm for two general class of convex and nonconvex smooth loss functions. All the proofs are deferred to appendix.

**Technical challenge.** Before stating the main results we would like to highlight one of the main theoretical challenges in proving the convergence rate. In particular, a key step in analyzing the local descent methods with periodic averaging is to bound the deviation between local and (virtual) global at each iteration. In minimizing empirical risk (finite sum), [20] gives a tight bound on the deviation of a local model from averaged model which depends on the quantity $\frac{1}{N} \sum_{i=1}^{N} \|\nabla f_i(\boldsymbol{x}^*)\|^2$, where $\boldsymbol{x}^*$ is the minimizer of $\frac{1}{N} \sum_{i=1}^{N} f_i(\boldsymbol{x})$. However, their analysis is not generalizable to minimax setting, as the dynamic of primal-dual method will change the minimizer of $F(\cdot, \boldsymbol{\lambda}^{(s)})$ every time $\boldsymbol{\lambda}^{(s)}$ is updated, which makes the analysis more challenging compared to the average loss case. In light of this and in order to subject heterogeneity of local distributions to a more formal treatment in minimax setting, we introduce a quantity to measure dissimilarity among local gradients.

**Definition 1** (Weighted Gradient Dissimilarity). *A set of local objectives $f_i(\cdot), i = 1, 2, \ldots, N$ exhibit $\Gamma$ gradient dissimilarity defined as $\Gamma := \sup_{\boldsymbol{w} \in \mathcal{W}, \boldsymbol{p} \in \Lambda, i \in [n]}, \sum_{j \in [n]} p_j \|\nabla f_i(\boldsymbol{w}) - \nabla f_j(\boldsymbol{w})\|^2$.*

The above notion is a generalization of gradient dissimilarity, which is employed in the analysis of local SGD in federated setting [27, 8, 26, 46]. This quantity will be zero if and only if all local functions are identical. The obtained bounds will depend on the gradient dissimilarity as local updates only employ samples from local data with possibly different statistical realization.

We now turn to analyzing the convergence of the proposed algorithm. Before, we make the following customary assumptions:

**Assumption 1** (Smoothness/Gradient Lipschitz). *Each component function $f_i(\cdot), i = 1, 2, \ldots, N$ and global function $F(\cdot, \cdot)$ are $L$-smooth, which implies: $\|\nabla f_i(\boldsymbol{x}_1) - \nabla f_i(\boldsymbol{x}_2)\| \leq L\|\boldsymbol{x}_1 - \boldsymbol{x}_2\|, \forall i \in [N], \forall \boldsymbol{x}_1, \boldsymbol{x}_2$ and $\|\nabla F(\boldsymbol{x}_1, \boldsymbol{y}_1) - \nabla F(\boldsymbol{x}_2, \boldsymbol{y}_2)\| \leq L\|(\boldsymbol{x}_1, \boldsymbol{y}_1) - (\boldsymbol{x}_2, \boldsymbol{y}_2)\|, \forall (\boldsymbol{x}_1, \boldsymbol{y}_1), (\boldsymbol{x}_2, \boldsymbol{y}_2)$.*

**Assumption 2** (Gradient Boundedness). *The gradient w.r.t $\boldsymbol{w}$ and $\boldsymbol{\lambda}$ are bounded, i.e., $\|\nabla f_i(\boldsymbol{w})\| \leq G_w$ and $\|\nabla_{\boldsymbol{\lambda}} F(\boldsymbol{w}, \boldsymbol{\lambda})\| \leq G_{\lambda}$.*

**Assumption 3** (Bounded Domain). *The diameters of $\mathcal{W}$ and $\Lambda$ are bounded by $D_{\mathcal{W}}$ and $D_{\Lambda}$.*

**Assumption 4** (Bounded Variance). *Let $\tilde{\nabla} F(\boldsymbol{w}; \boldsymbol{\lambda})$ be stochastic gradient for $\boldsymbol{\lambda}$, which is the $N$-dimensional vector such that the $i$th entry is $f_i(\boldsymbol{w}; \xi)$, and the rest are zero. Then we assume $\|\nabla f_i(\boldsymbol{w}; \xi) - \nabla f_i(\boldsymbol{w})\| \leq \sigma_w^2, \forall i \in [N]$ and $\|\tilde{\nabla} F(\boldsymbol{w}; \boldsymbol{\lambda}) - \nabla F(\boldsymbol{w}; \boldsymbol{\lambda})\| \leq \sigma_{\lambda}^2$.*

## 4.1 Convex losses

The following theorem establishes the convergence rate of primal-dual gap for convex objectives.

**Theorem 1.** *Let each local function $f_i$ be convex, and global function $F$ be linear in $\boldsymbol{\lambda}$. Assume the conditions in Assumptions 1-4 hold. If we optimize (2) using Algorithm 1 with synchronization gap $\tau = \frac{T^{1/4}}{\sqrt{m}}$, learning rates $\eta = \frac{1}{4L\sqrt{T}}$ and $\gamma = \frac{1}{T^{5/8}}$, for the returned solutions $\hat{\boldsymbol{w}}$ and $\hat{\boldsymbol{\lambda}}$ it holds that*

$$\max_{\boldsymbol{\lambda}\in\Lambda}\mathbb{E}[F(\hat{\boldsymbol{w}},\boldsymbol{\lambda})] - \min_{\boldsymbol{w}\in\mathcal{W}}\mathbb{E}[F(\boldsymbol{w},\hat{\boldsymbol{\lambda}})] \leq O\Big(\frac{D_{\mathcal{W}}^2 + G_w^2}{\sqrt{T}} + \frac{D_{\Lambda}^2}{T^{3/8}} + \frac{G_{\lambda}^2}{m^{1/2}T^{3/8}} + \frac{\sigma_{\lambda}^2}{m^{3/2}T^{3/8}} + \frac{\sigma_w^2 + \Gamma}{m\sqrt{T}}\Big).$$

The proof of Theorem 1 is deferred to Appendix C. Since we update $\boldsymbol{\lambda}$ only at the synchronization stages, it will almost inevitably hurt the convergence. The original agnostic federated learning [35] using SGD can achieve an $O(1/\sqrt{T})$ convergence rate, but we achieve a slightly slower rate $O\left(1/T^{3/8}\right)$ to reduce the communication complexity from $O(T)$ to $O(T^{3/4})$. Indeed, we trade $O(T^{1/8})$ convergence rate for $O(T^{1/4})$ communication rounds. As we will show in the proof, if we choose $\tau$ to be a constant, then we recover the same $O(1/\sqrt{T})$ rate as [35]. Also, the dependency of the obtained rate does not demonstrate a linear speedup in the number of sampled workers $m$. However, increasing $m$ will also accelerate the rate, but does not affect the dominating term. We leave tightening the obtained rate to achieve a linear speedup in terms of $m$ as an interesting future work.

## 4.2 Nonconvex losses

We now proceed to state the convergence in the case where local objectives $f_i, i \in [N]$ are nonconvex, e.g., neural networks. Since $f_i$ is no longer convex, the primal-dual gap is not a meaningful quantity to measure the convergence. Alternatively, following the standard analysis of nonconvex minimax optimization, one might consider the following functions to facilitate the analysis.

**Definition 2.** *We define function $\Phi(\cdot)$ at any primal parameter $\boldsymbol{w}$ as:*

$$\Phi(\boldsymbol{w}) := F(\boldsymbol{w},\boldsymbol{\lambda}^*(\boldsymbol{w})), \quad where \ \boldsymbol{\lambda}^*(\boldsymbol{w}) := \arg\max_{\boldsymbol{\lambda}\in\Lambda}F(\boldsymbol{w},\boldsymbol{\lambda}). \tag{4}$$

However, as argued in [29], on nonconvex-concave(linear) but not strongly-concave objective, directly using $\|\nabla\Phi(\boldsymbol{w})\|$ as convergence measure is still difficult for analysis. Hence, Moreau envelope of $\Phi$ can be utilized to analyze the convergence as used in several recent studies [7, 29, 42].

**Definition 3** (Moreau Envelope). *A function $\Phi_p(\boldsymbol{x})$ is the $p$-Moreau envelope of a function $\Phi$ if $\Phi_p(\boldsymbol{x}) := \min_{\boldsymbol{w}\in\mathcal{W}}\left\{\Phi(\boldsymbol{w}) + \frac{1}{2p}\|\boldsymbol{w}-\boldsymbol{x}\|^2\right\}.$*

We will use $1/2L$-Moreau envelope of $\Phi$, following the setting in [29, 42], and state the convergence rates in terms of $\|\nabla\Phi_{1/2L}(\boldsymbol{w})\|$.

**Theorem 2.** *Assume each local function $f_i$ is nonconvex, and global function $F$ is linear in $\boldsymbol{\lambda}$. Also, assume the conditions in Assumptions 1-4 hold. If we optimize (2) using Algorithm 1 with synchronization gap $\tau = T^{1/4}$, letting $\boldsymbol{w}^t = \frac{1}{m}\sum_{\mathcal{D}^{(\lfloor\frac{t}{\tau}\rfloor)}}\boldsymbol{w}_i^{(t)}$ to denote the virtual average model at $t$th iterate, by choosing $\eta = \frac{1}{4LT^{3/4}}$ and $\gamma = \frac{1}{\sqrt{T}}$, we have:*

$$\frac{1}{T}\sum_{t=1}^T\mathbb{E}\left[\|\nabla\Phi_{1/2L}(\boldsymbol{w}^t)\|^2\right] \leq O\left(\frac{D_{\Lambda}^2}{T^{1/8}} + \frac{\sigma_{\lambda}^2}{mT^{1/4}} + \frac{G_{\lambda}^2}{T^{1/4}} + \frac{G_w\sqrt{G_w^2 + \sigma_w^2}}{T^{1/8}} + \frac{D_{\mathcal{W}}(\sigma_w + \sqrt{\Gamma})}{T^{1/2}}\right).$$

The proof of Theorem 2 is deferred to Appendix D. We obtain an $O\left(1/T^{1/8}\right)$ rate here, with $O\left(1/T^{3/4}\right)$ communication rounds. Compared to SOTA algorithms proposed in [29, 42] in nonconvex-concave setting which achieves an $O\left(1/T^{1/4}\right)$ rate in a single machine setting, our algorithm is distributed and communication efficient. Indeed, we trade $O\left(1/T^{1/8}\right)$ rate for saving $O\left(1/T^{1/4}\right)$ communications. One thing worth noticing is that in [29], it is proposed to use a smaller step size for the primal variable than dual variable, while here we choose a small step size for dual variable too. That is mainly because the approximation of dual gradients in our setting introduces a large variance which necessities to employ smaller rate to compensate for high variance. Also, the number of participated clients will not accelerate the leading term, unlike vanilla local SGD or its variants [34, 44, 18].

**Algorithm 2:** Distributionally Robust Federated Averaging: Proximal Method (DRFA-Prox)

**Input:** The algorithm is identical to Algorithm 1 except the updating rule for $\boldsymbol{\lambda}$.

1:      Server uniformly samples a subset $\mathcal{U} \subset [N]$ of clients with size $m$      `// Update λ`
2:      Server **broadcasts** $\boldsymbol{w}^{(t')}$ to each client $i \in \mathcal{U}$
3:      Each client $i \in \mathcal{U}$ computes $f_i(\boldsymbol{w}^{(t')}; \xi_i)$ over a local minibatch $\xi_i$ and sends to server
4:      Server computes $N$-dimensional vector $\boldsymbol{v}$: $v_i = \frac{N}{m} f_i(\boldsymbol{w}^{(t')}; \xi_i)$ if $i \in \mathcal{U}$, otherwise $v_i = 0$
5:      Server updates $\boldsymbol{\lambda}^{(s+1)} = \arg\max_{\boldsymbol{u} \in \Lambda} \left\{ \tau g(\boldsymbol{u}) - \frac{1}{2\gamma} \|\boldsymbol{\lambda}^{(s)} + \gamma\tau\boldsymbol{v} - \boldsymbol{u}\|^2 \right\}.$

## 5    DRFA-Prox: Optimizing Regularized Objective

As mentioned before, our algorithm can be generalized to impose a regularizer on $\boldsymbol{\lambda}$ captured by a regularization function $g(\boldsymbol{\lambda})$ and to solve the following minimax optimization problem:

$$\min_{\boldsymbol{w} \in \mathcal{W}} \max_{\boldsymbol{\lambda} \in \Lambda} F(\boldsymbol{w}, \boldsymbol{\lambda}) := \left\{ f(\boldsymbol{w}, \boldsymbol{\lambda}) := \sum_{i=1}^{N} \lambda_i f_i(\boldsymbol{w}) \right\} + g(\boldsymbol{\lambda}). \tag{5}$$

The regularizer $g(\boldsymbol{\lambda})$ can be introduced to leverage the domain prior, or to make the $\boldsymbol{\lambda}$ update robust to adversary (e.g., the malicious node may send a very large fake gradient of $\boldsymbol{\lambda}$). The choices of $g$ include KL-divergence, optimal transport [16, 36], or $\ell_p$ distance.

In regularized setting, by examining the structure of the gradient w.r.t. $\boldsymbol{\lambda}$, i.e., $\nabla_{\boldsymbol{\lambda}} F(\boldsymbol{w}, \boldsymbol{\lambda}) = \nabla_{\boldsymbol{\lambda}} f(\boldsymbol{w}, \boldsymbol{\lambda}) + \nabla_{\boldsymbol{\lambda}} g(\boldsymbol{\lambda})$., while $\nabla_{\boldsymbol{\lambda}} f(\boldsymbol{w}, \boldsymbol{\lambda})$ is independent of $\boldsymbol{\lambda}$, but $\nabla_{\boldsymbol{\lambda}} g(\boldsymbol{\lambda})$ has dependency on $\boldsymbol{\lambda}$, and consequently our approximation method in Section 3 is not fully applicable here. Inspired by the proximal gradient methods [2, 39, 3], which is widely employed in the problems where the gradient of the regularized term is hard to obtain, we adapt a similar idea, and propose a proximal variant of DRFA, called DRFA-Prox, to tackle regularized objectives. In DRFA-Prox, the only difference is the updating rule of $\boldsymbol{\lambda}$ as detailed in Algorithm 2. We still employ the gradient approximation in DRFA to estimate history gradients of $\nabla_{\boldsymbol{\lambda}} f$, however we utilize proximity operation to update $\boldsymbol{\lambda}$:

$$\boldsymbol{\lambda}^{(s+1)} = \arg\max_{\boldsymbol{u} \in \Lambda} \left\{ \tau g(\boldsymbol{u}) - \frac{1}{2\gamma} \|\boldsymbol{\lambda}^{(s)} + \gamma\tau\boldsymbol{v} - \boldsymbol{u}\|^2 \right\}.$$

As we will show in the next subsection, DRFA-Prox enjoys the same convergence rate as DRFA, both on convex and nonconvex losses.

### 5.1    Convergence of DRFA-Prox

The following theorems establish the convergence rate of DRFA-Prox for convex and nonconvex objectives in federated setting.

**Theorem 3** (Convex loss). *Let each local function $f_i$ be convex. Assume the conditions in Assumptions 1-4 hold. If we optimize (5) using Algorithm 2 with synchronization gap $\tau = \frac{T^{1/4}}{\sqrt{m}}$, $\eta = \frac{1}{4L\sqrt{T}}$, and $\gamma = \frac{1}{T^{5/8}}$, for the returned solutions $\hat{\boldsymbol{w}}$ and $\hat{\boldsymbol{\lambda}}$ it holds that:*

$$\min_{\boldsymbol{w} \in \mathcal{W}} \max_{\boldsymbol{\lambda} \in \Lambda} \mathbb{E}[F(\hat{\boldsymbol{w}}, \boldsymbol{\lambda}) - F(\boldsymbol{w}, \hat{\boldsymbol{\lambda}})] \leq O\left( \frac{D_{\mathcal{W}}^2 + G_w^2}{\sqrt{T}} + \frac{D_{\Lambda}^2}{T^{3/8}} + \frac{G_{\lambda}^2}{m^{1/2} T^{3/8}} + \frac{\sigma_{\lambda}^2}{m^{3/2} T^{3/8}} + \frac{\sigma_w^2 + \Gamma}{m\sqrt{T}} \right).$$

The proof of Theorem 3 is deferred to Appendix E.1. Clearly, we obtain a convergence rate of $O\left(1/T^{3/8}\right)$, which is same as rate obtained in Theorem 1 for DRFA in non-regularized case.

**Theorem 4** (Nonconvex loss). *Assume each local function $f_i$ is nonconvex. Also, assume the conditions in Assumptions 1-4 hold. If we optimize (5) using Algorithm 2 with synchronization gap $\tau = T^{1/4}$, letting $\boldsymbol{w}^t = \frac{1}{m} \sum_{\mathcal{D}(\lfloor \frac{t}{\tau} \rfloor)} \boldsymbol{w}_i^{(t)}$ to denote the virtual average model at tth iterate, by choosing $\eta = \frac{1}{4LT^{3/4}}$ and $\gamma = \frac{1}{\sqrt{T}}$, we have:*

$$\frac{1}{T} \sum_{t=1}^{T} \mathbb{E}\left[ \|\nabla \Phi_{1/2L}(\boldsymbol{w}^t)\|^2 \right] \leq O\left( \frac{D_{\Lambda}^2}{T^{1/8}} + \frac{\sigma_{\lambda}^2}{m T^{1/4}} + \frac{G_{\lambda}^2}{T^{1/4}} + \frac{G_w \sqrt{G_w^2 + \sigma_w^2}}{T^{1/8}} + \frac{D_{\mathcal{W}}(\sigma_w + \sqrt{\Gamma})}{T^{1/2}} \right).$$

The proof of Theorem 4 is deferred to Appendix E.2. Note that, we recover the same convergence rate as DRFA on nonconvex losses (Theorem 5). However, we should remark that solving the proximal problem will take extra computation time, which is not reflected in the convergence rate.

## 5.2 An alternative algorithm for regularized objective

Here we present an alternative method similar to vanilla AFL [35] to optimize regularized objective (5), where we choose to do the full batch gradient ascent for $\boldsymbol{\lambda}$ every $\tau$ iterations according to $\boldsymbol{\lambda}^{(s+1)} = \prod_{\Lambda} \left( \boldsymbol{\lambda}^{(s)} + \gamma \nabla_{\lambda} F \left( \bar{\boldsymbol{w}}^{(s)}, \boldsymbol{\lambda}^{(s)} \right) \right)$. We establish convergence rates in terms of $\Phi(\boldsymbol{w})$ as in Definition 2, under assumption that $F(\cdot, \boldsymbol{\lambda})$ is strongly-convex or satisfies PL-condition [17] in $\boldsymbol{w}$, and strongly-concave in $\boldsymbol{\lambda}$. Due to lack of space, we present a summary of the rates and defer the exact statements to Appendix B and the proofs to Appendices F and G.

**Strongly-convex-strongly-concave case.** In this setting, we obtain an $\tilde{O}\left(\tau/T\right)$ rate. If we choose $\tau = 1$, which is fully synchronized SGDA, then we recover the same rate $\tilde{O}\left(1/T\right)$ as in [35]. If we choose $\tau$ to be $O(\sqrt{T/m})$, we recover the rate $\tilde{O}\left(1/\sqrt{mT}\right)$, which achieves a linear speedup in the number of sampled workers (see Theorem 5 in Appendix B).

**Nonconvex (PL condition)-strongly-concave case.** We also provide the convergence analysis when $F$ is nonconvex but satisfying the PL condition [17] in $\boldsymbol{w}$, and strongly concave in $\boldsymbol{\lambda}$. In this setting, we also obtain an $\tilde{O}\left(\tau/T\right)$ convergence rate which is slightly worse than that of strongly-convex-strongly-concave case. The best known result of non-distributionally robust version of FedAvg on PL condition is $O(1/T)$ [12], with $O(T^{1/3})$ communication rounds. It turns out that we trade some convergence rates to guarantee worst-case performance (see Theorem 6 in Appendix B).

# 6 Experiments

In this section, we empirically verify DRFA and compare its performance to other baselines. More experimental results are discussed in the Appendix A. We implement our algorithm based on `Distributed` API of PyTorch [41] using `MPI` as our main communication interface, and on an `Intel Xeon E5-2695` CPU with 28 cores. We use three datasets, namely, Fashion MNIST [48], Adult [1], and Shakespeare [4] datasets. The code repository used for these experiments can be found at: https://github.com/MLOPTPSU/TorchFed/

**Synchronization gap.** To show the effects of synchronization gap on DRFA algorithm, we run the first experiment on the Fashion MNIST dataset with logistic regression as the model. We run the experiment with 10 devices and a server, where each device has access to only one class of data, making it distributionally heterogeneous. We use different synchronization gaps of $\tau \in \{5, 10, 15\}$, and set $\eta = 0.1$ and $\gamma = 8 \times 10^{-3}$. The results are depicted in Figure 1, where out of all the test accuracies on each single local distribution, we report the worst one as the worst distribution accuracy. Based on our optimization scheme, we aim at optimizing the worst distribution accuracy (or loss), thus the measure depicted in Figure 1 is in accordance with our goal in the optimization. It can be inferred that the smaller the synchronization gap is, the fewer number of iterations required to achieve 50% accuracy in the worst distribution (Figure 1(a)). However, the larger synchronization gap needs fewer number of communication and shorter amount of time to achieve 50% accuracy in the worst distribution (Figure 1(b) and 1(c)).

**Comparison with baselines.** From the algorithmic point of view, the AFL algorithm [35] is a special case of our DRFA algorithm, by setting the synchronization gap $\tau = 1$. Hence, the first experiment suggests that we can increase the synchronization gap and achieve the same level of worst accuracy among distributions with fewer number of communications. In addition to AFL, q-FedAvg proposed by Li et al. [25] aims at balancing the performance among different clients, and hence, improving the worst distribution accuracy. In this part, we compare DRFA with AFL, q-FedAVG, and FedAvg.

To compare them, we run our algorithm, as well as AFL, q-FedAvg and FedAvg on Fashion MNIST dataset with logistic regression model on 10 devices, each of which has access to one class of data. We set $\eta = 0.1$ for all algorithms, $\gamma = 8 \times 10^{-3}$ for DRFA and AFL, and $q = 0.2$ for q-FedAvg. The batch size is 50 and synchronization gap is $\tau = 10$. Figure 2(b) shows that AFL can reach to the 50% worst distribution accuracy with fewer number of local iterations, because it updates the primal and dual

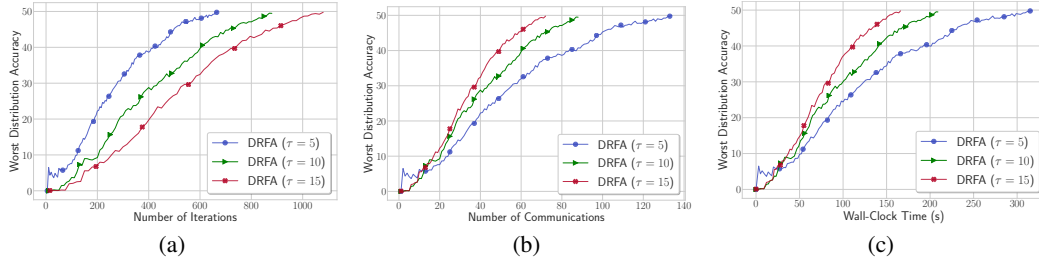

(a)                                    (b)                                    (c)

Figure 1: Comparing the effects of synchronization gap on the DRFA algorithm on the Fashion MNIST dataset with a logistic regression model. The figures are showing the worst distribution accuracy during the training.

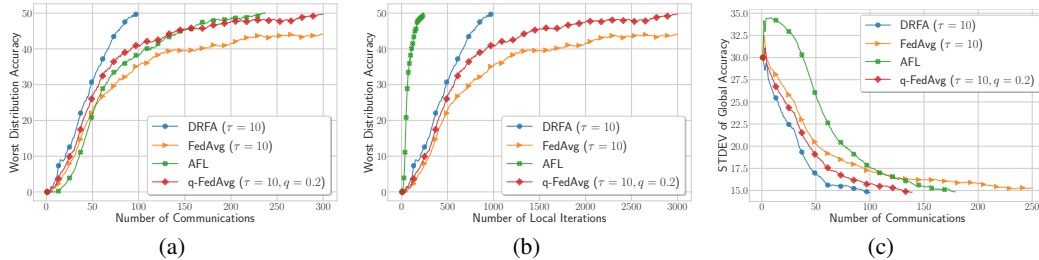

(a)                                    (b)                                    (c)

Figure 2: Comparing DRFA algorithm with AFL [35], q-FedAvg [25], and FedAvg on Fashion MNIST dataset with logistic regression. DRFA can achieve the same level of worst distribution accuracy, with fewer number of communication rounds, and hence, lower runtime. It also efficiently decreases the variance among the performance of different nodes with fewer communication rounds.

variables at every iteration. However, Figure 2(a) shows that DRFA outperforms AFL, q-FedAvg and FedAvg in terms of number of communications, and subsequently, wall-clock time required to achieve the same level of worst distribution accuracy (due to much lower number of communication needed).

Note that, q-FedAvg has is very close to AFL in terms of communication rounds, but it is far behind it in terms of local computations. Also, note that FedAvg has the same computation complexity as DRFA and q-FedAvg at each round but cannot reach the 50% accuracy even after 300 rounds of communication. Similar to q-FedAvg, to show how different devices are performing, Figure 2(c) depicts the standard deviation among the accuracy of different clients, which shows the level of fairness of the learned model among different clients. It can be inferred that DRFA can achieve the same level as AFL and q-FedAvg with fewer number of communication rounds, making it more efficient. To compare the average performance of these algorithms, Figure 3 shows the global training accuracy of them over 100 rounds of communication on Fashion MNIST with logistic regression, where DRFA performs as good as FedAvg in this regard. AFL needs more communication rounds to reach to the same level.

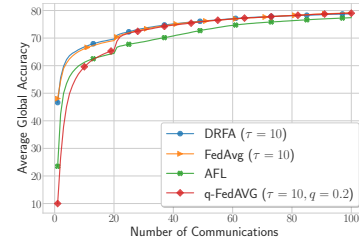

Figure 3: Averag global accuracy for each algorithm for 100 rounds of communication. It shows that DRFA keeps the same level of global accuracy as FedAvg, while it boosts its worst performing distribution accuracy.

## 7   Conclusion

In this paper we propose a communication efficient scheme for distributionally robust federated model training. In addition, we give the first analysis of local SGD in distributed minimax optimization, under general smooth convex-linear, and nonconvex linear, strongly-convex-strongly-concave and nonconvex (PL-condition)-strongly concave settings. The experiments demonstrate the convergence of our method, and the distributional robustness of the learned model. The future work would be improving obtained convergence rates due to gap we observed compared to centralized case. Another interesting question worth exploring will be investigating variance reduction schemes to achieve faster rates, in particular for updating mixing parameter.

## Broader Impact

This work advocates a distributionally robust algorithm for federated learning. The algorithmic solution is designed to preserve the privacy of users, while training a high quality model. The proposed algorithm tries to minimize the maximum loss among worst case distribution over clients' data. Hence, we can ensure that even if the data distribution among users is highly heterogeneous, the trained model is reasonably good for everyone, and not benefiting only a group of clients. This will ensure the fairness in training a global model with respect to every user, and it is vitally important for critical decision making systems such as healthcare. In such a scenario, the model learned by simple algorithms such as FedAvg would have an inconsistent performance over different distributions, which is not acceptable. However, the resulting model from our algorithm will have robust performance over different distributions it has been trained on.

## Acknowledgements

This work has been done using the Extreme Science and Engineering Discovery Environment (XSEDE) resources, which is supported by National Science Foundation under grant number ASC200045. We are also grateful for the GPU donated by NVIDIA that was used in this research.

## Footnotes

[1]Beyond robustness, agnostic loss yields a notion of fairness [35], which is not the focus of present work.

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
