[Supplementary Material · DRFA_SUPP.pdf]

# Supplementary Material:
# Distributionally Robust Federated Averaging

## Table of Contents

# A   Additional Experiments

In this section, we further investigate the effectiveness of the proposed DRFA algorithm. To do so, we use the Adult and Shakespeare datasets.

**Experiments on Adult dataset.** The Adult dataset contains census data, with the target of predicting whether the income is greater or less than $50K$. The data has 14 features from age, race, gender, among others. It has 32561 samples for training distributed across different groups of sensitive features. One of these sensitive features is gender, which has two groups of "male" and "female". The other sensitive feature we will use is the race, where it has 5 groups of "black", "white", "Asian-Pac-Islander", "Amer-Indian-Eskimo", and "other". We can distribute data among nodes based on the value of these features, hence make it heterogeneously distributed.

For the first experiment, we distribute the training data across 10 nodes, 5 of which contain only data from the female group and the other 5 have the male group's data. Since the size of different groups' data is not equal, the data distribution is unbalanced among nodes. Figure 4 compares DRFA with AFL [35], q-FedAvg [25], and FedAvg [34] on the Adult dataset, where the data is distributed among the nodes based on the gender feature. We use logistic regression as the loss function, the learning rate is set to $0.1$ and batch size is $50$ for all algorithms, $\gamma$ is set to $0.2$ for both DRFA and AFL, and $q = 0.5$ is tuned for the best results for q-FedAvg. The worst distribution or node accuracy during the communication rounds shows that DRFA can achieve the same level of worst accuracy with a far fewer number of communication rounds, and hence, less overall wall-clock time. However, AFL computational cost is less than that of DRFA. Between each communication rounds DRFA, q-FedAvg and FedAvg have 10 update steps. FedAvg after the same number of communications as AFL still cannot reach the same level of worst accuracy. Figure 4(c) shows the standard deviation of accuracy among different nodes as a measure for the fairness of algorithms. It can be inferred that DRFA efficiently decreases the variance with a much fewer number of communication rounds with respect to other algorithms.

Figure 4: Comparing the worst distribution accuracy on DRFA, AFL, q-FedAvg, and FedAVG on the Adult dataset. We have 10 nodes, and data is distributed among them based on the gender feature. The loss function is logistic regression. DRFA needs a fewer number of communications to reach the same worst distribution accuracy than the AFL and q-FedAvg algorithms. Also, DRFA efficiently decreases the variance of the performance of different clients.

Next, we distribute the Adult data among clients based on the "race" feature, which has 5 different groups. Again the size of data among these groups is not equal and makes the distribution unbalanced. We distribute the data among 10 nodes, where every node has only data from one group of the race feature. For this experiment, we use a nonconvex loss function, where the model is a multilayer perceptron (MLP) with 2 hidden layers, each with $50$ neurons. The first layer has $14$ and the last layer has $2$ neurons. The learning rate is set to $0.1$ and batch size is $50$ for all algorithms, the $\gamma$ is set to $0.2$ for DRFA and AFL, and the $q$ parameter in q-FedAvg is tuned for $0.5$. Figure 5 shows the results of this experiment, where again, DRFA can achieve the same worst-case accuracy with a much fewer number of communications than AFL and q-FedAvg. In this experiment, with the same number of local iterations, AFL still cannot reach to the DRFA performance. In addition, the variance on the performance of different clients in Figure 5(c) suggests that DRFA is more successful than q-FedAvg to balance the performance of clients.

**Experiments on Shakespeare dataset.** Now, we run the same experiments on the Shakespeare dataset. This dataset contains the scripts from different Shakespeare's plays divided based on the

Figure 5: Comparing the worst distribution accuracy on DRFA, AFL, q-FedAvg, and FedAvg with the Adult dataset. We have 10 nodes, and data is distributed among them based on the race feature. The model is an MLP with 2 hidden layers, each with 50 neurons and a cross-entropy loss function. DRFA needs a fewer number of communications to reach the same worst distribution accuracy than the AFL and q-FedAvg algorithms. Moreover, DRFA is more efficient in reducing the performance variance among different clients than q-FedAvg.

Figure 6: Comparing different algorithms on training an RNN on Shakespeare dataset using 100 clients. DRFA and FedAvg outperform the other two algorithms in terms of communication efficiency, however, AFL can achieve the same level with lower computation cost. In the average performance, AFL requires much more communication to reach to the same level as FedAvg and DRFA.

character in each play. The task is to predict the next character in the text, providing the preceding characters. For this experiment, we use 100 clients' data to train our RNN model. The RNN model comprises an embedding layer from 86 characters to 50, followed by a layer of GRU [5] with 50 units. The output is going through a fully connected layer with an output size of 86 and a cross-entropy loss function. We use the batch size of 2 with 50 characters in each batch. The learning rate is optimized to 0.8 for the FedAvg and used for all algorithms. The $\gamma$ is tuned to the 0.01 for AFL and DRFA, and $q = 0.1$ is the best for the q-FedAvg. Figure 6 shows the results of this experiment on the Shakespeare dataset. It can be seen that DRFA and FedAvg can reach to the same worst distribution accuracy compared to AFL and q-FedAvg. The reason that FedAvg is working very well in this particular dataset is that the distribution of data based on the characters in the plays does not make it heterogeneous. In settings close to homogeneous distribution, FedAvg can achieve the best results, with DRFA having a slight advantage over that.

# B Formal Convergence Theory for Alternative Algorithm in Regularized Case

Here, we will present the formal convergence theory of the algorithm we described in Section 5.2, where we use full batch gradient ascent to update $\boldsymbol{\lambda}^{(s)}$. To do so, the server sends the current global model $\bar{\boldsymbol{w}}^{(s)}$ to all clients and each client evaluates the global model on its local data shards and send $f_i(\bar{\boldsymbol{w}}^{(s)})$ back to the server. Then the server can compute the full gradient over dual parameter $\boldsymbol{\lambda}$ and take a gradient ascent (GA) step to update it. The algorithm is named DRFA-GA and described in Algorithm 3. We note that DRFA-GA can be considered as communication-efficient variant of AFL, but without sampling clients to evaluate the gradient at dual parameter. We conduct the convergence analysis on the setting where the regularized term is strongly-concave in $\boldsymbol{\lambda}$, and loss function is

---

**Algorithm 3:** Distributionally Robust Federated Averaging: Gradient Ascent (DRFA-GA)

---

**Input:** $N$ clients , synchronization gap $\tau$, total number of iterations $T$, $S = T/\tau$, learning rates $\eta, \gamma$, sampling size $m$, initial model $\bar{\boldsymbol{w}}^{(0)}$ and initial $\boldsymbol{\lambda}^{(0)}$.

**Output:** Final solutions $\hat{\boldsymbol{w}} = \frac{2}{mT} \sum_{t=T/2}^{T} \sum_{i \in \mathcal{D}^{(\lfloor \frac{t}{\tau} \rfloor)}} \boldsymbol{w}_i^{(t)}$, $\hat{\boldsymbol{\lambda}} = \frac{1}{S} \sum_{s=0}^{S-1} \boldsymbol{\lambda}^{(s)}$, or (2) $\boldsymbol{w}^T, \boldsymbol{\lambda}^S$.

1: **for** $s = 0$ to $S - 1$ **do**
2:     Server **samples** $\mathcal{D}^{(s)} \subset [N]$ according to $\boldsymbol{\lambda}^{(s)}$ with size of $m$
3:     Server **broadcasts** $\bar{\boldsymbol{w}}^{(s)}$ to all clients $i \in \mathcal{D}^{(s)}$

4:     **for** clients $i \in \mathcal{D}^{(s)}$ **parallel do**
5:         Client **sets** $\boldsymbol{w}_i^{(s\tau)} = \bar{\boldsymbol{w}}^{(s)}$
6:         **for** $t = s\tau, \ldots, (s+1)\tau - 1$ **do**
7:             $\boldsymbol{w}_i^{(t+1)} = \prod_{\mathcal{W}} \left( \boldsymbol{w}_i^{(t)} - \eta \nabla f_i(\boldsymbol{w}_i^{(t)}; \xi_i^{(t)}) \right)$
8:         **end for**
9:     **end for**
10:    Client $i \in \mathcal{D}^{(s)}$ **sends** $\boldsymbol{w}_i^{((s+1)\tau)}$ back to the server

11:    Server sends $\bar{\boldsymbol{w}}^{(s)}$ to all clients             // Update $\boldsymbol{\lambda}$
12:    Each client $i \in [N]$ evaluates $\bar{\boldsymbol{w}}^{(s)}$ on its local data and sends $f_i(\bar{\boldsymbol{w}}^{(s)})$ back to server
13:    Server updates $\boldsymbol{\lambda}^{(s+1)} = \prod_{\Lambda} \left( \boldsymbol{\lambda}^{(s)} + \gamma \nabla_{\boldsymbol{\lambda}} F\left( \bar{\boldsymbol{w}}^{(s)}, \boldsymbol{\lambda}^{(s)} \right) \right)$

14:    Server **computes** $\bar{\boldsymbol{w}}^{(s+1)} = \frac{1}{m} \sum_{i \in \mathcal{D}^{(s)}} \boldsymbol{w}_i^{((s+1)\tau)}$
15: **end for**

---

strongly-convex and nonconvex but satisfying Polyak-Łojasiewicz (PL) condition in $\boldsymbol{w}$. So, our theory includes strongly-convex-strongly-concave and nonconvex (PL condition)-strongly-concave cases.

**Strongly-Convex-Strongly-Concave case.** We start by stating the convergence rate when the individual local objectives are strongly convex and the regularizer $g(\boldsymbol{\lambda})$ is strongly concave in $\boldsymbol{\lambda}$, making the global objective $F(\boldsymbol{w}, \boldsymbol{\lambda}) := \sum_{i=1}^{N} \lambda_i f_i(\boldsymbol{w}) + g(\boldsymbol{\lambda})$ also strongly concave in $\boldsymbol{\lambda}$.

**Theorem 5.** *Let each local function $f_i$ be $\mu$-strongly convex, and global function $F$ is $\mu$-strongly concave in $\boldsymbol{\lambda}$. Under Assumptions 1, 2, 3, 4, if we optimize (5) using the DRFA-GA (Algorithm 3) with synchronization gap $\tau$, choosing learning rates as $\eta = \frac{4 \log T}{\mu T}$ and $\gamma = \frac{1}{L}$ and $T \geq \frac{16 \alpha \log T}{\mu}$, where $\alpha = \kappa L + L$, using the averaging scheme $\hat{\boldsymbol{w}} = \frac{2}{mT} \sum_{t=T/2}^{T} \sum_{i \in \mathcal{D}^{(\lfloor \frac{t}{\tau} \rfloor)}} \boldsymbol{w}_i^{(t)}$ we have:*

$$\mathbb{E}[\Phi(\hat{\boldsymbol{w}}) - \Phi(\boldsymbol{w}^*)] = \tilde{O}\left( \frac{\mu D_{\mathcal{W}}^2}{T} + \frac{\kappa^2 L \tau D_{\Lambda}^2}{T} + \frac{\sigma_w^2 + G_w^2}{\mu m T} + \frac{\kappa^2 \tau^2 (\sigma_w^2 + \Gamma)}{\mu T^2} + \frac{\kappa^6 \tau^2 G_w^2}{\mu T^2} \right),$$

*where $\kappa = L/\mu$, and $\boldsymbol{w}^*$ is the minimizer of $\Phi$.*

*Proof.* The proof is given in Section F.       □

**Corollary 1.** *Continuing with Theorem 5, if we choose $\tau = \sqrt{T/m}$, we recover the rate:*

$$\mathbb{E}[\Phi(\hat{\boldsymbol{w}}) - \Phi(\boldsymbol{w}^*)] = \tilde{O}\left( \frac{\kappa^2 L D_{\Lambda}^2}{\sqrt{mT}} + \frac{\mu D_{\mathcal{W}}^2}{T} + \frac{\kappa^2 (\sigma_w^2 + \Gamma) + \kappa^6 G_w^2}{\mu m T} \right).$$

Here we obtain $\tilde{O}\left( \frac{\tau}{T} \right)$ rate in Theorem 5. If we choose $\tau = 1$, which is fully synchronized SGD, then we recover the same rate $\tilde{O}\left( \frac{1}{T} \right)$ as in vanilla agnostic federated learning [35]. If we choose $\tau$ to be $O(\sqrt{T/m})$, we recover the rate $\tilde{O}\left( \frac{1}{\sqrt{mT}} + \frac{1}{mT} \right)$, which can achieve linear speedup with respect to number of sampled workers. The dependency on gradient dissimilarity $\Gamma$ shows that the data heterogeneity will slow down the rate, but will not impact the dominating term.

**Nonconvex (PL condition)-Strongly-Concave Setting.** We provide the convergence analysis under the condition where $F$ is nonconvex but satisfies PL condition in $\boldsymbol{w}$, and strongly concave in $\boldsymbol{\lambda}$.

In the constraint problem, to prove the convergence, we have to consider a generalization of PL condition [17] as formally stated below.

**Definition 4** (($\mu,\eta$)-generalized Polyak-Łojasiewicz (PL)). *The global objective function $F(\cdot, \boldsymbol{\lambda})$ is differentiable and satisfies the ($\mu,\eta$)-generalized Polyak-Łojasiewicz condition with constant $\mu$ if the following holds:*

$$\frac{1}{2\eta^2} \left\| \boldsymbol{w} - \prod_{\mathcal{W}} (\boldsymbol{w} - \eta \nabla_{\boldsymbol{w}} F(\boldsymbol{w}, \boldsymbol{\lambda})) \right\|_2^2 \geq \mu(F(\boldsymbol{w}, \boldsymbol{\lambda}) - \min_{\boldsymbol{w}' \in \mathcal{W}} F(\boldsymbol{w}', \boldsymbol{\lambda})), \forall \boldsymbol{\lambda} \in \Lambda$$

.

**Remark 1.** *When the constraint is absent, it reduces to vanilla PL condition [17]. The similar generalization of PL condition is also mentioned in [17], where they introduce a variant of PL condition to prove the convergence of proximal gradient method. Also we will show that, if $F$ satisfies $\mu$-PL condition in $\boldsymbol{w}$, $\Phi(\boldsymbol{w})$ also satisfies $\mu$-PL condition.*

We now proceed to provide the global convergence of $\Phi$ in this setting.

**Theorem 6.** *Let global function $F$ satisfy ($\mu,\eta$)-generalized PL condition in $\boldsymbol{w}$ and $\mu$-strongly-concave in $\boldsymbol{\lambda}$. Under Assumptions 1,2,3,4, if we optimize (5) using the DRFA-GA (Algorithm 3) with synchronization gap $\tau$, choosing learning rates $\eta = \frac{4\log T}{\mu T}$, $\gamma = \frac{1}{L}$ and $m \geq T$, with the total iterations satisfying $T \geq \frac{8\alpha \log T}{\mu}$ where $\alpha = L + \kappa L$, $\kappa = \frac{L}{\mu}$, we have:*

$$\mathbb{E}\left[\Phi(\boldsymbol{w}^{(T)}) - \Phi(\boldsymbol{w}^*)\right] \leq O\left(\frac{\Phi(\boldsymbol{w}^{(0)}) - \Phi(\boldsymbol{w}^*)}{T}\right) + \tilde{O}\left(\frac{\sigma_w^2 + G_w^2}{\mu T}\right) + \tilde{O}\left(\frac{\kappa^2 L \tau D_\Lambda^2}{T}\right)$$
$$+ \tilde{O}\left(\frac{\kappa^6 \tau^2 G_w^2}{\mu T^2}\right) + \tilde{O}\left(\frac{\kappa^2 \tau^2 (\sigma_w^2 + \Gamma)}{\mu T^2}\right).$$

*where $\boldsymbol{w}^* \in \arg\min_{\boldsymbol{w} \in \mathcal{W}} \Phi(\boldsymbol{w})$.*

*Proof.* The proof is given in Section G. □

**Corollary 2.** *Continuing with Theorem 6, if we choose $\tau = \sqrt{T/m}$, we recover the rate:*

$$\mathbb{E}[\Phi(\hat{\boldsymbol{w}}) - \Phi(\boldsymbol{w}^*)] = \tilde{O}\left(\frac{\kappa^2 L D_\Lambda^2}{\sqrt{T}} + \frac{\Phi(\boldsymbol{w}^{(0)}) - \Phi(\boldsymbol{w}^*)}{T} + \frac{\kappa^2(\sigma_w^2 + \Gamma) + \kappa^6 G_w^2}{\mu T}\right).$$

We obtain $\tilde{O}\left(\frac{\tau}{T}\right)$ convergence rate here, slightly worse than that of strongly-convex-strongly-concave case. We also get linear speedup in the number of sampled workers if properly choose $\tau$. The best known result of non-distributionally robust version of FedAvg on PL condition is $O(\frac{1}{T})$ [12], with $O(T^{1/3})$ communication rounds. It turns out that we trade some convergence rate to guarantee a worst case performance. We would like to mention that, here we require $m$, the number of sampled clients to be a large number, which is the imperfection of our analysis. However, we would note that, this is similar to the analysis in [10] for projected SGD on constrained nonconvex minimization problems, where it is required to employ growing mini-batch sizes with iterations to guarantee convergence to a first-order stationary point (i.e., imposing a constraint on minibatch size based on target accuracy $\epsilon$ which plays a similar rule to $m$ in our case).

## C Proof of Convergence of DRFA for Convex Losses (Theorem 1)

In this section we will present the proof of Theorem 1, which states the convergence of DRFA in convex-linear setting.

### C.1 Preliminary

Before delving into the proof, let us introduce some useful variables and lemmas for ease of analysis. We define a virtual sequence $\{\boldsymbol{w}^{(t)}\}_{t=1}^{T}$ that will be used in our proof, and we also define some

intermediate variables:

$$\boldsymbol{w}^{(t)} = \frac{1}{m} \sum_{i \in \mathcal{D}^{(\lfloor \frac{t}{\tau} \rfloor)}} \boldsymbol{w}_i^{(t)}, \qquad \text{(average model of selected devices)}$$

$$\bar{\boldsymbol{u}}^{(t)} = \frac{1}{m} \sum_{i \in \mathcal{D}^{(\lfloor \frac{t}{\tau} \rfloor)}} \nabla f_i(\boldsymbol{w}_i^{(t)}), \qquad \text{(average full gradient of selected devices)}$$

$$\boldsymbol{u}^{(t)} = \frac{1}{m} \sum_{i \in \mathcal{D}^{(\lfloor \frac{t}{\tau} \rfloor)}} \nabla f_i(\boldsymbol{w}_i^{(t)}; \xi_i^{(t)}) \qquad \text{(average stochastic gradient of selected devices)}$$

$$\bar{\boldsymbol{v}}^{(t)} = \nabla_{\boldsymbol{\lambda}} F(\boldsymbol{w}^{(t)}, \boldsymbol{\lambda}) = \left[ f_1(\boldsymbol{w}^{(t)}), \ldots, f_N(\boldsymbol{w}^{(t)}) \right] \quad \text{(full gradient w.r.t. dual)}$$

$$\bar{\Delta}_s = \sum_{t=s\tau+1}^{(s+1)\tau} \gamma \bar{\boldsymbol{v}}^{(t)},$$

$$\Delta_s = \tau\gamma\boldsymbol{v}, \qquad \text{(see below)}$$

$$\delta^{(t)} = \frac{1}{m} \sum_{i \in \mathcal{D}^{(\lfloor \frac{t}{\tau} \rfloor)}} \left\| \boldsymbol{w}_i^{(t)} - \boldsymbol{w}^{(t)} \right\|^2,$$

where $\boldsymbol{v} \in \mathbb{R}^N$ is the stochastic gradient for dual variable generated by Algorithm 1 for updating $\boldsymbol{\lambda}$, such that $v_i = f_i(\boldsymbol{w}^{(t')}; \xi_i)$ for $i \in \mathcal{U} \subset [N]$ where $\xi_i$ is stochastic minibatch sampled from $i$th local data shard, and $t'$ is the snapshot index sampled from $s\tau + 1$ to $(s + 1)\tau$.

## C.2 Overview of the Proof

The proof techniques consist of analyzing the one-step progress for the virtual iterates $\boldsymbol{w}^{(t+1)}$ and $\boldsymbol{\lambda}^{(s+1)}$, however periodic decoupled updating along with sampling makes the analysis more involved compared to fully synchronous primal-dual schemes for minimax optimization. Let us start from analyzing one iteration on $\boldsymbol{w}$. From the updating rule we can show that

$$\mathbb{E}\|\boldsymbol{w}^{(t+1)} - \boldsymbol{w}\|^2 \leq \mathbb{E}\|\boldsymbol{w}^{(t)} - \boldsymbol{w}\|^2 - 2\eta\mathbb{E}\left[ F(\boldsymbol{w}^{(t)}, \boldsymbol{\lambda}^{(\lfloor \frac{t}{\tau} \rfloor)}) - F(\boldsymbol{w}, \boldsymbol{\lambda}^{(\lfloor \frac{t}{\tau} \rfloor)}) \right]$$
$$+ L\eta\mathbb{E}\left[ \delta^{(t)} \right] + \eta^2 \mathbb{E}\|\bar{\boldsymbol{u}}^{(t)} - \boldsymbol{u}^{(t)}\|^2 + \eta^2 G_w^2.$$

Note that, similar to analysis of local SGD, e.g., [44], the key question is how to bound the deviation $\delta^{(t)}$ between local and (virtual) averaged model. By the definition of gradient dissimilarity, we establish that:

$$\frac{1}{T} \sum_{t=0}^{T} \mathbb{E}\left[ \delta^{(t)} \right] = 10\eta^2\tau^2 \left( \sigma_w^2 + \frac{\sigma_w^2}{m} + \Gamma \right).$$

It turns out the deviation can be upper bounded by variance of stochastic graident, and the gradient dissimilarity. The latter term controls how heterogenous the local component functions are, and it becomes zero when all local functions are identical, which means we are doing minibatch SGD on the same objective function in parallel.

Now we switch to the one iteration analysis on $\boldsymbol{\lambda}$:

$$\mathbb{E}\|\boldsymbol{\lambda}^{(s+1)} - \boldsymbol{\lambda}\|^2 \leq \mathbb{E}\|\boldsymbol{\lambda}^{(s)} - \boldsymbol{\lambda}\|^2$$
$$- \sum_{t=s\tau+1}^{(s+1)\tau} \mathbb{E}[2\gamma(F(\boldsymbol{w}^{(t)}, \boldsymbol{\lambda}^{(s)}) - F(\boldsymbol{w}^{(t)}, \boldsymbol{\lambda}))] + \mathbb{E}\|\bar{\Delta}_s\|^2 + \mathbb{E}\|\Delta_s - \bar{\Delta}_s\|^2.$$

It suffices to bound the variance of $\Delta_s$. Using the identity of independent variables we can prove:

$$\mathbb{E}[\|\Delta_s - \bar{\Delta}_s\|^2] \leq \gamma^2\tau^2\frac{\sigma_\lambda^2}{m}.$$

It shows that the variance depends quadratically on $\tau^2$, and can achieve linear speed up with respect to the number of sampled workers. Putting all pieces together, and doing the telescoping sum will yield the result in Theorem 1.

## C.3 Proof of Technical Lemmas

In this section we are going to present some technical lemmas that will be used in the proof of Theorem 1.

**Lemma 1.** *The stochastic gradient $\boldsymbol{u}^{(t)}$ is unbiased, and its variance is bounded, which implies:*

$$\mathbb{E}_{\xi_i^{(t)}, \mathcal{D}^{(\lfloor \frac{t}{\tau} \rfloor)}} \left[ \boldsymbol{u}^{(t)} \right] = \mathbb{E}_{\mathcal{D}^{(\lfloor \frac{t}{\tau} \rfloor)}} \left[ \bar{\boldsymbol{u}}^{(t)} \right] = \mathbb{E} \left[ \sum_{i=1}^{N} \lambda_i^{(\lfloor \frac{t}{\tau} \rfloor)} \nabla f_i(\boldsymbol{w}_i^{(t)}) \right],$$

$$\mathbb{E} \left[ \|\boldsymbol{u}^{(t)} - \bar{\boldsymbol{u}}^{(t)}\|^2 \right] = \frac{\sigma_w^2}{m}.$$

*Proof.* The unbiasedness is due to the fact that we sample the clients according to $\boldsymbol{\lambda}^{(\lfloor \frac{t}{\tau} \rfloor)}$. The variance term is due to the identity $\mathrm{Var}(\sum_{i=1}^{m} \boldsymbol{X}_i) = \sum_{i=1}^{m} \mathrm{Var}(\boldsymbol{X}_i)$. □

**Lemma 2.** *The stochastic gradient at $\boldsymbol{\lambda}$ generated by Algorithm 1 is unbiased, and its variance is bounded, which implies:*

$$\mathbb{E}[\Delta_s] = \bar{\Delta}_s, \qquad \mathbb{E}[\|\Delta_s - \bar{\Delta}_s\|^2] \le \gamma^2 \tau^2 \frac{\sigma_\lambda^2}{m}. \tag{6}$$

*Proof.* The unbiasedness is due to we sample the workers uniformly. The variance term is due to the identity $\mathrm{Var}(\sum_{i=1}^{m} \boldsymbol{X}_i) = \sum_{i=1}^{m} \mathrm{Var}(\boldsymbol{X}_i)$. □

**Lemma 3** (One Iteration Primal Analysis). *For DRFA, under the same conditions as in Theorem 1, for all $\boldsymbol{w} \in \mathcal{W}$, the following holds:*

$$\mathbb{E}\|\boldsymbol{w}^{(t+1)} - \boldsymbol{w}\|^2 \le \mathbb{E}\|\boldsymbol{w}^{(t)} - \boldsymbol{w}\|^2 - 2\eta \mathbb{E}\left[ F(\boldsymbol{w}^{(t)}, \boldsymbol{\lambda}^{(\lfloor \frac{t}{\tau} \rfloor)}) - F(\boldsymbol{w}, \boldsymbol{\lambda}^{(\lfloor \frac{t}{\tau} \rfloor)}) \right]$$

$$+ L\eta \mathbb{E}\left[ \delta^{(t)} \right] + \eta^2 \mathbb{E}\|\bar{\boldsymbol{u}}^{(t)} - \boldsymbol{u}^{(t)}\|^2 + \eta^2 G_w^2.$$

*Proof.* From the updating rule we have:

$$\mathbb{E}\|\boldsymbol{w}^{(t+1)} - \boldsymbol{w}\|^2 = \mathbb{E}\left\| \prod_{\mathcal{W}} (\boldsymbol{w}^{(t)} - \eta \boldsymbol{u}^{(t)}) - \boldsymbol{w} \right\|^2 \le \mathbb{E}\|\boldsymbol{w}^{(t)} - \eta \bar{\boldsymbol{u}}^{(t)} - \boldsymbol{w}\|^2 + \eta^2 \mathbb{E}\|\bar{\boldsymbol{u}}^{(t)} - \boldsymbol{u}^{(t)}\|^2$$

$$\le \mathbb{E}\|\boldsymbol{w}^{(t)} - \boldsymbol{w}^*\|^2 + \underbrace{\mathbb{E}[-2\eta \langle \bar{\boldsymbol{u}}^{(t)}, \boldsymbol{w}^{(t)} - \boldsymbol{w}^* \rangle]}_{T_1} + \underbrace{\eta^2 \mathbb{E}\|\bar{\boldsymbol{u}}^{(t)}\|^2}_{T_2} + \mathbb{E}\|\bar{\boldsymbol{u}}^{(t)} - \boldsymbol{u}^{(t)}\|^2 \tag{7}$$

We are going to bound $T_1$ first:

$$T_1 = \mathbb{E}_{\mathcal{D}^{(\lfloor \frac{t}{\tau} \rfloor)}} \left[ \frac{1}{m} \sum_{i \in \mathcal{D}^{(\lfloor \frac{t}{\tau} \rfloor)}} \left[ -2\eta \left\langle \nabla f_i(\boldsymbol{w}_i^{(t)}), \boldsymbol{w}^{(t)} - \boldsymbol{w}_i^{(t)} \right\rangle - 2\eta \left\langle \nabla f_i(\boldsymbol{w}_i^{(t)}), \boldsymbol{w}_i^{(t)} - \boldsymbol{w}^* \right\rangle \right] \right] \tag{8}$$

$$\leq \mathbb{E}_{\mathcal{D}^{(\lfloor \frac{t}{\tau} \rfloor)}} \left[ 2\eta \frac{1}{m} \sum_{i \in \mathcal{D}^{(\lfloor \frac{t}{\tau} \rfloor)}} \left[ f_i(\boldsymbol{w}_i^{(t)}) - f_i(\boldsymbol{w}^{(t)}) + \frac{L}{2} \|\boldsymbol{w}^{(t)} - \boldsymbol{w}_i^{(t)}\|^2 + f_i(\boldsymbol{w}) - f_i(\boldsymbol{w}_i^{(t)}) \right] \right] \tag{9}$$

$$= -2\eta \mathbb{E} \left[ \sum_{i=1}^N \lambda_i^{(\lfloor \frac{t}{\tau} \rfloor)} f_i(\boldsymbol{w}^{(t)}) - \lambda_i^{(\lfloor \frac{t}{\tau} \rfloor)} f_i(\boldsymbol{w}) \right] + L\eta \mathbb{E} \left[ \delta^{(t)} \right]$$

$$= -2\eta \mathbb{E} \left[ F(\boldsymbol{w}^{(t)}, \boldsymbol{\lambda}^{(\lfloor \frac{t}{\tau} \rfloor)}) - F(\boldsymbol{w}, \boldsymbol{\lambda}^{(\lfloor \frac{t}{\tau} \rfloor)}) \right] + L\eta \mathbb{E} \left[ \delta^{(t)} \right],$$

where from (8) to (9) we use the smoothness and convexity properties.

We then turn to bounding $T_2$ as follows:

$$T_2 = \eta^2 \mathbb{E} \left\| \frac{1}{m} \sum_{i \in \mathcal{D}^{(\lfloor \frac{t}{\tau} \rfloor)}} \nabla f_i(\boldsymbol{w}_i^{(t)}) \right\|^2 \leq \eta^2 \frac{1}{m} \sum_{i \in \mathcal{D}^{(\lfloor \frac{t}{\tau} \rfloor)}} \mathbb{E} \left\| \nabla f_i(\boldsymbol{w}_i^{(t)}) \right\|^2 \leq \eta^2 G_w^2.$$

Plugging $T_1$ and $T_2$ back to (7) gives:

$$\mathbb{E}\|\boldsymbol{w}^{(t+1)} - \boldsymbol{w}\|^2 \leq \mathbb{E}\|\boldsymbol{w}^{(t)} - \boldsymbol{w}\|^2 - 2\eta \mathbb{E} \left[ F(\boldsymbol{w}^{(t)}, \boldsymbol{\lambda}^{(\lfloor \frac{t}{\tau} \rfloor)}) - F(\boldsymbol{w}, \boldsymbol{\lambda}^{(\lfloor \frac{t}{\tau} \rfloor)}) \right]$$
$$+ L\eta \mathbb{E} \left[ \delta^{(t)} \right] + \eta^2 \mathbb{E}\|\bar{\boldsymbol{u}}^{(t)} - \boldsymbol{u}^{(t)}\|^2 + \eta^2 G_w^2,$$

thus concluding the proof. $\qquad \square$

The following lemma bounds the deviation between local models and (virtual) global average model over sampled devices over $T$ iterations. We note that the following result is general and will be used in all variants.

**Lemma 4** (Bounded Squared Deviation). *For* DRFA*,* DRFA-Prox *and* DRFA-GA *algorithms, the expected average squared norm distance of local models* $\boldsymbol{w}_i^{(t)}, i \in \mathcal{D}^{(\lfloor \frac{t}{\tau} \rfloor)}$ *and* $\boldsymbol{w}^{(t)}$ *is bounded as follows:*

$$\frac{1}{T} \sum_{t=0}^T \mathbb{E} \left[ \delta^{(t)} \right] \leq 10\eta^2 \tau^2 \left( \sigma_w^2 + \frac{\sigma_w^2}{m} + \Gamma \right).$$

*where expectation is taken over sampling of devices at each iteration.*

*Proof.* Consider $s\tau \le t \le (s+1)\tau$. Recall that, we only perform the averaging based on a uniformly sampled subset of workers $\mathcal{D}^{(\lfloor \frac{t}{\tau} \rfloor)}$ of $[N]$. Following the updating rule we have:

$$
\mathbb{E}[\delta^{(t)}] = \mathbb{E}\left[\frac{1}{m}\sum_{i\in\mathcal{D}^{(\lfloor \frac{t}{\tau} \rfloor)}} \|\boldsymbol{w}_i^{(t)} - \boldsymbol{w}^{(t)}\|^2\right]
$$

$$
\le \mathbb{E}\left[\frac{1}{m}\sum_{i\in\mathcal{D}^{(\lfloor \frac{t}{\tau} \rfloor)}} \mathbb{E}\left\|\boldsymbol{w}^{(s\tau)} - \sum_{r=s\tau}^{t-1}\eta\nabla f_i(\boldsymbol{w}_i^{(r)};\xi_i^{(r)}) - \left(\boldsymbol{w}^{(s\tau)} - \frac{1}{m}\sum_{i'\in\mathcal{D}}\sum_{r=s\tau}^{t-1}\eta\nabla f_{i'}(\boldsymbol{w}_{i'}^{(r)};\xi_{i'}^{(r)})\right)\right\|^2\right]
$$

$$
= \mathbb{E}\left[\frac{1}{m}\sum_{i\in\mathcal{D}^{(\lfloor \frac{t}{\tau} \rfloor)}} \left\|\sum_{r=s\tau}^{t-1}\eta\nabla f_i(\boldsymbol{w}_i^{(r)};\xi_i^{(r)}) - \frac{1}{m}\sum_{i'\in\mathcal{D}^{(\lfloor \frac{t}{\tau} \rfloor)}}\sum_{r=s\tau}^{t-1}\eta\nabla f_{i'}(\boldsymbol{w}_{i'}^{(r)};\xi_{i'}^{(r)})\right\|^2\right]
$$

$$
\le \mathbb{E}\left[\frac{1}{m}\sum_{i\in\mathcal{D}^{(\lfloor \frac{t}{\tau} \rfloor)}} \eta^2\tau\sum_{r=s\tau}^{(s+1)\tau} \left\|\nabla f_i(\boldsymbol{w}_i^{(r)};\xi_i^{(r)}) - \frac{1}{m}\sum_{i'\in\mathcal{D}^{(\lfloor \frac{t}{\tau} \rfloor)}}\nabla f_{i'}(\boldsymbol{w}_{i'}^{(r)};\xi_{i'}^{(r)})\right\|^2\right]
$$

$$
= \eta^2\tau\mathbb{E}\left[\frac{1}{m}\sum_{i\in\mathcal{D}^{(\lfloor \frac{t}{\tau} \rfloor)}}\sum_{r=s\tau}^{(s+1)\tau} \left\|\nabla f_i(\boldsymbol{w}_i^{(r)};\xi_i^{(r)}) - \nabla f_i(\boldsymbol{w}_i^{(r)}) + \nabla f_i(\boldsymbol{w}_i^{(r)}) - \nabla f_i(\boldsymbol{w}^{(r)})\right.\right.
$$

$$
+ \nabla f_i(\boldsymbol{w}^{(r)}) - \frac{1}{m}\sum_{i'\in\mathcal{D}^{(\lfloor \frac{t}{\tau} \rfloor)}}\nabla f_{i'}(\boldsymbol{w}^{(r)}) + \frac{1}{m}\sum_{i'\in\mathcal{D}^{(\lfloor \frac{t}{\tau} \rfloor)}}\nabla f_{i'}(\boldsymbol{w}^{(r)})
$$

$$
\left.\left. - \frac{1}{m}\sum_{i'\in\mathcal{D}^{(\lfloor \frac{t}{\tau} \rfloor)}}\nabla f_{i'}(\boldsymbol{w}_{i'}^{(r)}) + \frac{1}{m}\sum_{i'\in\mathcal{D}^{(\lfloor \frac{t}{\tau} \rfloor)}}\nabla f_{i'}(\boldsymbol{w}_{i'}^{(r)}) - \frac{1}{m}\sum_{i'\in\mathcal{D}^{(\lfloor \frac{t}{\tau} \rfloor)}}\nabla f_{i'}(\boldsymbol{w}_{i'}^{(r)};\xi_{i'}^{(r)})\right\|^2\right] \tag{10}
$$

Applying Jensen's inequality to split the norm yields:

$$
\mathbb{E}[\delta^{(t)}] \le 5\eta^2\tau\sum_{r=s\tau}^{(s+1)\tau}\left(\sigma_w^2 + L^2\mathbb{E}\left[\frac{1}{m}\sum_{i\in\mathcal{D}^{(\lfloor \frac{t}{\tau} \rfloor)}}\left\|\boldsymbol{w}_i^{(r)} - \boldsymbol{w}^{(r)}\right\|^2\right] + L^2\mathbb{E}\left[\frac{1}{m}\sum_{i'\in\mathcal{D}^{(\lfloor \frac{t}{\tau} \rfloor)}}\left\|\boldsymbol{w}_{i'}^{(r)} - \boldsymbol{w}^{(r)}\right\|^2\right]\right.
$$

$$
\left. + \mathbb{E}\left[\frac{1}{m}\sum_{i'\in\mathcal{D}^{(\lfloor \frac{t}{\tau} \rfloor)}}\left\|\nabla f_i(\boldsymbol{w}^{(r)}) - \nabla f_{i'}(\boldsymbol{w}^{(r)})\right\|^2\right] + \frac{\sigma_w^2}{m}\right) \tag{11}
$$

$$
\le 5\eta^2\tau\sum_{r=s\tau}^{(s+1)\tau}\left(\sigma_w^2 + 2L^2\mathbb{E}[\delta^{(r)}] + \Gamma + \frac{\sigma_w^2}{m}\right), \tag{12}
$$

where from (10) to (11) we use the Jensen's inequality.

Now we sum (12) over $t = s\tau$ to $(s+1)\tau$ to get:

$$
\sum_{t=s\tau}^{(s+1)\tau}\mathbb{E}[\delta^{(t)}] \le 5\eta^2\tau\sum_{t=s\tau}^{(s+1)\tau}\sum_{r=s\tau}^{(s+1)\tau}\left(\sigma_w^2 + 2L^2\mathbb{E}[\delta^{(r)}] + \Gamma + \frac{\sigma_w^2}{m}\right)
$$

$$
= 5\eta^2\tau^2\sum_{r=s\tau}^{(s+1)\tau}\left(\sigma_w^2 + 2\mathbb{E}[\delta^{(r)}] + \Gamma + \frac{\sigma_w^2}{m}\right).
$$

Re-arranging the terms and using the fact $1 - 10\eta^2\tau^2L^2 \ge \frac{1}{2}$ yields:

$$
\sum_{t=s\tau}^{(s+1)\tau}\mathbb{E}[\delta^{(t)}] \le 10\eta^2\tau^2\sum_{r=s\tau}^{(s+1)\tau}\left(\sigma_w^2 + \Gamma + \frac{\sigma_w^2}{m}\right).
$$

Summing over communication steps $s = 0$ to $S - 1$, and dividing both sides by $T = S\tau$ yields:

$$\frac{1}{T}\sum_{t=0}^{T}\mathbb{E}[\delta^{(t)}] \leq 10\eta^2\tau^2\left(\sigma_w^2 + \frac{\sigma_w^2}{m} + \Gamma\right),$$

as desired. $\qquad\square$

**Lemma 5** (Bounded Norm Deviation). *For DRFA, DRFA-Prox and DRFA-GA, $\forall i \in \mathcal{D}^{(\lfloor\frac{t}{\tau}\rfloor)}$, the norm distance between $\boldsymbol{w}^{(t)}$ and $\boldsymbol{w}_i^{(t)}$ is bounded as follows:*

$$\frac{1}{T}\sum_{t=0}^{T}\mathbb{E}\left[\frac{1}{m}\sum_{i\in\mathcal{D}^{(\lfloor\frac{t}{\tau}\rfloor)}}\left\|\boldsymbol{w}_i^{(t)} - \boldsymbol{w}^{(t)}\right\|\right] \leq 2\eta\tau\left(\sigma_w + \frac{\sigma_w}{m} + \sqrt{\Gamma}\right).$$

*Proof.* Similar to what we did in Lemma 4, we assume $s\tau \leq t \leq (s+1)\tau$. Again, we only apply the averaging based on a uniformly sampled subset of workers $\mathcal{D}^{(\lfloor\frac{t}{\tau}\rfloor)}$ of $[N]$. From the updating rule we have:

$$\mathbb{E}\left[\frac{1}{m}\sum_{i\in\mathcal{D}^{(\lfloor\frac{t}{\tau}\rfloor)}}\|\boldsymbol{w}_i^{(t)} - \boldsymbol{w}^{(t)}\|\right]$$

$$= \mathbb{E}\left[\frac{1}{m}\sum_{i\in\mathcal{D}^{(\lfloor\frac{t}{\tau}\rfloor)}}\left\|\boldsymbol{w}^{(s\tau)} - \sum_{r=s\tau}^{t-1}\eta\nabla f_i(\boldsymbol{w}_i^{(r)};\xi_i^{(r)}) - \left(\boldsymbol{w}^{(s\tau)} - \frac{1}{m}\sum_{i'\in\mathcal{D}}\sum_{r=s\tau}^{t-1}\eta\nabla f_{i'}(\boldsymbol{w}_{i'}^{(r)};\xi_{i'}^{(r)})\right)\right\|\right]$$

$$= \mathbb{E}\left[\frac{1}{m}\sum_{i\in\mathcal{D}^{(\lfloor\frac{t}{\tau}\rfloor)}}\mathbb{E}\left\|\sum_{r=s\tau}^{t-1}\eta\nabla f_i(\boldsymbol{w}_i^{(r)};\xi_i^{(r)}) - \frac{1}{m}\sum_{i'\in\mathcal{D}^{(\lfloor\frac{t}{\tau}\rfloor)}}\sum_{r=s\tau}^{t-1}\eta\nabla f_{i'}(\boldsymbol{w}_{i'}^{(r)};\xi_{i'}^{(r)})\right\|\right]$$

$$\leq \mathbb{E}\left[\frac{1}{m}\sum_{i\in\mathcal{D}^{(\lfloor\frac{t}{\tau}\rfloor)}}\eta\sum_{r=s\tau}^{(s+1)\tau}\mathbb{E}\left\|\nabla f_i(\boldsymbol{w}_i^{(r)};\xi_i^{(r)}) - \frac{1}{m}\sum_{i'\in\mathcal{D}^{(\lfloor\frac{t}{\tau}\rfloor)}}\nabla f_{i'}(\boldsymbol{w}_{i'}^{(r)};\xi_{i'}^{(r)})\right\|\right]$$

$$= \eta\mathbb{E}\left[\frac{1}{m}\sum_{i\in\mathcal{D}^{(\lfloor\frac{t}{\tau}\rfloor)}}\sum_{r=s\tau}^{(s+1)\tau}\left\|\nabla f_i(\boldsymbol{w}_i^{(r)};\xi_i^{(r)}) - \nabla f_i(\boldsymbol{w}_i^{(r)}) + \nabla f_i(\boldsymbol{w}_i^{(r)}) - \nabla f_i(\boldsymbol{w}^{(r)}) + \nabla f_i(\boldsymbol{w}^{(r)})\right.\right.$$

$$- \frac{1}{m}\sum_{i'\in\mathcal{D}^{(\lfloor\frac{t}{\tau}\rfloor)}}\nabla f_{i'}(\boldsymbol{w}^{(r)}) + \frac{1}{m}\sum_{i'\in\mathcal{D}^{(\lfloor\frac{t}{\tau}\rfloor)}}\nabla f_{i'}(\boldsymbol{w}^{(r)}) - \frac{1}{m}\sum_{i'\in\mathcal{D}^{(\lfloor\frac{t}{\tau}\rfloor)}}\nabla f_{i'}(\boldsymbol{w}_{i'}^{(r)})$$

$$\left.\left.+ \frac{1}{m}\sum_{i'\in\mathcal{D}^{(\lfloor\frac{t}{\tau}\rfloor)}}\nabla f_{i'}(\boldsymbol{w}_{i'}^{(r)}) - \frac{1}{m}\sum_{i'\in\mathcal{D}^{(\lfloor\frac{t}{\tau}\rfloor)}}\nabla f_{i'}(\boldsymbol{w}_{i'}^{(r)};\xi_{i'}^{(r)})\right\|\right]$$

Applying the triangular inequality to split the norm yields:

$$\mathbb{E}\left[\frac{1}{m}\sum_{i\in\mathcal{D}^{(\lfloor\frac{t}{\tau}\rfloor)}}\|\boldsymbol{w}_i^{(t)} - \boldsymbol{w}^{(t)}\|\right]$$

$$\leq \eta\mathbb{E}\left[\frac{1}{m}\sum_{i\in\mathcal{D}^{(\lfloor\frac{t}{\tau}\rfloor)}}\sum_{r=s\tau}^{(s+1)\tau}\left(\sigma_w + L\left\|\boldsymbol{w}_i^{(r)} - \boldsymbol{w}^{(r)}\right\|\right.\right.$$

$$\left.\left.+ \frac{1}{m}\sum_{i'\in\mathcal{D}^{(\lfloor\frac{t}{\tau}\rfloor)}}L\left\|\boldsymbol{w}_{i'}^{(r)} - \boldsymbol{w}^{(r)}\right\| + \frac{1}{m}\sum_{i'\in\mathcal{D}^{(\lfloor\frac{t}{\tau}\rfloor)}}\left\|\nabla f_i(\boldsymbol{w}^{(r)}) - \nabla f_{i'}(\boldsymbol{w}^{(r)})\right\| + \frac{\sigma_w}{m}\right)\right]$$

$$= \eta\sum_{r=s\tau}^{(s+1)\tau}\left(\sigma_w + 2L\mathbb{E}\left[\frac{1}{m}\sum_{i'\in\mathcal{D}^{(r)}}\mathbb{E}\|\boldsymbol{w}_{i'}^{(r)} - \boldsymbol{w}^{(r)}\|\right] + \sqrt{\Gamma} + \frac{\sigma_w}{m}\right). \tag{13}$$

Now summing (13) over $t = s\tau$ to $(s+1)\tau$ gives:

$$\sum_{t=s\tau}^{(s+1)\tau} \mathbb{E}\left[\frac{1}{m}\sum_{i\in\mathcal{D}^{(\lfloor\frac{t}{\tau}\rfloor)}}\|\boldsymbol{w}_i^{(t)} - \boldsymbol{w}^{(t)}\|\right]$$

$$\leq \eta \sum_{t=s\tau}^{(s+1)\tau}\sum_{r=s\tau}^{(s+1)\tau}\left(\sigma_w + 2L\mathbb{E}\left[\frac{1}{m}\sum_{i'\in\mathcal{D}^{(r)}}\|\boldsymbol{w}_{i'}^{(r)} - \boldsymbol{w}^{(r)}\|\right] + \sqrt{\Gamma} + \frac{\sigma_w}{m}\right)$$

$$= \eta\tau\sum_{r=s\tau}^{(s+1)\tau}\left(\sigma_w + 2L\mathbb{E}\left[\frac{1}{m}\sum_{i'\in\mathcal{D}^{(r)}}\|\boldsymbol{w}_{i'}^{(r)} - \boldsymbol{w}^{(r)}\|\right] + \sqrt{\Gamma} + \frac{\sigma_w}{m}\right).$$

Re-arranging the terms and using the fact $1 - 2\eta\tau L \geq \frac{1}{2}$ yields:

$$\sum_{t=s\tau}^{(s+1)\tau} \mathbb{E}\left[\frac{1}{m}\sum_{i\in\mathcal{D}^{(\lfloor\frac{t}{\tau}\rfloor)}}\|\boldsymbol{w}_i^{(t)} - \boldsymbol{w}^{(t)}\|\right] \leq 2\eta\tau\sum_{r=s\tau}^{(s+1)\tau}\left(\sigma_w + \sqrt{\Gamma} + \frac{\sigma_w}{m}\right).$$

Summing over $s = 0$ to $S - 1$, and dividing both sides by $T = S\tau$ yields:

$$\frac{1}{T}\sum_{t=0}^{T}\mathbb{E}\left[\frac{1}{m}\sum_{i\in\mathcal{D}^{(\lfloor\frac{t}{\tau}\rfloor)}}\|\boldsymbol{w}_i^{(t)} - \boldsymbol{w}^{(t)}\|\right] \leq 2\eta\tau\left(\sigma_w + \frac{\sigma_w}{m} + \sqrt{\Gamma}\right),$$

which concludes the proof. $\qquad\square$

**Lemma 6** (One Iteration Dual Analysis). *For* DRFA, *under the assumption of Theorem 1, the following holds true for any* $\boldsymbol{\lambda} \in \Lambda$:

$$\mathbb{E}\|\boldsymbol{\lambda}^{(s+1)} - \boldsymbol{\lambda}\|^2 \leq \mathbb{E}\|\boldsymbol{\lambda}^{(s)} - \boldsymbol{\lambda}\|^2$$
$$- \sum_{t=s\tau+1}^{(s+1)\tau}\mathbb{E}[2\gamma(F(\boldsymbol{w}^{(t)}, \boldsymbol{\lambda}^{(\lfloor\frac{t}{\tau}\rfloor)}) - F(\boldsymbol{w}^{(t)}, \boldsymbol{\lambda}))] + \mathbb{E}\|\bar{\Delta}_t\|^2 + \mathbb{E}\|\Delta_t - \bar{\Delta}_t\|^2.$$

*Proof.* According to the updating rule for $\boldsymbol{\lambda}$ and the fact $F$ is linear in $\boldsymbol{\lambda}$ we have:

$$\mathbb{E}\left\|\boldsymbol{\lambda}^{(s+1)} - \boldsymbol{\lambda}\right\|^2 = \mathbb{E}\left\|\prod_\Lambda(\boldsymbol{\lambda}^{(s)} + \Delta_s) - \boldsymbol{\lambda}\right\|^2$$

$$\leq \mathbb{E}\left\|\boldsymbol{\lambda}^{(s)} - \boldsymbol{\lambda} + \Delta_s\right\|^2$$

$$= \mathbb{E}\left\|\boldsymbol{\lambda}^{(s)} - \boldsymbol{\lambda} + \bar{\Delta}_s\right\|^2 + \mathbb{E}\left\|\Delta_s - \bar{\Delta}_s\right\|^2$$

$$= \mathbb{E}\|\boldsymbol{\lambda}^{(s)} - \boldsymbol{\lambda}\|^2 + \mathbb{E}\left[2\left\langle\bar{\Delta}_s, \boldsymbol{\lambda}^{(s)} - \boldsymbol{\lambda}\right\rangle\right] + \mathbb{E}\|\bar{\Delta}_s\|^2 + \mathbb{E}\|\Delta_s - \bar{\Delta}_s\|^2$$

$$= \mathbb{E}\|\boldsymbol{\lambda}^{(s)} - \boldsymbol{\lambda}\|^2$$

$$+ 2\gamma\sum_{t=s\tau+1}^{(s+1)\tau}\mathbb{E}\left[\left\langle\nabla_{\boldsymbol{\lambda}}F(\boldsymbol{w}^{(t)}, \boldsymbol{\lambda}^{(s)}), \boldsymbol{\lambda}^{(s)} - \boldsymbol{\lambda}\right\rangle\right] + \mathbb{E}\|\bar{\Delta}_s\|^2 + \mathbb{E}\|\Delta_s - \bar{\Delta}_s\|^2$$

$$= \|\boldsymbol{\lambda}^{(s)} - \boldsymbol{\lambda}\|^2$$

$$- 2\gamma\sum_{t=s\tau+1}^{(s+1)\tau}\mathbb{E}\left[F(\boldsymbol{w}^{(t)}, \boldsymbol{\lambda}) - F(\boldsymbol{w}^{(t)}, \boldsymbol{\lambda}^{(s)}))\right] + \mathbb{E}\|\bar{\Delta}_s\|^2 + \mathbb{E}\|\Delta_s - \bar{\Delta}_s\|^2,$$

as desired.

$\qquad\square$

## C.4 Proof for Theorem 1

*Proof.* Equipped with above results, we are now turn to proving the Theorem 1. We start by noting that $\forall \boldsymbol{w} \in \mathcal{W}$, $\forall \boldsymbol{\lambda} \in \Lambda$, according the convexity of global objective w.r.t. $\boldsymbol{w}$ and its linearity in terms of $\boldsymbol{\lambda}$ we have:

$$\mathbb{E}[F(\hat{\boldsymbol{w}}, \boldsymbol{\lambda}) - \mathbb{E}[F(\boldsymbol{w}, \hat{\boldsymbol{\lambda}})]$$

$$\leq \frac{1}{T} \sum_{t=1}^{T} \left\{ \mathbb{E}\left[F(\boldsymbol{w}^{(t)}, \boldsymbol{\lambda})\right] - \mathbb{E}\left[F(\boldsymbol{w}, \boldsymbol{\lambda}^{(\lfloor \frac{t}{\tau} \rfloor)})\right] \right\}$$

$$\leq \frac{1}{T} \sum_{t=1}^{T} \left\{ \mathbb{E}\left[F(\boldsymbol{w}^{(t)}, \boldsymbol{\lambda})\right] - \mathbb{E}\left[F(\boldsymbol{w}^{(t)}, \boldsymbol{\lambda}^{(\lfloor \frac{t}{\tau} \rfloor)})\right] + \mathbb{E}\left[F(\boldsymbol{w}^{(t)}, \boldsymbol{\lambda}^{(\lfloor \frac{t}{\tau} \rfloor)})\right] - \mathbb{E}\left[F(\boldsymbol{w}, \boldsymbol{\lambda}^{(\lfloor \frac{t}{\tau} \rfloor)})\right] \right\}$$

$$\leq \frac{1}{T} \sum_{s=0}^{S-1} \sum_{t=s\tau+1}^{(s+1)\tau} \mathbb{E}\{F(\boldsymbol{w}^{(t)}, \boldsymbol{\lambda}) - F(\boldsymbol{w}^{(t)}, \boldsymbol{\lambda}^{(s)})\} \tag{14}$$

$$+ \frac{1}{T} \sum_{t=1}^{T} \mathbb{E}\{F(\boldsymbol{w}^{(t)}, \boldsymbol{\lambda}^{(\lfloor \frac{t}{\tau} \rfloor)}) - F(\boldsymbol{w}, \boldsymbol{\lambda}^{(\lfloor \frac{t}{\tau} \rfloor)})\}, \tag{15}$$

To bound the term in (14), pluggin Lemma 2 into Lemma 6, we have:

$$\frac{1}{T} \sum_{s=0}^{S-1} \sum_{t=s\tau+1}^{(s+1)\tau} \mathbb{E}(F(\boldsymbol{w}^{(t)}, \boldsymbol{\lambda}) - F(\boldsymbol{w}^{(t)}, \boldsymbol{\lambda}^{(\lfloor \frac{t}{\tau} \rfloor)})) \leq \frac{1}{2\gamma T} \|\boldsymbol{\lambda}^{(0)} - \boldsymbol{\lambda}\|^2 + \frac{\gamma\tau}{2} G_\lambda^2 + \frac{\gamma\tau\sigma_\lambda^2}{2m}$$

$$\leq \frac{D_\Lambda^2}{2\gamma T} + \frac{\gamma\tau G_{\boldsymbol{\lambda}}^2}{2} + \frac{\gamma\tau\sigma_\lambda^2}{2m}.$$

To bound the term in (15), we plug Lemma 1 into Lemma 3 and apply the telescoping sum from $t = 1$ to $T$ to get:

$$\frac{1}{T} \sum_{t=1}^{T} \mathbb{E}(F(\boldsymbol{w}^{(t)}, \boldsymbol{\lambda}^{(\lfloor \frac{t}{\tau} \rfloor)}) - F(\boldsymbol{w}, \boldsymbol{\lambda}^{(\lfloor \frac{t}{\tau} \rfloor)}))$$

$$\leq \frac{1}{2T\eta} \mathbb{E}\| \boldsymbol{w}^{(0)} - \boldsymbol{w}\|^2 + 5L\eta^2\tau^2 \left(\sigma_w^2 + \frac{\sigma_w^2}{m} + \Gamma\right) + \frac{\eta G_w^2}{2} + \frac{\eta\sigma_w^2}{2m}$$

$$\leq \frac{D_\mathcal{W}^2}{2T\eta} + 5L\eta^2\tau^2 \left(\sigma_w^2 + \frac{\sigma_w^2}{m} + \Gamma\right) + \frac{\eta G_w^2}{2} + \frac{\eta\sigma_w^2}{2m}.$$

Putting pieces together, and taking max over dual $\boldsymbol{\lambda}$, min over primal $\boldsymbol{w}$ yields:

$$\min_{\boldsymbol{w} \in \mathcal{W}} \max_{\boldsymbol{\lambda} \in \Lambda} \mathbb{E}[F(\hat{\boldsymbol{w}}, \boldsymbol{\lambda}) - \mathbb{E}[F(\boldsymbol{w}, \hat{\boldsymbol{\lambda}})]$$

$$\leq \frac{D_\mathcal{W}^2}{2T\eta} + 5L\eta^2\tau^2 \left(\sigma_w^2 + \frac{\sigma_w^2}{m} + \Gamma\right) + \frac{\eta G_w^2}{2} + \frac{\eta\sigma_w^2}{2m} + \frac{D_\Lambda^2}{2\gamma T} + \frac{\gamma\tau G_{\boldsymbol{\lambda}}^2}{2} + \frac{\gamma\tau\sigma_\lambda^2}{2m}.$$

Plugging in $\tau = \frac{T^{1/4}}{\sqrt{m}}$, $\eta = \frac{1}{4L\sqrt{T}}$, and $\gamma = \frac{1}{T^{5/8}}$, we conclude the proof by getting:

$$\max_{\boldsymbol{\lambda} \in \Lambda} \mathbb{E}[F(\hat{\boldsymbol{w}}, \boldsymbol{\lambda})] - \min_{\boldsymbol{w} \in \mathcal{W}} \mathbb{E}[F(\boldsymbol{w}, \hat{\boldsymbol{\lambda}})] \leq O\Big(\frac{D_\mathcal{W}^2 + G_w^2}{\sqrt{T}} + \frac{D_\Lambda^2}{T^{3/8}}.$$

$$+ \frac{G_\lambda^2}{m^{1/2}T^{3/8}} + \frac{\sigma_\lambda^2}{m^{3/2}T^{3/8}} + \frac{\sigma_w^2 + \Gamma}{m\sqrt{T}}\Big),$$

as desired. □

# D  Proof of Convergence of **DRFA** for Nonconvex Losses (Theorem 2)

This section is devoted to the proof of Theorem 2).

## D.1 Overview of Proofs

Inspired by the techniques in [29] for analyzing the behavior of stochastic gradient descent ascent (SGDA) algorithm on nonconvex-concave objectives, we consider the Moreau Envelope of $\Phi$:

$$\Phi_p(\boldsymbol{x}) := \min_{\boldsymbol{w} \in \mathcal{W}} \left\{ \Phi(\boldsymbol{w}) + \frac{1}{2p} \|\boldsymbol{w} - \boldsymbol{x}\| \right\}.$$

We first examine the one iteration dynamic of DRFA:

$$\mathbb{E}[\Phi_{1/2L}(\boldsymbol{w}^{(t)})] \leq \mathbb{E}[\Phi_{1/2L}(\boldsymbol{w}^{(t-1)})] + 2\eta D_{\mathcal{W}} L^2 \mathbb{E}\left[ \frac{1}{m} \sum_{i \in \mathcal{D}^{(\lfloor \frac{t-1}{\tau} \rfloor)}} \left\| \boldsymbol{w}_i^{(t-1)} - \boldsymbol{w}^{(t-1)} \right\| \right]$$

$$2\eta L \left( \mathbb{E}[\Phi(\boldsymbol{w}^{(t-1)})] - \mathbb{E}[F(\boldsymbol{w}^{(t-1)}, \boldsymbol{\lambda}^{\lfloor \frac{t-1}{\tau} \rfloor})] \right) - \frac{\eta}{4} \mathbb{E}\left[ \left\| \nabla \Phi_{1/2L}(\boldsymbol{w}^{(t-1)}) \right\|^2 \right].$$

We already know how to bound $\mathbb{E}\left[ \frac{1}{m} \sum_{i \in \mathcal{D}^{(\lfloor \frac{t-1}{\tau} \rfloor)}} \left\| \boldsymbol{w}_i^{(t-1)} - \boldsymbol{w}^{(t-1)} \right\| \right]$ in Lemma 5. Then the key is to bound $\mathbb{E}[\Phi(\boldsymbol{w}^{(t-1)})] - \mathbb{E}[F(\boldsymbol{w}^{(t-1)}, \boldsymbol{\lambda}^{(\lfloor \frac{t-1}{\tau} \rfloor)})]$. Indeed this term characterizes how far the current dual variable drifts from the optimal dual variable $\boldsymbol{\lambda}^*(\boldsymbol{w}^{(t-1)})$. Then by examining the dynamic of dual variable we have $\forall \boldsymbol{\lambda} \in \Lambda$:

$$\sum_{t=(s-1)\tau+1}^{s\tau} \left( \mathbb{E}\left[ \Phi(\boldsymbol{w}^{(t)}) \right] - \mathbb{E}\left[ F(\boldsymbol{w}^{(t)}, \boldsymbol{\lambda}^{(s-1)}) \right] \right)$$

$$\leq \sum_{t=(s-1)\tau+1}^{s\tau} \left( \mathbb{E}\left[ F(\boldsymbol{w}^{(t)}, \boldsymbol{\lambda}^*(\boldsymbol{w}^t)) \right] - \mathbb{E}\left[ F(\boldsymbol{w}^{(t)}, \boldsymbol{\lambda}) \right] \right)$$

$$+ \gamma \tau^2 \frac{\sigma_\lambda^2}{m} + \gamma \tau^2 G_\lambda^2 + \frac{1}{2\gamma} \left( \mathbb{E}\left[ \left\| \boldsymbol{\lambda} - \boldsymbol{\lambda}^{(s-1)} \right\|^2 \right] - \mathbb{E}\left[ \left\| \boldsymbol{\lambda} - \boldsymbol{\lambda}^{(s)} \right\|^2 \right] \right).$$

The above inequality makes it possible to replace $\boldsymbol{\lambda}$ with $\boldsymbol{\lambda}^*$, and doing the telescoping sum so that the last term cancels up. However, in the minimax problem, the optimal dual variable changes every time when we update primal variable. Thus, we divide $S$ global stages into $\sqrt{S}$ groups, and applying the telescoping sum within one group, by setting $\boldsymbol{\lambda} = \boldsymbol{\lambda}^*(\boldsymbol{w}^{c\sqrt{S}\tau})$ at $c$th stage.

## D.2 Proof of Useful Lemmas

Before presenting the proof of Theorem 2, let us introduce the following useful lemmas.

**Lemma 7** (One iteration analysis). *For DRFA, under the assumptions of Theorem 2, the following statement holds:*

$$\mathbb{E}[\Phi_{1/2L}(\boldsymbol{w}^{(t)})] \leq \mathbb{E}[\Phi_{1/2L}(\boldsymbol{w}^{(t-1)})] + 2\eta D_{\mathcal{W}} L^2 \mathbb{E}\left[ \frac{1}{m} \sum_{i \in \mathcal{D}^{(\lfloor \frac{t-1}{\tau} \rfloor)}} \left\| \boldsymbol{w}_i^{(t-1)} - \boldsymbol{w}^{(t-1)} \right\| \right]$$

$$+ 2\eta L \left( \mathbb{E}[\Phi(\boldsymbol{w}^{(t-1)})] - \mathbb{E}[F(\boldsymbol{w}^{(t-1)}, \boldsymbol{\lambda}^{(\lfloor \frac{t-1}{\tau} \rfloor)})] \right) - \frac{\eta}{4} \mathbb{E}\left[ \left\| \nabla \Phi_{1/2L}(\boldsymbol{w}^{(t-1)}) \right\|^2 \right].$$

*Proof.* Define $\tilde{\boldsymbol{w}}^{(t)} = \min_{\boldsymbol{w} \in \mathcal{W}} \Phi(\boldsymbol{w}) + L \|\boldsymbol{w} - \boldsymbol{w}^{(t)}\|^2$, the by the definition of $\Phi_{1/2L}$ we have:

$$\Phi_{1/2L}(\boldsymbol{w}^{(t)}) \leq \Phi(\tilde{\boldsymbol{w}}^{(t-1)}) + L \|\tilde{\boldsymbol{w}}^{(t-1)} - \boldsymbol{w}^{(t)}\|^2. \tag{16}$$

Meanwhile according to updating rule we have:

$$\mathbb{E}\left[\left\|\tilde{\boldsymbol{w}}^{(t-1)} - \boldsymbol{w}^{(t)}\right\|^2\right]$$

$$= \mathbb{E}\left[\left\|\tilde{\boldsymbol{w}}^{(t-1)} - \prod_{\mathcal{W}}\left(\boldsymbol{w}^{(t-1)} - \frac{1}{m}\sum_{i\in\mathcal{D}^{(\lfloor\frac{t-1}{\tau}\rfloor)}}\nabla_x f_i(\boldsymbol{w}_i^{(t-1)};\xi_i^{(t-1)})\right)\right\|^2\right]$$

$$\leq \mathbb{E}\left[\left\|\tilde{\boldsymbol{w}}^{(t-1)} - \boldsymbol{w}^{(t-1)}\right\|^2\right] + \mathbb{E}\left[\left\|\frac{1}{m}\sum_{i\in\mathcal{D}^{(\lfloor\frac{t-1}{\tau}\rfloor)}}\nabla_x f_i(\boldsymbol{w}_i^{(t-1)};\xi_i^{(t-1)})\right\|^2\right]$$

$$+ 2\eta\mathbb{E}\left[\left\langle\tilde{\boldsymbol{w}}^{(t-1)} - \boldsymbol{w}^{(t-1)}, \frac{1}{m}\sum_{i\in\mathcal{D}^{(\lfloor\frac{t-1}{\tau}\rfloor)}}\nabla_x f_i(\boldsymbol{w}_i^{(t-1)})\right\rangle\right].$$

Applying Cauchy inequality to the last inner product term yields:

$$\mathbb{E}\left[\left\|\tilde{\boldsymbol{w}}^{(t-1)} - \boldsymbol{w}^{(t)}\right\|^2\right]$$

$$\leq \mathbb{E}\left[\left\|\tilde{\boldsymbol{w}}^{(t-1)} - \boldsymbol{w}^{(t-1)}\right\|^2\right] + \eta^2(G_w^2 + \sigma_w^2) + 2\eta\left\langle\tilde{\boldsymbol{w}}^{(t-1)} - \boldsymbol{w}^{(t-1)}, \frac{1}{m}\sum_{i\in\mathcal{D}^{(\lfloor\frac{t-1}{\tau}\rfloor)}}\nabla_x f_i(\boldsymbol{w}^{(t-1)})\right\rangle$$

$$+ \eta\mathbb{E}\left[\left\|\tilde{\boldsymbol{w}}^{(t-1)} - \boldsymbol{w}^{(t-1)}\right\|\right]\mathbb{E}\left[\frac{1}{m}\sum_{i\in\mathcal{D}^{(\lfloor\frac{t-1}{\tau}\rfloor)}}\left\|\nabla_x f_i(\boldsymbol{w}_i^{(t-1)}) - \nabla_x f_i(\boldsymbol{w}^{(t-1)})\right\|\right]$$

$$\leq \mathbb{E}\left[\left\|\tilde{\boldsymbol{w}}^{(t-1)} - \boldsymbol{w}^{(t-1)}\right\|^2\right] + \eta^2(G_w^2 + \sigma_w^2) + \eta D_{\mathcal{W}}L\mathbb{E}\left[\frac{1}{m}\sum_{i\in\mathcal{D}^{(\lfloor\frac{t-1}{\tau}\rfloor)}}\left\|\boldsymbol{w}_i^{(t-1)} - \boldsymbol{w}^{(t-1)}\right\|\right]$$

$$+ 2\eta\mathbb{E}\left[\left\langle\tilde{\boldsymbol{w}}^{(t-1)} - \boldsymbol{w}^{(t-1)}, \nabla_x F(\boldsymbol{w}^{(t-1)}, \boldsymbol{\lambda}^{\lfloor\frac{t-1}{\tau}\rfloor})\right\rangle\right]. \tag{17}$$

According to smoothness of $F$ we obtain:

$$\mathbb{E}\left[\left\langle\tilde{\boldsymbol{w}}^{(t-1)} - \boldsymbol{w}^{(t-1)}, \nabla_x F(\boldsymbol{w}^{(t-1)}, \boldsymbol{\lambda}^{\lfloor\frac{t-1}{\tau}\rfloor})\right\rangle\right]$$

$$\leq \mathbb{E}\left[F(\tilde{\boldsymbol{w}}^{(t-1)}, \boldsymbol{\lambda}^{\lfloor\frac{t-1}{\tau}\rfloor})\right] - \mathbb{E}\left[F(\boldsymbol{w}^{(t-1)}, \boldsymbol{\lambda}^{\lfloor\frac{t-1}{\tau}\rfloor})\right] + \frac{L}{2}\mathbb{E}\left[\left\|\tilde{\boldsymbol{w}}^{(t-1)} - \boldsymbol{w}^{(t-1)}\right\|^2\right]$$

$$\leq \mathbb{E}\left[\Phi(\tilde{\boldsymbol{w}}^{(t-1)})\right] - \mathbb{E}\left[F(\boldsymbol{w}^{(t-1)}, \boldsymbol{\lambda}^{\lfloor\frac{t-1}{\tau}\rfloor})\right] + \frac{L}{2}\mathbb{E}\left[\left\|\tilde{\boldsymbol{w}}^{(t-1)} - \boldsymbol{w}^{(t-1)}\right\|^2\right]$$

$$\leq \underbrace{\mathbb{E}\left[\Phi(\tilde{\boldsymbol{w}}^{(t-1)})\right] + L\mathbb{E}\left[\left\|\tilde{\boldsymbol{w}}^{(t-1)} - \boldsymbol{w}^{(t-1)}\right\|^2\right]}_{\leq\mathbb{E}[\Phi(\boldsymbol{w}^{(t-1)})]+L\mathbb{E}\left[\|\boldsymbol{w}^{(t-1)}-\boldsymbol{w}^{(t-1)}\|^2\right]} - \mathbb{E}\left[F(\boldsymbol{w}^{(t-1)}, \boldsymbol{\lambda}^{\lfloor\frac{t-1}{\tau}\rfloor})\right] - \frac{L}{2}\mathbb{E}\left[\left\|\tilde{\boldsymbol{w}}^{(t-1)} - \boldsymbol{w}^{(t-1)}\right\|^2\right]$$

$$\leq \mathbb{E}\left[\Phi(\boldsymbol{w}^{(t-1)})\right] - \mathbb{E}\left[F(\boldsymbol{w}^{(t-1)}, \boldsymbol{\lambda}^{\lfloor\frac{t-1}{\tau}\rfloor})\right] - \frac{L}{2}\mathbb{E}\left[\left\|\tilde{\boldsymbol{w}}^{(t-1)} - \boldsymbol{w}^{(t-1)}\right\|^2\right]. \tag{18}$$

Plugging (17) and (18) into (16) yields:

$$\Phi_{1/2L}(\boldsymbol{w}^{(t)}) \le \Phi(\tilde{\boldsymbol{w}}^{(t-1)}) + L\mathbb{E}\left[\left\|\tilde{\boldsymbol{w}}^{(t-1)} - \boldsymbol{w}^{(t-1)}\right\|^2\right]$$

$$+ L\eta^2(G_w^2 + \sigma_w^2) + \eta D_{\mathcal{W}}L^2\mathbb{E}\left[\frac{1}{m}\sum_{i\in\mathcal{D}^{(\lfloor\frac{t-1}{\tau}\rfloor)}}\left\|\boldsymbol{w}_i^{(t-1)} - \boldsymbol{w}^{(t-1)}\right\|\right]$$

$$+ 2L\eta\left(\mathbb{E}\left[\Phi(\boldsymbol{w}^{(t-1)})\right] - \mathbb{E}\left[F(\boldsymbol{w}^{(t-1)}, \boldsymbol{\lambda}^{\lfloor\frac{t-1}{\tau}\rfloor})\right] - \frac{L}{2}\mathbb{E}\left[\left\|\tilde{\boldsymbol{w}}^{(t-1)} - \boldsymbol{w}^{(t-1)}\right\|^2\right]\right)$$

$$\le \Phi_{1/2L}(\boldsymbol{w}^{(t-1)}) + L\mathbb{E}\left[\left\|\tilde{\boldsymbol{w}}^{(t-1)} - \boldsymbol{w}^{(t-1)}\right\|^2\right]$$

$$+ L\eta^2(G_w^2 + \sigma_w^2) + \eta D_{\mathcal{W}}L^2\mathbb{E}\left[\frac{1}{m}\sum_{i\in\mathcal{D}^{(\lfloor\frac{t-1}{\tau}\rfloor)}}\left\|\boldsymbol{w}_i^{(t-1)} - \boldsymbol{w}^{(t-1)}\right\|\right]$$

$$+ 2L\eta\left(\mathbb{E}\left[\Phi(\boldsymbol{w}^{(t-1)})\right] - \mathbb{E}\left[F(\boldsymbol{w}^{(t-1)}, \boldsymbol{\lambda}^{\lfloor\frac{t-1}{\tau}\rfloor})\right]\right) - \frac{\eta}{4}\mathbb{E}\left[\left\|\nabla\Phi_{1/2L}(\boldsymbol{w}^{(t-1)})\right\|^2\right],$$

where we use the result from Lemma 2.2 in [7], i.e, $\nabla\Phi_{1/2L}(\boldsymbol{w}) = 2L(\boldsymbol{w} - \tilde{\boldsymbol{w}})$. $\qquad\square$

**Lemma 8.** *For* DRFA, $\forall\boldsymbol{\lambda}\in\Lambda$*, under the same conditions as in Theorem* 2*, the following statement holds true:*

$$\sum_{t=(s-1)\tau+1}^{s\tau}\left(\mathbb{E}\left[\Phi(\boldsymbol{w}^{(t)})\right] - \mathbb{E}\left[F(\boldsymbol{w}^{(t)}, \boldsymbol{\lambda}^{(s-1)})\right]\right)$$

$$\le \sum_{t=(s-1)\tau+1}^{s\tau}\left(\mathbb{E}\left[F(\boldsymbol{w}^{(t)}, \boldsymbol{\lambda}^*(\boldsymbol{w}^t))\right] - \mathbb{E}\left[F(\boldsymbol{w}^{(t)}, \boldsymbol{\lambda})\right]\right)$$

$$+ \gamma\tau^2\frac{\sigma_\lambda^2}{m} + \gamma\tau^2 G_\lambda^2 + \frac{1}{2\gamma}\left(\mathbb{E}\left[\left\|\boldsymbol{\lambda} - \boldsymbol{\lambda}^{(s-1)}\right\|^2\right] - \mathbb{E}\left[\left\|\boldsymbol{\lambda} - \boldsymbol{\lambda}^{(s)}\right\|^2\right]\right).$$

*Proof.* $\forall\boldsymbol{\lambda}\in\Lambda$, according to updating rule for $\boldsymbol{\lambda}^{(s-1)}$, we have:

$$\left\langle\boldsymbol{\lambda} - \boldsymbol{\lambda}^{(s)}, \boldsymbol{\lambda}^{(s)} - \boldsymbol{\lambda}^{(s-1)} - \Delta_{s-1}\right\rangle \ge 0.$$

Taking expectation on both sides, and doing some algebraic manipulation yields:

$$\mathbb{E}\left[\left\|\boldsymbol{\lambda} - \boldsymbol{\lambda}^{(s)}\right\|^2\right]$$

$$\le 2\mathbb{E}\left[\left\langle\boldsymbol{\lambda}^{(s-1)} - \boldsymbol{\lambda}, \Delta_{s-1}\right\rangle\right] + 2\mathbb{E}\left[\left\langle\boldsymbol{\lambda}^{(s)} - \boldsymbol{\lambda}^{(s-1)}, \Delta_{s-1}\right\rangle\right]$$

$$+ \mathbb{E}\left[\left\|\boldsymbol{\lambda} - \boldsymbol{\lambda}^{(s-1)}\right\|^2\right] - \mathbb{E}\left[\left\|\boldsymbol{\lambda}^{(s)} - \boldsymbol{\lambda}^{(s-1)}\right\|^2\right]$$

$$\le 2\mathbb{E}\left[\left\langle\boldsymbol{\lambda}^{(s-1)} - \boldsymbol{\lambda}, \bar{\Delta}_{s-1}\right\rangle\right] + 2\mathbb{E}\left[\left\langle\boldsymbol{\lambda}^{(s)} - \boldsymbol{\lambda}^{(s-1)}, \bar{\Delta}_{s-1}\right\rangle\right]$$

$$+ 2\mathbb{E}\left[\left\langle\boldsymbol{\lambda}^{(s)} - \boldsymbol{\lambda}^{(s-1)}, \Delta_{s-1} - \bar{\Delta}_{s-1}\right\rangle\right] + \mathbb{E}\left[\left\|\boldsymbol{\lambda} - \boldsymbol{\lambda}^{(s-1)}\right\|^2\right] - \mathbb{E}\left[\left\|\boldsymbol{\lambda}^{(s)} - \boldsymbol{\lambda}^{(s-1)}\right\|^2\right].$$

Applying the Cauchy-Schwartz and aritmetic mean-geometric mean inequality: $2\langle \boldsymbol{p}, \boldsymbol{q} \rangle \leq 2\|\boldsymbol{p}\|\|\boldsymbol{q}\| \leq \frac{1}{2}\|\boldsymbol{p}\|^2 + 2\|\boldsymbol{q}\|^2$, we have:

$$\mathbb{E}\left[\left\| \boldsymbol{\lambda} - \boldsymbol{\lambda}^{(s)} \right\|^2\right]$$

$$\leq 2\gamma \mathbb{E}\left[\sum_{t=(s-1)\tau+1}^{s\tau} F(\boldsymbol{w}^{(t)}, \boldsymbol{\lambda}^{(s-1)}) - F(\boldsymbol{w}^{(t)}, \boldsymbol{\lambda})\right] + \mathbb{E}\left[\left\|\boldsymbol{\lambda} - \boldsymbol{\lambda}^{(s-1)}\right\|^2\right]$$

$$+ \mathbb{E}\left[\frac{1}{2}\left\|\boldsymbol{\lambda}^{(s)} - \boldsymbol{\lambda}^{(s-1)}\right\|^2 + 2\left\|\Delta_{s-1} - \bar{\Delta}_{s-1}\right\|^2\right] + \mathbb{E}\left[\frac{1}{2}\left\|\boldsymbol{\lambda}^{(s)} - \boldsymbol{\lambda}^{(s-1)}\right\|^2 + 2\left\|\bar{\Delta}_{s-1}\right\|^2\right]$$

$$- \mathbb{E}\left[\left\|\boldsymbol{\lambda}^{(s)} - \boldsymbol{\lambda}^{(s-1)}\right\|^2\right]$$

$$\leq 2\gamma \mathbb{E}\left[\sum_{t=(s-1)\tau+1}^{s\tau} F(\boldsymbol{w}^{(t)}, \boldsymbol{\lambda}^{(s-1)}) - F(\boldsymbol{w}^{(t)}, \boldsymbol{\lambda})\right] + \gamma^2\tau^2\frac{\sigma_\lambda^2}{m} + \gamma^2\tau^2 G_\lambda^2 + \mathbb{E}\left[\left\|\boldsymbol{\lambda} - \boldsymbol{\lambda}^{(s-1)}\right\|^2\right].$$

By adding $\sum_{t=(s-1)\tau+1}^{s\tau} F(\boldsymbol{w}^{(t)}, \boldsymbol{\lambda}^*(\boldsymbol{w}^{(t)}))$ on both sides and re-arranging the terms we have:

$$\sum_{t=(s-1)\tau+1}^{s\tau}\left(\mathbb{E}\left[\Phi(\boldsymbol{w}^{(t)})\right] - \mathbb{E}\left[F(\boldsymbol{w}^{(t)}, \boldsymbol{\lambda}^{(s-1)})\right]\right)$$

$$\leq \sum_{t=(s-1)\tau+1}^{s\tau}\left(\mathbb{E}\left[F(\boldsymbol{w}^{(t)}, \boldsymbol{\lambda}^*(\boldsymbol{w}^t))\right] - \mathbb{E}\left[F(\boldsymbol{w}^{(t)}, \boldsymbol{\lambda})\right]\right) + \gamma\tau^2\frac{\sigma_\lambda^2}{m} + \gamma\tau^2 G_\lambda^2$$

$$+ \frac{1}{2\gamma}\left(\mathbb{E}\left[\left\|\boldsymbol{\lambda} - \boldsymbol{\lambda}^{(s-1)}\right\|^2\right] - \mathbb{E}\left[\left\|\boldsymbol{\lambda} - \boldsymbol{\lambda}^{(s)}\right\|^2\right]\right).$$

$\square$

**Lemma 9.** *For* DRFA, *under the assumptions in Theorem* 2, *the following statement holds true:*

$$\frac{1}{T}\sum_{t=1}^{T}\left(\mathbb{E}\left[\Phi(\boldsymbol{w}^{(t)})\right] - \mathbb{E}\left[F(\boldsymbol{w}^{(t)}, \boldsymbol{\lambda}^{(\lfloor \frac{t}{\tau}\rfloor)})\right]\right) \leq 2\sqrt{S}\tau\eta G_w\sqrt{G_w^2 + \sigma_w^2} + \gamma\tau\frac{\sigma_\lambda^2}{m} + \gamma\tau G_\lambda^2 + \frac{D_\Lambda^2}{2\sqrt{S}\tau\gamma}$$

*Proof.* Without loss of generality we assume $\sqrt{S}$ is an integer, so we can equally divide index 0 to $S - 1$ into $\sqrt{S}$ groups. Then we have:

$$\frac{1}{T}\sum_{t=1}^{T}\left(\mathbb{E}\left[\Phi(\boldsymbol{w}^{(t)})\right] - \mathbb{E}\left[F(\boldsymbol{w}^{(t)}, \boldsymbol{\lambda}^{(\lfloor \frac{t}{\tau}\rfloor)})\right]\right)$$

$$= \frac{1}{T}\sum_{c=0}^{\sqrt{S}-1}\left[\sum_{s=c\sqrt{S}+1}^{(c+1)\sqrt{S}}\sum_{t=(s-1)\tau+1}^{s\tau}\left(\mathbb{E}\left[\Phi(\boldsymbol{w}^{(t)})\right] - \mathbb{E}\left[F(\boldsymbol{w}^{(t)}, \boldsymbol{\lambda}^{(s-1)})\right]\right)\right]. \tag{19}$$

Now we and examine one group. Plugging in Lemma 8 and letting $\boldsymbol{\lambda} = \boldsymbol{\lambda}^*(\boldsymbol{w}^{(c+1)\sqrt{S}\tau})$ yields:

$$
\sum_{s=c\sqrt{S}+1}^{(c+1)\sqrt{S}} \sum_{t=(s-1)\tau+1}^{s\tau} \left( \mathbb{E}\left[\Phi(\boldsymbol{w}^{(t)})\right] - \mathbb{E}\left[F(\boldsymbol{w}^{(t)}, \boldsymbol{\lambda}^{(s-1)})\right] \right)
$$

$$
\leq \sum_{s=c\sqrt{S}+1}^{(c+1)\sqrt{S}} \sum_{t=(s-1)\tau+1}^{s\tau} \left( \mathbb{E}\left[F(\boldsymbol{w}^{(t)}, \boldsymbol{\lambda}^*(\boldsymbol{w}^t))\right] - \mathbb{E}\left[F(\boldsymbol{w}^{(t)}, \boldsymbol{\lambda}^*(\boldsymbol{w}^{(c+1)\sqrt{S}\tau}))\right] \right)
$$

$$
+ \gamma\tau^2 \frac{\sqrt{S}\sigma_\lambda^2}{m} + \gamma\tau^2\sqrt{S}G_\lambda^2 + \frac{1}{2\gamma}\sum_{s=c\sqrt{S}+1}^{(c+1)\sqrt{S}} \left( \mathbb{E}\left[\left\|\boldsymbol{\lambda}^*(\boldsymbol{w}^{(c+1)\sqrt{S}\tau}) - \boldsymbol{\lambda}^{(s-1)}\right\|^2\right] - \mathbb{E}\left[\left\|\boldsymbol{\lambda}^*(\boldsymbol{w}^{(c+1)\sqrt{S}\tau}) - \boldsymbol{\lambda}^{(s)}\right\|^2\right] \right)
$$

$$
\leq \sum_{s=c\sqrt{S}+1}^{(c+1)\sqrt{S}} \sum_{t=(s-1)\tau+1}^{s\tau} \left( \mathbb{E}\left[F(\boldsymbol{w}^{(t)}, \boldsymbol{\lambda}^*(\boldsymbol{w}^t))\right] - \mathbb{E}\left[F(\boldsymbol{w}^{((c+1)\sqrt{S}\tau)}, \boldsymbol{\lambda}^*(\boldsymbol{w}^t))\right] \right.
$$

$$
\left. + \mathbb{E}\left[F(\boldsymbol{w}^{((c+1)\sqrt{S}\tau)}, \boldsymbol{\lambda}^*(\boldsymbol{w}^{(c+1)\sqrt{S}\tau}))\right] - \mathbb{E}\left[F(\boldsymbol{w}^{(t)}, \boldsymbol{\lambda}^*(\boldsymbol{w}^{(c+1)\sqrt{S}\tau}))\right] \right)
$$

$$
+ \gamma\tau^2 \frac{\sqrt{S}\sigma_\lambda^2}{m} + \gamma\tau^2\sqrt{S}G_\lambda^2 + \frac{1}{2\gamma}\sum_{s=c\sqrt{S}+1}^{(c+1)\sqrt{S}} \left( \mathbb{E}\left[\left\|\boldsymbol{\lambda}^*(\boldsymbol{w}^{(c+1)\sqrt{S}\tau}) - \boldsymbol{\lambda}^{(s-1)}\right\|^2\right] - \mathbb{E}\left[\left\|\boldsymbol{\lambda}^*(\boldsymbol{w}^{(c+1)\sqrt{S}\tau}) - \boldsymbol{\lambda}^{(s)}\right\|^2\right] \right)
$$

$$
\tag{20}
$$

$$
\leq \sum_{s=c\sqrt{S}+1}^{(c+1)\sqrt{S}} \sum_{t=(s-1)\tau+1}^{s\tau} (2\sqrt{S}\tau\eta G_w\sqrt{G_w^2 + \sigma_w^2}) + \gamma\tau\frac{\sqrt{S}\sigma_\lambda^2}{m} + \gamma\tau\sqrt{S}G_\lambda^2 + \frac{D_\Lambda^2}{2\gamma} \tag{21}
$$

$$
\leq 2S\tau^2\eta G_w\sqrt{G_w^2 + \sigma_w^2} + \gamma\tau^2\frac{\sqrt{S}\sigma_\lambda^2}{m} + \gamma\tau^2\sqrt{S}G_\lambda^2 + \frac{D_\Lambda^2}{2\gamma}, \tag{22}
$$

where from (20) to (21) we use the $G_w$-Lipschitz property of $F(\cdot, \boldsymbol{\lambda})$ so that $F(\boldsymbol{w}^{t_1}, \boldsymbol{\lambda}) - F(\boldsymbol{w}^{t_2}, \boldsymbol{\lambda}) \leq G_w\|\boldsymbol{w}^{t_1} - \boldsymbol{w}^{t_2}\|$.

Now plugging (22) back to (19) yields:

$$
\frac{1}{T}\sum_{t=1}^T \left( \mathbb{E}\left[\Phi(\boldsymbol{w}^{(t)})\right] - \mathbb{E}\left[F(\boldsymbol{w}^{(t)}, \boldsymbol{\lambda}^{(s)})\right] \right) \leq \frac{1}{T}2\sqrt{S}S\tau^2\eta G_w\sqrt{G_w^2 + \sigma_w^2} + \gamma\tau\frac{\sigma_\lambda^2}{m} + \gamma\tau G_\lambda^2 + \frac{\sqrt{S}D_\Lambda^2}{2T\gamma}
$$

$$
\leq 2\sqrt{S}\tau\eta G_w\sqrt{G_w^2 + \sigma_w^2} + \gamma\tau\frac{\sigma_\lambda^2}{m} + \gamma\tau G_\lambda^2 + \frac{D_\Lambda^2}{2\sqrt{S}\tau\gamma}.
$$

$\square$

## D.3 Proof of Theorem 2

Now we proceed to the formal proof of Theorem 2. Re-arranging terms in Lemma 7, summing over $t = 1$ to $T$, and dividing by $T$ yields:

$$
\frac{1}{T}\sum_{t=1}^T \mathbb{E}\left[\left\|\nabla\Phi_{1/2L}\ (\boldsymbol{w}^{(t)})\right\|^2\right]
$$

$$
\leq \frac{4}{\eta T}\mathbb{E}[\Phi_{1/2L}(\boldsymbol{w}^{(0)})] + \frac{1}{2T}\sum_{t=1}^T D_\mathcal{W} L^2 \mathbb{E}\left[\frac{1}{m}\sum_{i\in\mathcal{D}^{(\lfloor\frac{t}{\tau}\rfloor)}} \left\|\boldsymbol{w}_i^{(t)} - \boldsymbol{w}^{(t)}\right\|\right]
$$

$$
+ L\frac{1}{2T}\sum_{t=1}^T \left( \mathbb{E}[\Phi(\boldsymbol{w}^{(t)})] - \mathbb{E}[F(\boldsymbol{w}^{(t)}, \boldsymbol{\lambda}^{\lfloor\frac{t}{\tau}\rfloor})] \right).
$$

Plugging in Lemma 5 and 9 yields:

$$\frac{1}{T}\sum_{t=1}^{T}\mathbb{E}\left[\left\|\nabla\Phi_{1/2L}(\boldsymbol{w}^{(t)})\right\|^2\right] \leq \frac{4}{\eta T}\mathbb{E}[\Phi_{1/2L}(\boldsymbol{w}^{(0)})] + \eta\tau D_{\mathcal{W}}L^2\left(\sigma_w + \frac{\sigma_w}{m} + \sqrt{\Gamma}\right).$$

$$+ \frac{L}{2}\left(2\sqrt{S}\tau\eta G_w\sqrt{G_w^2 + \sigma_w^2} + \gamma\tau\frac{\sigma_\lambda^2}{m} + \gamma\tau G_\lambda^2 + \frac{D_\Lambda^2}{2\sqrt{S}\tau\gamma}\right)$$

$$\leq \frac{4}{\eta T}\mathbb{E}[\Phi_{1/2L}(\boldsymbol{w}^{(0)})] + \eta\tau D_{\mathcal{W}}L^2\left(\sigma_w + \frac{\sigma_w}{m} + \sqrt{\Gamma}\right)$$

$$+ \sqrt{S}\tau\eta G_w L\sqrt{G_w^2 + \sigma_w^2} + \gamma\tau\frac{\sigma_\lambda^2 L}{2m} + \gamma\tau\frac{G_\lambda^2 L}{2} + \frac{D_\Lambda^2 L}{4\sqrt{S}\tau\gamma}.$$

Plugging in $\eta = \frac{1}{4LT^{3/4}}$, $\gamma = \frac{1}{T^{1/2}}$ and $\tau = T^{1/4}$ we recover the convergence rate as cliamed:

$$\frac{1}{T}\sum_{t=1}^{T}\mathbb{E}\left[\left\|\nabla\Phi_{1/2L}(\boldsymbol{w}^{(t)})\right\|^2\right] \leq \frac{4}{T^{1/4}}\mathbb{E}[\Phi_{1/2L}(\boldsymbol{w}^{(0)})] + \frac{L^2}{T^{1/2}}\left(\sigma_w + \frac{\sigma_w}{m} + \sqrt{\Gamma}\right)$$

$$+ \frac{1}{T^{1/8}}G_w L\sqrt{G_w^2 + \sigma_w^2} + \frac{\sigma_\lambda^2 L}{2mT^{1/4}} + \frac{G_\lambda^2 L}{2T^{1/4}} + \frac{D_\Lambda^2 L}{4T^{1/8}},$$

which concludes the proof. □

## E  Proof of Convergence of **DRFA-Prox**

This section is devoted to the proof of convergence of DRFA-Prox algorithm in both convex and nonconvex settings.

### E.1  Convex Setting

In this section we are going to provide the proof of Theorem 3, the convergence of DRFA-Prox on convex losses, i.e., global objective $F$ is convex in $\boldsymbol{w}$. Let us first introduce a key lemma:

**Lemma 10.** *For DRFA-Prox, $\forall\boldsymbol{\lambda}\in\Lambda$, and for any $s$ such that $0 \leq s \leq \frac{T}{\tau} - 1$ we have:*

$$\sum_{t=s\tau+1}^{(s+1)\tau}\left(\mathbb{E}\left[F(\boldsymbol{w}^{(t)},\boldsymbol{\lambda})\right] - \mathbb{E}\left[F(\boldsymbol{w}^{(t)},\boldsymbol{\lambda}^{(s)})\right]\right)$$

$$\leq -\frac{1}{2\gamma}\mathbb{E}\|\boldsymbol{\lambda}^{(s+1)} - \boldsymbol{\lambda}\|^2 + \frac{1}{2\gamma}\mathbb{E}[\|\boldsymbol{\lambda}^{(s)} - \boldsymbol{\lambda}\|^2] + \frac{1}{2\gamma}\mathbb{E}[\|\bar{\Delta}_s - \Delta_s\|^2]$$

$$+ \tau^2\gamma G_w(G_w + \sqrt{G_w^2 + G_\lambda^2 + \sigma_\lambda^2}) + \tau^2\gamma G_\lambda^2$$

*Proof.* Recall that to update $\boldsymbol{\lambda}^{(s)}$, we sampled a index $t'$ from $s\tau + 1$ to $(s+1)\tau$, and obtain the averaged model $\boldsymbol{w}^{(t')}$. Now, consider iterations from $s\tau + 1$ to $(s+1)\tau$. Define following function:

$$\Psi(\boldsymbol{u}) = \tau f(\boldsymbol{w}^{(t')},\boldsymbol{y}) + \tau g(\boldsymbol{u}) - \frac{1}{2\gamma}\|\boldsymbol{y} + \Delta_s - \boldsymbol{u}\|^2$$

$$= \tau f(\boldsymbol{w}^{(t')},\boldsymbol{y}) + \tau g(\boldsymbol{u}) - \frac{1}{2\gamma}\|\boldsymbol{y} + \bar{\Delta}_s - \boldsymbol{u}\|^2 - \frac{1}{2\gamma}\|\bar{\Delta}_s - \Delta_s\|^2 \quad (23)$$

$$+ \frac{1}{\gamma}\langle\bar{\Delta}_s - \Delta_s, \boldsymbol{y} + \bar{\Delta}_s - \boldsymbol{u}\rangle.$$

By taking the expectation on both side, we get:

$$\mathbb{E}[\Psi(\boldsymbol{u})]$$

$$= \mathbb{E}[\tau f(\boldsymbol{w}^{(t')},\boldsymbol{y})] + \frac{1}{\gamma}\mathbb{E}[\langle\bar{\Delta}_s, \boldsymbol{u} - \boldsymbol{y}\rangle] + \mathbb{E}[\tau g(\boldsymbol{u})] - \frac{1}{2\gamma}\mathbb{E}\|\boldsymbol{u} - \boldsymbol{y}\|^2 - \frac{1}{2\gamma}\mathbb{E}\|\bar{\Delta}_s - \Delta_s\|^2 - \frac{1}{2\gamma}\mathbb{E}\|\bar{\Delta}_s\|^2$$

$$= \mathbb{E}\left[\sum_{t=s\tau+1}^{(s+1)\tau} F(\boldsymbol{w}^{(t)},\boldsymbol{u})\right] - \frac{1}{2\gamma}\mathbb{E}\|\boldsymbol{u} - \boldsymbol{y}\|^2 - \frac{1}{2\gamma}\mathbb{E}\|\bar{\Delta}_s - \Delta_s\|^2 - \frac{1}{2\gamma}\mathbb{E}\|\bar{\Delta}_s\|^2$$

where we used the fact that $\mathbb{E}[\tau f(\boldsymbol{w}^{(t')}, \boldsymbol{y})] = \mathbb{E}\left[\sum_{t=s\tau+1}^{(s+1)\tau} f(\boldsymbol{w}^{(t)}, \boldsymbol{y})\right]$ and $\frac{1}{\gamma}\mathbb{E}[\langle \Delta_s, \boldsymbol{u} - \boldsymbol{y}\rangle] = \sum_{t=s\tau+1}^{(s+1)\tau} \mathbb{E}\left[f(\boldsymbol{w}^{(t)}, \boldsymbol{u}) - f(\boldsymbol{w}^{(t)}, \boldsymbol{y})\right]$.

Define the operator:

$$T_g(\boldsymbol{y}) := \arg\max_{\boldsymbol{u}\in\Lambda} \left\{ \tau g(\boldsymbol{u}) - \frac{1}{2\gamma}\|\boldsymbol{y} + \Delta_s - \boldsymbol{u}\|^2 \right\} \tag{24}$$

Since $\Psi(\boldsymbol{u})$ is $\frac{1}{2\gamma}$-strongly concave, and $T_g(\boldsymbol{y})$ is the maximizer of $\Psi(\boldsymbol{u})$, we have:

$$\mathbb{E}[\Psi(T_g(\boldsymbol{y}))] - \mathbb{E}[\Psi(\boldsymbol{u})] \geq \frac{1}{2\gamma}\mathbb{E}\|T_g(\boldsymbol{y}) - \boldsymbol{u}\|^2$$

Notice that:

$$\mathbb{E}[\Psi(T_g(\boldsymbol{y}))] = \mathbb{E}\left[\sum_{t=s\tau+1}^{(s+1)\tau} F(\boldsymbol{w}^{(t)}, T_g(\boldsymbol{y}))\right] - \frac{1}{2\gamma}\mathbb{E}[\|T_g(\boldsymbol{y}) - \boldsymbol{y}\|^2] - \frac{1}{2\gamma}\mathbb{E}[\|\bar{\Delta}_s - \Delta_s\|^2] - \frac{1}{2\gamma}\mathbb{E}\|\bar{\Delta}_s\|^2$$

So we know that $\mathbb{E}\left[\sum_{t=s\tau+1}^{(s+1)\tau} F(\boldsymbol{w}^{(t)}, T_g(\boldsymbol{y}))\right] \geq \mathbb{E}[\Psi(T_g(\boldsymbol{y}))]$, and hence:

$$\mathbb{E}\left[\sum_{t=s\tau+1}^{(s+1)\tau} F(\boldsymbol{w}^{(t)}, T_g(\boldsymbol{y}))\right] - \mathbb{E}[\Psi(\boldsymbol{u})] \geq \mathbb{E}[\Psi(T_g(\boldsymbol{y}))] - \mathbb{E}[\Psi(\boldsymbol{u})] \geq \frac{1}{2\gamma}\mathbb{E}\|T_g(\boldsymbol{y}) - \boldsymbol{u}\|^2$$

Plugging in $\mathbb{E}[\Psi(\boldsymbol{u})]$ results in:

$$\mathbb{E}\left[\sum_{t=s\tau+1}^{(s+1)\tau} F(\boldsymbol{w}^{(t)}, T_g(\boldsymbol{y}))\right] - \mathbb{E}[\Psi(\boldsymbol{u})]$$

$$= \mathbb{E}\left[\sum_{t=s\tau+1}^{(s+1)\tau} F(\boldsymbol{w}^{(t)}, T_g(\boldsymbol{y}))\right] - \left(\mathbb{E}\left[\sum_{t=s\tau+1}^{(s+1)\tau} F(\boldsymbol{w}^{(t)}, \boldsymbol{u})\right] - \frac{1}{2\gamma}\mathbb{E}[\|\boldsymbol{u} - \boldsymbol{y}\|^2] - \frac{1}{2\gamma}\mathbb{E}[\|\bar{\Delta}_s - \Delta_s\|^2] - \frac{1}{2\gamma}\mathbb{E}\|\bar{\Delta}_s\|^2\right)$$

$$\geq \frac{1}{2\gamma}\mathbb{E}\|T_g(\boldsymbol{y}) - \boldsymbol{u}\|^2.$$

Re-arranging the terms yields:

$$\mathbb{E}\left[\sum_{t=s\tau+1}^{(s+1)\tau} F(\boldsymbol{w}^{(t)}, \boldsymbol{u})\right] - \mathbb{E}\left[\sum_{t=s\tau+1}^{(s+1)\tau} F(\boldsymbol{w}^{(t)}, T_g(\boldsymbol{y}))\right]$$

$$\leq -\frac{1}{2\gamma}\mathbb{E}\|T_g(\boldsymbol{y}) - \boldsymbol{u}\|^2 + \frac{1}{2\gamma}\mathbb{E}[\|\boldsymbol{y} - \boldsymbol{u}\|^2] + \frac{1}{2\gamma}\mathbb{E}[\|\bar{\Delta}_s - \Delta_s\|^2] + \frac{1}{2\gamma}\mathbb{E}\|\bar{\Delta}_s\|^2.$$
$$\tag{25}$$

Let $\boldsymbol{u} = \boldsymbol{\lambda}$, $\boldsymbol{y} = \boldsymbol{\lambda}^{(s)}$, then we have:

$$\sum_{t=s\tau+1}^{(s+1)\tau} \left(\mathbb{E}\left[F(\boldsymbol{w}^{(t)}, \boldsymbol{\lambda})\right] - \mathbb{E}\left[F(\boldsymbol{w}^{(t)}, T_g(\boldsymbol{\lambda}^{(s)}))\right]\right)$$

$$\leq -\frac{1}{2\gamma}\mathbb{E}\|T_g(\boldsymbol{\lambda}^{(s)}) - \boldsymbol{\lambda}\|^2 + \frac{1}{2\gamma}\mathbb{E}[\|\boldsymbol{\lambda}^{(s)} - \boldsymbol{\lambda}\|^2] + \frac{1}{2\gamma}\mathbb{E}[\|\bar{\Delta}_s - \Delta_s\|^2] + \frac{1}{2\gamma}\mathbb{E}\|\bar{\Delta}_s\|^2.$$

Since $T_g(\boldsymbol{\lambda}^{(s)}) = \boldsymbol{\lambda}^{(s+1)}$, we have:

$$\sum_{t=s\tau+1}^{(s+1)\tau} \left(\mathbb{E}\left[F(\boldsymbol{w}^{(t)}, \boldsymbol{\lambda})\right] - \mathbb{E}\left[F(\boldsymbol{w}^{(t)}, \boldsymbol{\lambda}^{(s)})\right]\right)$$

$$\leq -\frac{1}{2\gamma}\mathbb{E}\|T_g(\boldsymbol{\lambda}^{(s)}) - \boldsymbol{\lambda}\|^2 + \frac{1}{2\gamma}\mathbb{E}[\|\boldsymbol{\lambda}^{(s)} - \boldsymbol{\lambda}\|^2] + \frac{1}{2\gamma}\mathbb{E}[\|\bar{\Delta}_s - \Delta_s\|^2] + \frac{1}{2\gamma}\mathbb{E}\|\bar{\Delta}_s\|^2$$

$$+ \underbrace{\sum_{t=s\tau+1}^{(s+1)\tau} \left(\mathbb{E}\left[F(\boldsymbol{w}^{(t)}, \boldsymbol{\lambda}^{(s+1)})\right] - \mathbb{E}\left[F(\boldsymbol{w}^{(t)}, \boldsymbol{\lambda}^{(s)})\right]\right)}_{T_1}.$$

Now our remaining task is to bound $T_1$. By the Lipschitz property of $F$, we have the following upper bound for $T_1$:

$$T_1 \leq \tau G_w \mathbb{E}\|\boldsymbol{\lambda}^{(s+1)} - \boldsymbol{\lambda}^{(s)}\|. \tag{26}$$

Then, by plugging $\boldsymbol{u} = \boldsymbol{\lambda}^{(s)}$, $\boldsymbol{y} = \boldsymbol{\lambda}^{(s)}$ into (25), we have the following lower bound:

$$\frac{1}{2\gamma}\mathbb{E}\|\boldsymbol{\lambda}^{(s+1)} - \boldsymbol{\lambda}^{(s)}\|^2 - \frac{1}{2\gamma}\mathbb{E}[\|\bar{\Delta}_s - \Delta_s\|^2] - \frac{1}{2\gamma}\mathbb{E}\|\bar{\Delta}_s\|^2 \leq T_1. \tag{27}$$

Combining (26) and (27) we have:

$$\frac{1}{2\gamma}\mathbb{E}\|\boldsymbol{\lambda}^{(s+1)} - \boldsymbol{\lambda}^{(s)}\|^2 - \frac{1}{2\gamma}\mathbb{E}[\|\bar{\Delta}_s - \Delta_s\|^2] - \frac{1}{2\gamma}\mathbb{E}\|\bar{\Delta}_s\|^2$$

$$\leq \tau G_w \mathbb{E}\|\boldsymbol{\lambda}^{(s+1)} - \boldsymbol{\lambda}^{(s)}\| \leq \tau G_w \sqrt{\mathbb{E}\|\boldsymbol{\lambda}^{(s+1)} - \boldsymbol{\lambda}^{(s)}\|^2}. \tag{28}$$

Let $X = \sqrt{\mathbb{E}\|\boldsymbol{\lambda}^{(s+1)} - \boldsymbol{\lambda}^{(s)}\|^2}$, $A = \frac{1}{2\gamma}$, $B = -\tau G_w$ and $C = -\frac{1}{2\gamma}\mathbb{E}[\|\bar{\Delta}_s - \Delta_s\|^2] - \frac{1}{2\gamma}\mathbb{E}\|\bar{\Delta}_s\|^2$, then we can re-formulate (28) as:

$$AX^2 + BX + C \leq 0. \tag{29}$$

Obviously $A \geq 0$. According to the root of quadratic equation, we know that:

$$X \leq \frac{-B + \sqrt{B^2 - 4AC}}{2A} = \frac{\tau G_w + \sqrt{G_w^2 \tau^2 + \frac{1}{\gamma^2}(\mathbb{E}[\|\bar{\Delta}_s - \Delta_s\|^2] + \mathbb{E}\|\bar{\Delta}_s\|^2)}}{1/\gamma}$$

$$\leq \tau\gamma\left(G_w + \sqrt{G_w^2 + G_\lambda^2 + \sigma_\lambda^2}\right).$$

Hence, we have

$$T_1 \leq \tau G_w \mathbb{E}\|\boldsymbol{\lambda}^{(s+1)} - \boldsymbol{\lambda}^{(s)}\| \leq \tau^2 \gamma G_w \left(G_w + \sqrt{G_w^2 + G_\lambda^2 + \sigma_\lambda^2}\right),$$

which concludes the proof.

$$\square$$

**Proof of Theorem 3.** We start the proof by noting that $\forall \boldsymbol{w} \in \mathcal{W}$, $\forall \boldsymbol{\lambda} \in \Lambda$, according the convexity in $\boldsymbol{w}$ and concavity in $\boldsymbol{\lambda}$, we have:

$$\mathbb{E}[F(\hat{\boldsymbol{w}}, \boldsymbol{\lambda}) - \mathbb{E}[F(\boldsymbol{w}, \hat{\boldsymbol{\lambda}})]$$

$$\leq \frac{1}{T}\sum_{t=1}^{T}\left\{\mathbb{E}\left[F(\boldsymbol{w}^{(t)}, \boldsymbol{\lambda})\right] - \mathbb{E}\left[F(\boldsymbol{w}, \boldsymbol{\lambda}^{(\lfloor\frac{t}{\tau}\rfloor)})\right]\right\}$$

$$\leq \frac{1}{T}\sum_{t=1}^{T}\left\{\mathbb{E}\left[F(\boldsymbol{w}^{(t)}, \boldsymbol{\lambda})\right] - \mathbb{E}\left[F(\boldsymbol{w}^{(t)}, \boldsymbol{\lambda}^{(\lfloor\frac{t}{\tau}\rfloor)})\right] + \mathbb{E}\left[F(\boldsymbol{w}^{(t)}, \boldsymbol{\lambda}^{(\lfloor\frac{t}{\tau}\rfloor)})\right] - \mathbb{E}\left[F(\boldsymbol{w}, \boldsymbol{\lambda}^{(\lfloor\frac{t}{\tau}\rfloor)})\right]\right\}$$

$$\leq \frac{1}{T}\sum_{s=0}^{S-1}\sum_{t=s\tau+1}^{(s+1)\tau}\mathbb{E}[F(\boldsymbol{w}^{(t)}, \boldsymbol{\lambda}) - F(\boldsymbol{w}^{(t)}, \boldsymbol{\lambda}^{(s)})] + \frac{1}{T}\sum_{t=1}^{T}\mathbb{E}[F(\boldsymbol{w}^{(t)}, \boldsymbol{\lambda}^{(\lfloor\frac{t}{\tau}\rfloor)}) - F(\boldsymbol{w}, \boldsymbol{\lambda}^{(\lfloor\frac{t}{\tau}\rfloor)})].$$

$$\tag{30}$$

To bound the first term in (30), plugging Lemma 2 into Lemma 10, and summing over $s = 0$ to $S - 1$ where $S = T/\tau$, and dividing both sides with $T$ yields:

$$\frac{1}{T}\sum_{s=0}^{S-1}\sum_{t=s\tau+1}^{(s+1)\tau}\left\{\mathbb{E}\left[F(\boldsymbol{w}^{(t)}, \boldsymbol{\lambda})\right] - \mathbb{E}\left[F(\boldsymbol{w}^{(t)}, \boldsymbol{\lambda}^{(s)})\right]\right\}$$

$$\leq \frac{1}{2\gamma T}D_\Lambda^2 + \frac{1}{2\gamma\tau}\mathbb{E}[\|\bar{\Delta}_s - \Delta_s\|^2] + \tau\gamma G_w(G_w + \sqrt{G_w^2 + G_\lambda^2 + \sigma_\lambda^2}) + \gamma\tau G_\lambda^2$$

$$\leq \frac{1}{2\gamma T}D_\Lambda^2 + \frac{1}{2\gamma}\mathbb{E}[\|\bar{\Delta}_s - \Delta_s\|^2] + \tau\gamma(G_w + \sqrt{G_w^2 + G_\lambda^2 + \sigma_\lambda^2}) + \gamma\tau G_\lambda^2$$

$$\leq \frac{D_\Lambda^2}{2\gamma T} + \frac{\gamma\tau\sigma_\lambda^2}{2m} + \tau\gamma G_w(G_w + \sqrt{G_w^2 + G_\lambda^2 + \sigma_\lambda^2}) + \gamma\tau G_\lambda^2.$$

To bound the second term in (30), we plug Lemma 1 and Lemma 4 into Lemma 3 and apply the telescoping sum from $t = 1$ to $T$ to get:

$$\frac{1}{T}\sum_{t=1}^{T}\mathbb{E}[F(\boldsymbol{w}^{(t)}, \boldsymbol{\lambda}^{(\lfloor \frac{t}{\tau} \rfloor)}) - F(\boldsymbol{w}, \boldsymbol{\lambda}^{(\lfloor \frac{t}{\tau} \rfloor)})]$$

$$\leq \frac{1}{2T\eta}\mathbb{E}\| \boldsymbol{w}^{(0)} - \boldsymbol{w}\|^2 + 5L\eta^2\tau^2\left(\sigma_w^2 + \frac{\sigma_w^2}{m} + \Gamma\right) + \frac{\eta G_w^2}{2} + \frac{\eta\sigma_w^2}{2m}$$

$$\leq \frac{D_{\mathcal{W}}^2}{2T\eta} + 5L\eta^2\tau^2\left(\sigma_w^2 + \frac{\sigma_w^2}{m} + \Gamma\right) + \frac{\eta G_w^2}{2} + \frac{\eta\sigma_w^2}{2m},$$

So that we can conclude:

$$\mathbb{E}[F(\hat{\boldsymbol{w}}, \boldsymbol{\lambda}) - \mathbb{E}[F(\boldsymbol{w}, \hat{\boldsymbol{\lambda}})] \leq \frac{D_{\mathcal{W}}^2}{2T\eta} + 5L\eta^2\tau^2\left(\sigma_w^2 + \frac{\sigma_w^2}{m} + \Gamma\right) + \frac{\eta G_w^2}{2} + \frac{\eta\sigma_w^2}{2m} + \frac{D_{\Lambda}^2}{2\gamma T}$$

$$+ \gamma\tau G_\lambda^2 + \frac{\gamma\tau\sigma_\lambda^2}{2m} + \tau\gamma G_w(G_w + \sqrt{G_w^2 + G_\lambda^2 + \sigma_\lambda^2}).$$

Since the RHS does not depend on $\boldsymbol{w}$ and $\boldsymbol{\lambda}$, we can maximize over $\boldsymbol{\lambda}$ and minimize over $\boldsymbol{w}$ on both sides:

$$\min_{\boldsymbol{w} \in \mathcal{W}} \max_{\boldsymbol{\lambda} \in \Lambda} \mathbb{E}[F(\hat{\boldsymbol{w}}, \boldsymbol{\lambda}) - \mathbb{E}[F(\boldsymbol{w}, \hat{\boldsymbol{\lambda}})]$$

$$\leq \frac{D_{\mathcal{W}}^2}{2T\eta} + 5L\eta^2\tau^2\left(\sigma_w^2 + \frac{\sigma_w^2}{m} + \Gamma\right) + \frac{\eta G_w^2}{2} + \frac{\eta\sigma_w^2}{2m} + \frac{D_{\Lambda}^2}{2\gamma T}$$

$$+ \gamma\tau G_\lambda^2 + \frac{\gamma\tau\sigma_\lambda^2}{2m} + \tau\gamma G_w\left(G_w + \sqrt{G_w^2 + G_\lambda^2 + \sigma_\lambda^2}\right).$$

Plugging in $\tau = \frac{T^{1/4}}{\sqrt{m}}$, $\eta = \frac{1}{4L\sqrt{T}}$, and $\gamma = \frac{1}{T^{5/8}}$, we get:

$$\max_{\boldsymbol{\lambda} \in \Lambda}\mathbb{E}[F(\hat{\boldsymbol{w}}, \boldsymbol{\lambda})] - \min_{\boldsymbol{w} \in \mathcal{W}}\mathbb{E}[F(\boldsymbol{w}, \hat{\boldsymbol{\lambda}})] \leq O\left(\frac{D_{\mathcal{W}}^2 + G_w^2}{\sqrt{T}} + \frac{D_{\Lambda}^2 + G_w^2}{T^{3/8}} + \frac{G_\lambda^2}{m^{1/2}T^{3/8}} + \frac{\sigma_\lambda^2}{m^{3/2}T^{3/8}} + \frac{\sigma_w^2 + \Gamma}{m\sqrt{T}}\right),$$

thus concluding the proof.

### E.2 Nonconvex Setting

In this section we are going to prove Theorem 4. The whole framework is similar to the proof of Theorem 3, but to bound $\mathbb{E}\left[\Phi(\boldsymbol{w}^{(t)})\right] - \mathbb{E}\left[F(\boldsymbol{w}^{(t)}, \boldsymbol{\lambda}^{(\lfloor \frac{t}{\tau} \rfloor)})\right]$ term, we employ different technique for proximal method. The following lemma characterize the bound of $\mathbb{E}\left[\Phi(\boldsymbol{w}^{(t)})\right] - \mathbb{E}\left[F(\boldsymbol{w}^{(t)}, \boldsymbol{\lambda}^{(\lfloor \frac{t}{\tau} \rfloor)})\right]$:

**Lemma 11.** *For* DRFA-Prox, *under Theorem 4's assumption, the following statement holds true:*

$$\frac{1}{T}\sum_{t=1}^{T}\mathbb{E}\left[\Phi(\boldsymbol{w}^{(t)}) - F(\boldsymbol{w}^{(t)}, \boldsymbol{\lambda}^{(\lfloor \frac{t}{\tau} \rfloor)})\right]$$

$$\leq 2\sqrt{S}\tau\eta G_w\sqrt{G_w^2 + \sigma_w^2} + \gamma\tau\frac{\sigma_\lambda^2}{2m} + \gamma\tau\frac{G_\lambda^2}{2} + \frac{D_{\Lambda}^2}{2\sqrt{S}\tau} + \tau\gamma G_w\left(G_w + \sqrt{G_w^2 + G_\lambda^2 + \sigma_\lambda^2}\right).$$

*Proof.* We recall that in Lemma 10, we have:

$$\sum_{t=s\tau+1}^{(s+1)\tau}\left(\mathbb{E}\left[F(\boldsymbol{w}^{(t)}, \boldsymbol{\lambda})\right] - \mathbb{E}\left[F(\boldsymbol{w}^{(t)}, \boldsymbol{\lambda}^{(s)})\right]\right)$$

$$\leq -\frac{1}{2\gamma}\mathbb{E}\|\boldsymbol{\lambda}^{(s+1)} - \boldsymbol{\lambda}\|^2 + \frac{1}{2\gamma}\mathbb{E}[\|\boldsymbol{\lambda}^{(s)} - \boldsymbol{\lambda}\|^2] + \frac{1}{2\gamma}\mathbb{E}[\|\bar{\Delta}_s - \Delta_s\|^2] + \frac{1}{2\gamma}\mathbb{E}[\|\bar{\Delta}_s\|^2]$$

$$+ \tau^2\gamma\left(G_w + \sqrt{G_w^2 + G_\lambda^2 + \sigma_\lambda^2}\right).$$

Adding $\sum_{t=s\tau+1}^{(s+1)\tau} \mathbb{E}\left[\Phi(\boldsymbol{w}^{(t)})\right]$ to both sides, and re-arranging the terms give:

$$\sum_{t=s\tau+1}^{(s+1)\tau} \left(\mathbb{E}\left[\Phi(\boldsymbol{w}^{(t)})\right] - \mathbb{E}\left[F(\boldsymbol{w}^{(t)}, \boldsymbol{\lambda}^{(s)})\right]\right)$$

$$\leq \sum_{t=s\tau+1}^{(s)\tau} \left(\mathbb{E}\left[\Phi(\boldsymbol{w}^{(t)})\right] - \mathbb{E}\left[F(\boldsymbol{w}^{(t)}, \boldsymbol{\lambda})\right]\right) - \frac{1}{2\gamma}\mathbb{E}\|\boldsymbol{\lambda}^{(s+1)} - \boldsymbol{\lambda}\|^2$$

$$+ \frac{1}{2\gamma}\mathbb{E}[\|\boldsymbol{\lambda}^{(s)} - \boldsymbol{\lambda}\|^2] + \frac{1}{2\gamma}\mathbb{E}[\|\bar{\Delta}_s - \Delta_s\|^2] + \frac{1}{2\gamma}\mathbb{E}[\|\bar{\Delta}_s\|^2] + \tau^2\gamma G_w\left(G_w + \sqrt{G_w^2 + G_\lambda^2 + \sigma_\lambda^2}\right).$$

Then, we follow the same procedure as in Lemma 9. Without loss of generality we assume $\sqrt{S}$ is an integer, so we can equally divide index 0 to $S-1$ into $\sqrt{S}$ groups. Then we examine one block by summing $s$ from $s = c\sqrt{S}$ to $(c+1)\sqrt{S} - 1$, and set $\boldsymbol{\lambda} = \boldsymbol{\lambda}^*(\boldsymbol{w}^{(c+1)\sqrt{S}\tau})$:

$$\sum_{s=c\sqrt{S}}^{(c+1)\sqrt{S}-1} \sum_{t=s\tau+1}^{(s+1)\tau} \left(\mathbb{E}\left[\Phi(\boldsymbol{w}^{(t)})\right] - \mathbb{E}\left[F(\boldsymbol{w}^{(t)}, \boldsymbol{\lambda}^{(s)})\right]\right)$$

$$\leq \sum_{s=c\sqrt{S}}^{(c+1)\sqrt{S}-1} \sum_{t=s\tau+1}^{(s+1)\tau} \left(\mathbb{E}\left[F(\boldsymbol{w}^{(t)}, \boldsymbol{\lambda}^*(\boldsymbol{w}^t))\right] - \mathbb{E}\left[F(\boldsymbol{w}^{(t)}, \boldsymbol{\lambda}^*(\boldsymbol{w}^{(c+1)\sqrt{S}\tau}))\right]\right)$$

$$+ \sqrt{S}\tau^2\gamma G_w(G_w + \sqrt{G_w^2 + G_\lambda^2 + \sigma_\lambda^2}) + \gamma\tau^2\frac{\sqrt{S}\sigma_\lambda^2}{2m} + \gamma\tau^2\frac{\sqrt{S}G_\lambda^2}{2}$$

$$+ \frac{1}{2\gamma}\sum_{s=c\sqrt{S}}^{(c+1)\sqrt{S}-1} \left(\mathbb{E}\left[\left\|\boldsymbol{\lambda}^*(\boldsymbol{w}^{(c+1)\sqrt{S}\tau}) - \boldsymbol{\lambda}^{(s)}\right\|^2\right] - \mathbb{E}\left[\left\|\boldsymbol{\lambda}^*(\boldsymbol{w}^{(c+1)\sqrt{S}\tau}) - \boldsymbol{\lambda}^{(s+1)}\right\|^2\right]\right)$$

Adding and subtracting $\mathbb{E}\left[F(\boldsymbol{w}^{(t)}, \boldsymbol{\lambda}^*(\boldsymbol{w}^t))\right] - \mathbb{E}\left[F(\boldsymbol{w}^{(t)}, \boldsymbol{\lambda}^*(\boldsymbol{w}^{(c+1)\sqrt{S}\tau}))\right]$ yields:

$$\sum_{s=c\sqrt{S}}^{(c+1)\sqrt{S}-1} \sum_{t=s\tau+1}^{(s+1)\tau} \left(\mathbb{E}\left[\Phi(\boldsymbol{w}^{(t)})\right] - \mathbb{E}\left[F(\boldsymbol{w}^{(t)}, \boldsymbol{\lambda}^{(s)})\right]\right)$$

$$\leq \sum_{s=c\sqrt{S}}^{(c+1)\sqrt{S}-1} \sum_{t=s\tau+1}^{(s+1)\tau} \left(\mathbb{E}\left[F(\boldsymbol{w}^{(t)}, \boldsymbol{\lambda}^*(\boldsymbol{w}^t))\right] - \mathbb{E}\left[F(\boldsymbol{w}^{((c+1)\sqrt{S}\tau)}, \boldsymbol{\lambda}^*(\boldsymbol{w}^t))\right]\right.$$

$$\left. + \mathbb{E}\left[F(\boldsymbol{w}^{((c+1)\sqrt{S}\tau)}, \boldsymbol{\lambda}^*(\boldsymbol{w}^{(c+1)\sqrt{S}\tau}))\right] - \mathbb{E}\left[F(\boldsymbol{w}^{(t)}, \boldsymbol{\lambda}^*(\boldsymbol{w}^{(c+1)\sqrt{S}\tau}))\right]\right)$$

$$+ \gamma\tau^2\frac{\sqrt{S}\sigma_\lambda^2}{2m} + \gamma\tau^2\frac{\sqrt{S}G_\lambda^2}{2} + \sqrt{S}\tau^2\gamma G_w(G_w + \sqrt{G_w^2 + G_\lambda^2 + \sigma_\lambda^2})$$

$$+ \frac{1}{2\gamma}\sum_{s=c\sqrt{S}+1}^{(c+1)\sqrt{S}} \left(\mathbb{E}\left[\left\|\boldsymbol{\lambda}^*(\boldsymbol{w}^{(c+1)\sqrt{S}\tau}) - \boldsymbol{\lambda}^{(s)}\right\|^2\right] - \mathbb{E}\left[\left\|\boldsymbol{\lambda}^*(\boldsymbol{w}^{(c+1)\sqrt{S}\tau}) - \boldsymbol{\lambda}^{(s+1)}\right\|^2\right]\right)$$

$$\leq \sum_{s=c\sqrt{S}}^{(c+1)\sqrt{S}-1} \sum_{t=s\tau+1}^{(s+1)\tau} (2\sqrt{S}\tau\eta G_w\sqrt{G_w^2 + \sigma_w^2}) + \gamma\tau^2\frac{\sqrt{S}\sigma_\lambda^2}{2m} + \gamma\tau^2\frac{\sqrt{S}G_\lambda^2}{2} + \frac{D_\Lambda^2}{2\gamma}$$

$$+ \sqrt{S}\tau^2\gamma G_w\left(G_w + \sqrt{G_w^2 + G_\lambda^2 + \sigma_\lambda^2}\right)$$

$$\leq 2S\tau^2\eta G_w\sqrt{G_w^2 + \sigma_w^2} + \gamma\tau^2\frac{\sqrt{S}\sigma_\lambda^2}{2m} + \gamma\tau^2\frac{\sqrt{S}G_\lambda^2}{2} + \frac{D_\Lambda^2}{2\gamma}$$

$$+ \sqrt{S}\tau^2\gamma G_w\left(G_w + \sqrt{G_w^2 + G_\lambda^2 + \sigma_\lambda^2}\right).$$

So we can conclude that:

$$\sum_{s=c\sqrt{S}}^{(c+1)\sqrt{S}-1} \sum_{t=s\tau+1}^{(s+1)\tau} \left( \mathbb{E}\left[\Phi(\boldsymbol{w}^{(t)})\right] - \mathbb{E}\left[F(\boldsymbol{w}^{(t)}, \boldsymbol{\lambda}^{(s)})\right] \right)$$

$$\leq 2S\tau^2 \eta G_w \sqrt{G_w^2 + \sigma_w^2} + \gamma\tau^2 \frac{\sqrt{S}\sigma_\lambda^2}{2m} + \gamma\tau^2 \frac{\sqrt{S}G_\lambda^2}{2} \frac{D_\Lambda^2}{2\gamma} + \sqrt{S}\tau^2 \gamma G_w \left( G_w + \sqrt{G_w^2 + G_\lambda^2 + \sigma_\lambda^2} \right)$$

Summing above inequality over $c$ from $0$ to $\sqrt{S}-1$, and dividing both sides by $T$ gives

$$\frac{1}{T} \sum_{s=0}^{S-1} \sum_{t=s\tau+1}^{(s+1)\tau} \left( \mathbb{E}\left[\Phi(\boldsymbol{w}^{(t)})\right] - \mathbb{E}\left[F(\boldsymbol{w}^{(t)}, \boldsymbol{\lambda}^{(s)})\right] \right)$$

$$\leq 2\sqrt{S}\tau\eta G_w \sqrt{G_w^2 + \sigma_w^2} + \gamma\tau \frac{\sigma_\lambda^2}{2m} + \gamma\tau \frac{G_\lambda^2}{2} + \frac{D_\Lambda^2}{2\sqrt{S}\tau\gamma} + \tau\gamma G_w \left( G_w + \sqrt{G_w^2 + G_\lambda^2 + \sigma_\lambda^2} \right),$$

which concludes the proof. $\qquad\square$

**Proof of Theorem 4.** Now we proceed to the formal proof of Theorem 4. Re-arranging terms in Lemma 7, summing over $t = 1$ to $T$, and dividing by $T$ yields:

$$\frac{1}{T} \sum_{t=1}^{T} \mathbb{E}\left[ \left\| \nabla \Phi_{1/2L}\left(\boldsymbol{w}^{(t)}\right) \right\|^2 \right]$$

$$\leq \frac{4}{\eta T} \mathbb{E}[\Phi_{1/2L}(\boldsymbol{w}^{(0)})] + \frac{1}{2T} \sum_{t=1}^{T} D_{\mathcal{W}} L^2 \mathbb{E}\left[ \frac{1}{m} \sum_{i \in \mathcal{D}^{(\lfloor \frac{t}{\tau} \rfloor)}} \left\| \boldsymbol{w}_i^{(t)} - \boldsymbol{w}^{(t)} \right\| \right]$$

$$+ L\frac{1}{2T} \sum_{t=1}^{T} \left( \mathbb{E}[\Phi(\boldsymbol{w}^{(t)})] - \mathbb{E}[F(\boldsymbol{w}^{(t)}, \boldsymbol{\lambda}^{\lfloor \frac{t}{\tau} \rfloor})] \right).$$

Plugging in Lemmas 5 and 11 yields:

$$\frac{1}{T} \sum_{t=1}^{T} \mathbb{E}\left[ \left\| \nabla \Phi_{1/2L}(\boldsymbol{w}^{(t)}) \right\|^2 \right]$$

$$\leq \frac{4}{\eta T} \mathbb{E}[\Phi_{1/2L}(\boldsymbol{w}^{(0)})] + \eta\tau D_{\mathcal{W}} L^2 \left( \sigma_w + \frac{\sigma_w}{m} + \sqrt{\Gamma} \right).$$

$$+ \frac{L}{2} \left( 2\sqrt{S}\tau\eta G_w \sqrt{G_w^2 + \sigma_w^2} + \gamma\tau \frac{\sigma_\lambda^2}{2m} + \gamma\tau \frac{G_\lambda^2}{2} + \frac{D_\Lambda^2}{2\sqrt{S}\tau} + \tau\gamma G_w(G_w + \sqrt{G_w^2 + G_\lambda^2 + \sigma_\lambda^2}) \right)$$

$$\leq \frac{4}{\eta T} \mathbb{E}[\Phi_{1/2L}(\boldsymbol{w}^{(0)})] + \eta\tau D_{\mathcal{W}} L^2 \left( \sigma_w + \frac{\sigma_w}{m} + \sqrt{\Gamma} \right)$$

$$+ \sqrt{S}\tau\eta G_w L \sqrt{G_w^2 + \sigma_w^2} + \gamma\tau \frac{\sigma_\lambda^2 L}{4m} + \gamma\tau \frac{G_\lambda^2 L}{4} + \frac{D_\Lambda^2 L}{4\sqrt{S}\gamma\tau} + \frac{\tau\gamma L G_w(G_w + \sqrt{G_w^2 + G_\lambda^2 + \sigma_\lambda^2})}{2}.$$

Plugging in $\eta = \frac{1}{4LT^{3/4}}$, $\gamma = \frac{1}{T^{1/2}}$ and $\tau = T^{1/4}$ we recover the stated convergence rate as:

$$\frac{1}{T} \sum_{t=1}^{T} \mathbb{E}\left[ \left\| \nabla \Phi_{1/2L}(\boldsymbol{w}^{(t)}) \right\|^2 \right]$$

$$\leq \frac{4}{T^{1/4}} \mathbb{E}[\Phi_{1/2L}(\boldsymbol{w}^{(0)})] + \frac{L^2}{T^{1/2}} \left( \sigma_w + \frac{\sigma_w}{m} + \sqrt{\Gamma} \right)$$

$$+ \frac{1}{T^{1/8}} G_w L \sqrt{G_w^2 + \sigma_w^2} + \frac{\sigma_\lambda^2 L}{4mT^{1/4}} + \frac{G_\lambda^2 L}{4T^{1/4}} + \frac{D_\Lambda^2 L}{4T^{1/8}} + \frac{L G_w(G_w + \sqrt{G_w^2 + G_\lambda^2 + \sigma_\lambda^2})}{2T^{1/4}}.$$

$\qquad\square$

# F Proof of Convergence of **DRFA-GA** in Strongly-Convex-Strongly-Concave Setting

In this section we proceed to the proof of the convergence in strongly-convex-strongly-concave setting (Theorem 5). In this section we abuse the notation and use the following definition for $\bar{u}_t$:

$$\bar{u}_t = \sum_{i=1}^{N} \lambda_i^{(\lfloor \frac{t}{\tau} \rfloor)} \nabla f_i(\boldsymbol{w}_i^{(t)}).$$

## F.1 Overview of the Proof

We again start with the dynamic of one iteration:

$$\mathbb{E} \left\| \boldsymbol{w}^{(t+1)} - \boldsymbol{w}^* \right\|^2 \leq \left( 1 - \frac{\mu\eta}{2} \right) \mathbb{E} \left\| \boldsymbol{w}^{(t)} - \boldsymbol{w}^* \right\|^2 - \eta \mathbb{E} \left[ \Phi(\boldsymbol{w}^{(t)}) - \Phi(\boldsymbol{w}^*) \right]$$
$$+ \eta^2 \frac{2\sigma_w^2 + 4G_w^2}{m} 4L^2 \left( \eta^2 + \frac{\eta}{\mu} \right) \mathbb{E} \left[ \delta^{(t)} \right]$$
$$+ 4 \left( \frac{\eta}{\mu} + \eta^2 \right) \mathbb{E} \| \nabla_{\boldsymbol{w}} F(\boldsymbol{w}^{(t)}, \boldsymbol{\lambda}^{(\lfloor \frac{t}{\tau} \rfloor)}) - \nabla \Phi(\boldsymbol{w}^{(t)}) \|^2.$$

In addition to the local-global deviation, in this case we also have a new term $\| \nabla_{\boldsymbol{w}} F(\boldsymbol{w}^{(t)}, \boldsymbol{\lambda}^{(\lfloor \frac{t}{\tau} \rfloor)}) - \nabla \Phi(\boldsymbol{w}^{(t)}) \|^2$. Recall that $\nabla \Phi(\boldsymbol{w}^{(t)})$ is the gradient evaluated at $\boldsymbol{\lambda}^*(\boldsymbol{w}^{(t)})$. A straightforward approach is to use the smoothness of $\Phi$, to convert the difference between gradient to the difference between $\boldsymbol{\lambda}^{(\lfloor \frac{t}{\tau} \rfloor)}$ and $\boldsymbol{\lambda}^*(\boldsymbol{w}^{(t)})$. By examining the dynamic of $\boldsymbol{\lambda}$, we can prove that:

$$\mathbb{E} \left\| \boldsymbol{\lambda}^*(\boldsymbol{w}^{(t)}) - \boldsymbol{\lambda}^{(\lfloor \frac{t}{\tau} \rfloor)} \right\|^2 \leq 2 \left( 1 - \frac{1}{2\kappa} \right)^{(\lfloor \frac{t}{\tau} \rfloor)} \mathbb{E} \left\| \boldsymbol{\lambda}^{(0)} - \boldsymbol{\lambda}^*(\boldsymbol{w}^{(0)}) \right\|^2 + 2(4\kappa^2 + 1)\kappa^2 \tau^2 \eta^2 G_w^2.$$

Putting these pieces together, and unrolling the recursion will conclude the proof.

## F.2 Proof of Technical Lemmas

**Lemma 12** ( Lin et al. [29]. Properties of $\Phi(\cdot)$ and $\boldsymbol{\lambda}^*(\cdot)$). *If $F(\cdot, \boldsymbol{\lambda})$ is L-smooth function and $F(\boldsymbol{w}, \cdot)$ is $\mu$-strongly-concave, L-smooth function, let $\kappa = \frac{L}{\mu}$, then $\Phi(\boldsymbol{w})$ is $\alpha$-smooth function where $\alpha = L + \kappa L$ and $\boldsymbol{\lambda}^*(\boldsymbol{w})$ is $\kappa$-Lipschitz. Also $\nabla \Phi(\boldsymbol{w}) = \nabla_{\boldsymbol{w}} F(\boldsymbol{w}, \boldsymbol{\lambda}^*(\boldsymbol{w}))$.*

**Lemma 13.** *For DRFA-GA, under Theorem 5's assumptions, the following holds true:*

$$\mathbb{E} \left\| \boldsymbol{w}^{(t+1)} - \boldsymbol{w}^* \right\|^2 \leq \left( 1 - \frac{\mu\eta}{2} \right) \mathbb{E} \left\| \boldsymbol{w}^{(t)} - \boldsymbol{w}^* \right\|^2 - \eta \mathbb{E} \left[ \Phi(\boldsymbol{w}^{(t)}) - \Phi(\boldsymbol{w}^*) \right]$$
$$+ \eta^2 \frac{2\sigma_w^2 + 4G_w^2}{m} 4L^2 \left( \eta^2 + \frac{\eta}{\mu} \right) \mathbb{E} \left[ \delta^{(t)} \right] \qquad (31)$$
$$+ 4 \left( \frac{\eta}{\mu} + \eta^2 \right) \mathbb{E} \| \nabla_{\boldsymbol{w}} F(\boldsymbol{w}^{(t)}, \boldsymbol{\lambda}^{(\lfloor \frac{t}{\tau} \rfloor)}) - \nabla \Phi(\boldsymbol{w}^{(t)}) \|^2.$$

*Proof.* According to Lemma B2 in [30], if $F(\cdot, \boldsymbol{\lambda})$ is $\mu$-strongly-convex, then $\Phi(\cdot)$ is also $\mu$-strongly-convex. Noting this, from the strong convexity and the updating rule we have:

$$\mathbb{E} \| \boldsymbol{w}^{(t+1)} - \boldsymbol{w}^* \|^2$$
$$= \mathbb{E} \left\| \prod_{\mathcal{W}} \left( \boldsymbol{w}^{(t)} - \eta \boldsymbol{u}^{(t)} \right) - \boldsymbol{w}^* \right\|^2 \leq \mathbb{E} \| \boldsymbol{w}^{(t)} - \eta \bar{\boldsymbol{u}}^{(t)} - \boldsymbol{w}^* \|^2 + \eta^2 \mathbb{E} \| \bar{\boldsymbol{u}}^{(t)} - \boldsymbol{u}^{(t)} \|^2$$
$$= \mathbb{E} \| \boldsymbol{w}^{(t)} - \boldsymbol{w}^* \|^2 + \underbrace{\mathbb{E}[-2\eta \langle \bar{\boldsymbol{u}}^{(t)}, \boldsymbol{w}^{(t)} - \boldsymbol{w}^* \rangle]}_{T_1} + \underbrace{\eta^2 \mathbb{E} \| \bar{\boldsymbol{u}}^{(t)} \|^2}_{T_2} + \eta^2 \mathbb{E} \| \bar{\boldsymbol{u}}^{(t)} - \boldsymbol{u}^{(t)} \|^2$$
$$\qquad (32)$$

First we are to bound the variance $\mathbb{E}\|\bar{\boldsymbol{u}}^{(t)} - \boldsymbol{u}^{(t)}\|^2$:

$$\mathbb{E}\|\bar{\boldsymbol{u}}^{(t)} - \boldsymbol{u}^{(t)}\|^2 = \mathbb{E}\left\|\frac{1}{m}\sum_{i\in\mathcal{D}^{(\lfloor\frac{t}{\tau}\rfloor)}}\nabla f_i(\boldsymbol{w}_i^{(t)}) - \bar{\boldsymbol{u}}^{(t)}\right\|^2$$

$$= \mathbb{E}\left\|\frac{1}{m}\sum_{i\in\mathcal{D}^{(\lfloor\frac{t}{\tau}\rfloor)}}\nabla f_i(\boldsymbol{w}_i^{(t)};\xi_i^{(t)}) - \frac{1}{m}\sum_{i\in\mathcal{D}^{(\lfloor\frac{t}{\tau}\rfloor)}}\bar{\boldsymbol{u}}^{(t)}\right\|^2 \leq \frac{2\sigma_w^2 + 4G_w^2}{m},$$

where we use the fact $Var(\sum_{i=1}^m \boldsymbol{X}_i) = \sum_{i=1}^m Var(\boldsymbol{X}_i)$ for independent variables $\boldsymbol{X}_i, i = 1, \ldots, m$, and $Var(\nabla f_i(\boldsymbol{w}_i^{(t)};\xi_i^{(t)})) = \mathbb{E}\left\|\nabla f_i(\boldsymbol{w}_i^{(t)};\xi_i^{(t)}) - \bar{\boldsymbol{u}}^{(t)}\right\|^2 \leq 2\left\|\nabla f_i(\boldsymbol{w}_i^{(t)};\xi_i^{(t)}) - \nabla f_i(\boldsymbol{w}_i^{(t)})\right\|^2 + 2\left\|\nabla f_i(\boldsymbol{w}_i^{(t)}) - \bar{\boldsymbol{u}}^{(t)}\right\|^2 \leq 2\sigma_w^2 + 4G_w^2.$

Then we switch to bound $T_1$:

$$T_1 = 2\eta\mathbb{E}\left[-\left\langle\nabla\Phi(\boldsymbol{w}^{(t)}), \boldsymbol{w}^{(t)} - \boldsymbol{w}^*\right\rangle + \left\langle\nabla\Phi(\boldsymbol{w}^{(t)}) - \bar{\boldsymbol{u}}^{(t)}, \boldsymbol{w}^{(t)} - \boldsymbol{w}^*\right\rangle\right]$$

$$\leq 2\eta\mathbb{E}\left[-(\Phi(\boldsymbol{w}^{(t)}) - \Phi(\boldsymbol{w}^*)) - \frac{\mu}{2}\|\boldsymbol{w}^{(t)} - \boldsymbol{w}^*\|^2 + \frac{1}{\mu}\|\nabla\Phi(\boldsymbol{w}^{(t)}) - \bar{\boldsymbol{u}}^{(t)}\|^2 + \frac{\mu}{4}\|\boldsymbol{w}^{(t)} - \boldsymbol{w}^*\|^2\right]$$

$$\leq \mathbb{E}\left[-2\eta(\Phi(\boldsymbol{w}^{(t)}) - \Phi(\boldsymbol{w}^*)) - \frac{\mu\eta}{2}\|\boldsymbol{w}^{(t)} - \boldsymbol{w}^*\|^2 + \frac{2\eta}{\mu}\|\nabla\Phi(\boldsymbol{w}^{(t)}) - \bar{\boldsymbol{u}}^{(t)}\|^2\right]$$

$$\leq \mathbb{E}\left[-2\eta(\Phi(\boldsymbol{w}^{(t)}) - \Phi(\boldsymbol{w}^*)) - \frac{\mu\eta}{2}\|\boldsymbol{w}^{(t)} - \boldsymbol{w}^*\|^2 + \frac{4\eta}{\mu}\left\|\nabla\Phi(\boldsymbol{w}^{(t)}) - \nabla_{\boldsymbol{w}}F(\boldsymbol{w}^{(t)}, \boldsymbol{\lambda}^{(\lfloor\frac{t}{\tau}\rfloor)})\right\|^2\right.$$

$$\left.+\frac{4\eta}{\mu}\|\nabla_{\boldsymbol{w}}F(\boldsymbol{w}^{(t)}, \boldsymbol{\lambda}^{(\lfloor\frac{t}{\tau}\rfloor)}) - \bar{\boldsymbol{u}}^{(t)}\|^2\right]$$

$$\leq \mathbb{E}\left[-2\eta(\Phi(\boldsymbol{w}^{(t)}) - \Phi(\boldsymbol{w}^*)) - \frac{\mu\eta}{2}\|\boldsymbol{w}^{(t)} - \boldsymbol{w}^*\|^2 + \frac{4\eta}{\mu}\left\|\nabla\Phi(\boldsymbol{w}^{(t)}) - \nabla_{\boldsymbol{w}}F(\boldsymbol{w}^{(t)}, \boldsymbol{\lambda}^{(\lfloor\frac{t}{\tau}\rfloor)})\right\|^2\right.$$

$$\left.+\frac{4L^2\eta}{\mu}\sum_{k=1}^K \lambda_i^{(\lfloor\frac{t}{\tau}\rfloor)}\|\boldsymbol{w}^{(t)} - \boldsymbol{w}_i^{(t)}\|^2\right],$$

where in the second step we use the arithmetic and geometric inequality and the strong convexity of $\Phi(\cdot)$; and at the last step we use the smoothness, the convexity of $\|\cdot\|^2$ and Jensen's inequality.

Then, we can bound $T_2$ as:

$$T_2 \leq \eta^2\mathbb{E}\left[4\left\|\bar{\boldsymbol{u}}^{(t)} - \nabla_{\boldsymbol{w}}F(\boldsymbol{w}^{(t)}, \boldsymbol{\lambda}^{(\lfloor\frac{t}{\tau}\rfloor)})\right\|^2 + 4\|\nabla_{\boldsymbol{w}}F(\boldsymbol{w}^{(t)}, \boldsymbol{\lambda}^{(\lfloor\frac{t}{\tau}\rfloor)}) - \nabla\Phi(\boldsymbol{w}^{(t)})\|^2\right.$$

$$\left.+2\left\|\nabla\Phi(\boldsymbol{w}^{(t)})\right\|^2\right]$$

$$\leq \eta^2\mathbb{E}\left[4\left\|\bar{\boldsymbol{u}}^{(t)} - \nabla_{\boldsymbol{w}}F(\boldsymbol{w}^{(t)}, \boldsymbol{\lambda}^{(\lfloor\frac{t}{\tau}\rfloor)})\right\|^2 + 4\left\|\nabla_{\boldsymbol{w}}F(\boldsymbol{w}^{(t)}, \boldsymbol{\lambda}^{(\lfloor\frac{t}{\tau}\rfloor)}) - \nabla\Phi(\boldsymbol{w}^{(t)})\right\|^2\right.$$

$$\left.+4\alpha(\Phi(\boldsymbol{w}^{(t)}) - \Phi(\boldsymbol{w}^*))\right]$$

$$\leq \eta^2\mathbb{E}\left[4L^2\sum_{i=1}^N \lambda_i^{(\lfloor\frac{t}{\tau}\rfloor)}\|\boldsymbol{w}^{(t)} - \boldsymbol{w}_i^{(t)}\|^2 + 4\|\nabla_{\boldsymbol{w}}F(\boldsymbol{w}^{(t)}, \boldsymbol{\lambda}^{(\lfloor\frac{t}{\tau}\rfloor)}) - \nabla\Phi(\boldsymbol{w}^{(t)})\|^2\right.$$

$$\left.+4\alpha(\Phi(\boldsymbol{w}_t) - \Phi(\boldsymbol{w}^*))\right]$$

$$\leq \eta^2\mathbb{E}\left[4L^2\frac{1}{m}\sum_{i\in\mathcal{D}^{(\lfloor\frac{t}{\tau}\rfloor)}}\|\boldsymbol{w}^{(t)} - \boldsymbol{w}_i^{(t)}\|^2 + 4\|\nabla_{\boldsymbol{w}}F(\boldsymbol{w}^{(t)}, \boldsymbol{\lambda}^{(\lfloor\frac{t}{\tau}\rfloor)}) - \nabla\Phi(\boldsymbol{w}^{(t)})\|^2\right.$$

$$\left.+4\alpha(\Phi(\boldsymbol{w}_t) - \Phi(\boldsymbol{w}^*))\right].$$

Plugging $T_1$ and $T_2$ back to (32) results in:

$$\mathbb{E}\left\|\boldsymbol{w}^{(t+1)} - \boldsymbol{w}^*\right\|^2 \leq \left(1 - \frac{\mu\eta}{2}\right)\mathbb{E}\left\|\boldsymbol{w}^{(t)} - \boldsymbol{w}^*\right\|^2 + (4\alpha\eta^2 - 2\eta)\mathbb{E}\left[\Phi(\boldsymbol{w}^{(t)}) - \Phi(\boldsymbol{w}^*)\right]$$

$$+ \eta^2\frac{2\sigma_w^2 + 4G_w^2}{m} + 4L^2\left(\eta^2 + \frac{\eta}{\mu}\right)\mathbb{E}\left[\delta^{(t)}\right]$$

$$+ 4\left(\frac{\eta}{\mu} + \eta^2\right)\mathbb{E}\|\nabla_{\boldsymbol{w}}F(\boldsymbol{w}^{(t)}, \boldsymbol{\lambda}^{(\lfloor\frac{t}{\tau}\rfloor)}) - \nabla\Phi(\boldsymbol{w}^{(t)})\|^2. \tag{33}$$

By choosing $\eta < \frac{1}{4\alpha}$, it holds that $(4\alpha\eta^2 - 2\eta) < -\eta$, therefore we conclude the proof. $\qquad\square$

**Lemma 14** (Decreasing Optimal Gap of $\boldsymbol{\lambda}$). *For DRFA-GA, if $F(\boldsymbol{w}, \cdot)$ is $\mu$-strongly-concave, choosing $\gamma = \frac{1}{L}$, the optimality gap of $\boldsymbol{\lambda}$ is decreasing by the following recursive relation:*

$$\mathbb{E}\left\|\boldsymbol{\lambda}^*(\boldsymbol{w}^{(t)}) - \boldsymbol{\lambda}^{(\lfloor\frac{t}{\tau}\rfloor)}\right\|^2 \leq 2\left(1 - \frac{1}{2\kappa}\right)^{\lfloor\frac{t}{\tau}\rfloor}\mathbb{E}\left\|\boldsymbol{\lambda}^{(0)} - \boldsymbol{\lambda}^*(\boldsymbol{w}^{(0)})\right\|^2 + 2(4\kappa^2 + 1)\kappa^2\tau^2\eta^2 G_w^2.$$

*Proof:* Assume $s\tau + 1 \leq t \leq (s+1)\tau$. By the Jensen's inequality:

$$\mathbb{E}\|\boldsymbol{\lambda}^*(\boldsymbol{w}^{(t)}) - \boldsymbol{\lambda}^{(\lfloor\frac{t}{\tau}\rfloor)}\|^2 \leq 2\mathbb{E}\|\boldsymbol{\lambda}^*(\boldsymbol{w}^{(t)}) - \boldsymbol{\lambda}^*(\boldsymbol{w}^{(s\tau)})\|^2 + 2\mathbb{E}\|\boldsymbol{\lambda}^*(\boldsymbol{w}^{(s\tau)}) - \boldsymbol{\lambda}^{(s)}\|^2.$$

Firstly we are going to bound $\mathbb{E}\|\boldsymbol{\lambda}^*(\boldsymbol{w}^{(t)}) - \boldsymbol{\lambda}^*(\boldsymbol{w}^{(s\tau)})\|^2$. We use the $\kappa$-Lipschitz property of $\boldsymbol{\lambda}^*(\cdot)$:

$$\mathbb{E}\left\|\boldsymbol{\lambda}^*(\boldsymbol{w}^{(t)}) - \boldsymbol{\lambda}^*\left(\boldsymbol{w}^{(s\tau)}\right)\right\|^2 \leq \kappa^2\mathbb{E}\|\boldsymbol{w}^{(t)} - \boldsymbol{w}^{(s\tau)}\|^2 \leq \kappa^2\tau^2\eta^2 G_w^2.$$

Then we switch to bound $\mathbb{E}\|\boldsymbol{\lambda}^{(s)} - \boldsymbol{\lambda}^*(\boldsymbol{w}^{(s\tau)})\|^2$. We apply the Jensen's inequality first to get:

$$\mathbb{E}\left\|\boldsymbol{\lambda}^{(s)} - \boldsymbol{\lambda}^*(\boldsymbol{w}^{(s\tau)})\right\|^2 \leq \left(1 + \frac{1}{2(\kappa - 1)}\right)\mathbb{E}\left\|\boldsymbol{\lambda}^{(s)} - \boldsymbol{\lambda}^*\left(\boldsymbol{w}^{((s-1)\tau)}\right)\right\|^2$$

$$+ (1 + 2(\kappa - 1))\mathbb{E}\left\|\boldsymbol{\lambda}^*\left(\boldsymbol{w}^{((s-1)\tau)}\right) - \boldsymbol{\lambda}^*\left(\boldsymbol{w}^{(s\tau)}\right)\right\|^2$$

$$\leq \left(1 + \frac{1}{2(\kappa - 1)}\right)\mathbb{E}\left\|\boldsymbol{\lambda}^{(s)} - \boldsymbol{\lambda}^*\left(\boldsymbol{w}^{((s-1)\tau)}\right)\right\|^2 + 2\kappa^3\tau^2\eta^2 G_w^2, \quad (34)$$

where we use the fact that $\boldsymbol{\lambda}^*(\cdot)$ is $\kappa$-Lipschitz.

To bound $\mathbb{E}\left\|\boldsymbol{\lambda}^{(s)} - \boldsymbol{\lambda}^*\left(\boldsymbol{w}^{((s-1)\tau)}\right)\right\|^2$, by the updating rule of $\boldsymbol{\lambda}$ and the $\mu$-strongly-concavity of $F(\boldsymbol{w}, \cdot)$ we have:

$$\mathbb{E}\left\|\boldsymbol{\lambda}^{(s)} - \boldsymbol{\lambda}^*\left(\boldsymbol{w}^{((s-1)\tau)}\right)\right\|^2$$

$$\leq \mathbb{E}\left\|\boldsymbol{\lambda}^{(s-1)} - \boldsymbol{\lambda}^*\left(\boldsymbol{w}^{((s-1)\tau)}\right)\right\|^2 + \gamma^2\left\|\nabla_{\boldsymbol{\lambda}}F\left(\boldsymbol{w}^{((s-1)\tau)}, \boldsymbol{\lambda}^{(s-1)}\right)\right\|^2$$

$$+ 2\gamma\left\langle\nabla_{\boldsymbol{\lambda}}F\left(\boldsymbol{w}^{((s-1)\tau)}, \boldsymbol{\lambda}^{(s-1)}\right), \boldsymbol{\lambda}^{(s-1)} - \boldsymbol{\lambda}^*\left(\boldsymbol{w}^{((s-1)\tau)}\right)\right\rangle$$

$$\leq (1 - \mu\gamma)\mathbb{E}\left\|\boldsymbol{\lambda}^{(s-1)} - \boldsymbol{\lambda}^*\left(\boldsymbol{w}^{((s-1)\tau)}\right)\right\|^2$$

$$+ \underbrace{(2\gamma^2 L - 2\gamma)}_{\leq 0}\left[F\left(\boldsymbol{w}^{((s-1)\tau)}, \boldsymbol{\lambda}^*\left(\boldsymbol{w}^{((s-1)\tau)}\right)\right) - F\left(\boldsymbol{w}^{((s-1)\tau)}, \boldsymbol{\lambda}^{(s-1)}\right)\right]$$

$$\leq \left(1 - \frac{1}{\kappa}\right)\mathbb{E}\left\|\boldsymbol{\lambda}^{(s-1)} - \boldsymbol{\lambda}^*\left(\boldsymbol{w}^{((s-1)\tau)}\right)\right\|^2, \tag{35}$$

where we used the smoothness property of $F(\boldsymbol{w}, \cdot)$:

$$\left\|\nabla_{\boldsymbol{\lambda}}F\left(\boldsymbol{w}^{((s-1)\tau)}, \boldsymbol{\lambda}^{(s-1)}\right)\right\|^2 \leq 2L\left(F\left(\boldsymbol{w}^{((s-1)\tau)}, \boldsymbol{\lambda}^*\left(\boldsymbol{w}^{((s-1)\tau)}\right)\right) - F\left(\boldsymbol{w}^{((s-1)\tau)}, \boldsymbol{\lambda}^{(s-1)}\right)\right).$$

Plugging (35) into (34) yields:

$$\mathbb{E}\left\|\boldsymbol{\lambda}^{(s)} - \boldsymbol{\lambda}^*(\boldsymbol{w}^{(s\tau)})\right\|^2$$

$$\leq \left(1 + \frac{1}{2(\kappa-1)}\right)\left(1 - \frac{1}{\kappa}\right)\mathbb{E}\left\|\boldsymbol{\lambda}^{(s-1)} - \boldsymbol{\lambda}^*\left(\boldsymbol{w}^{((s-1)\tau)}\right)\right\|^2 + 2\kappa^3\tau^2\eta^2 G_w^2$$

$$\leq \left(1 - \frac{1}{2\kappa}\right)\mathbb{E}\left\|\boldsymbol{\lambda}^{(s-1)} - \boldsymbol{\lambda}^*\left(\boldsymbol{w}^{((s-1)\tau)}\right)\right\|^2 + 2\kappa^3\tau^2\eta^2 G_w^2.$$

Applying the recursion on the above relation gives:

$$\mathbb{E}\|\boldsymbol{\lambda}^{(s)} - \boldsymbol{\lambda}^*(\boldsymbol{w}^{(s\tau)})\|^2 \leq \left(1 - \frac{1}{2\kappa}\right)^s \mathbb{E}\left\|\boldsymbol{\lambda}^0 - \boldsymbol{\lambda}^*(\boldsymbol{w}^{(0)})\right\|^2 + 4\kappa^4\tau^2\eta^2 G_w^2.$$

Putting these pieces together concludes the proof:

$$\mathbb{E}\left\|\boldsymbol{\lambda}^*(\boldsymbol{w}^{(t)}) - \boldsymbol{\lambda}^{(\lfloor\frac{t}{\tau}\rfloor)}\right\|^2 \leq 2\left(1 - \frac{1}{2\kappa}\right)^{\lfloor\frac{t}{\tau}\rfloor}\mathbb{E}\left\|\boldsymbol{\lambda}_0 - \boldsymbol{\lambda}^*(\boldsymbol{w}^{(0)})\right\|^2 + 2(4\kappa^2+1)\kappa^2\tau^2\eta^2 G_w^2.$$

$\square$

**Lemma 15.** *For $\eta\mu \leq 1$, $\kappa > 1$, $\tau \geq 1$, the following inequalities holds:*

$$\sum_{t=0}^{T}\left(1 - \frac{1}{2}\eta\mu\right)^t\left(1 - \frac{1}{2\kappa}\right)^{\lfloor\frac{t}{\tau}\rfloor} \leq \frac{2\kappa\tau}{1 - \frac{1}{2}\eta\mu},$$

$$\sum_{t=0}^{T}\left(1 - \frac{1}{4}\eta\mu\right)^t\left(1 - \frac{1}{2\kappa}\right)^{\lfloor\frac{t}{\tau}\rfloor} \leq \frac{2\kappa\tau}{1 - \frac{1}{4}\eta\mu}.$$

*Proof.*

$$\sum_{t=0}^{T}(1 - \frac{1}{2}\eta\mu)^t(1 - \frac{1}{2\kappa})^{\lfloor\frac{t}{\tau}\rfloor} = \sum_{s=0}^{S-1}\sum_{t=1}^{\tau}(1 - \frac{1}{2}\eta\mu)^{s\tau+t}(1 - \frac{1}{2\kappa})^s$$

$$\leq \sum_{s=0}^{S-1}(1 - \frac{1}{2\kappa})^s\sum_{t=1}^{\tau}\left(1 - \frac{1}{2}\eta\mu\right)^{s\tau+t}$$

$$\leq 2\sum_{s=0}^{S-1}(1 - \frac{1}{2\kappa})^s\frac{\left(1 - \frac{1}{2}\eta\mu\right)^{s\tau}\left(1 - \left(1 - \frac{1}{2}\eta\mu\right)^{\tau}\right)}{\eta\mu}$$

$$= \frac{2(1 - \left(1 - \frac{1}{2}\eta\mu\right)^{\tau})}{\eta\mu}\sum_{s=0}^{S-1}\left(1 - \frac{1}{2\kappa}\right)^s\left(1 - \frac{1}{2}\eta\mu\right)^{s\tau}$$

$$\leq \frac{2(1 - \left(1 - \frac{1}{2}\eta\mu\right)^{\tau})}{\eta\mu}\sum_{s=0}^{S-1}\left(1 - \frac{1}{2\kappa}\right)^s\left(1 - \frac{1}{2}\eta\mu\right)^s \quad (36)$$

$$\leq \frac{2\tau\ln\frac{1}{\left(1-\frac{1}{2}\eta\mu\right)}}{\eta\mu}\frac{1}{1 - \left(1 - \frac{1}{2\kappa}\right)\left(1 - \frac{1}{2}\eta\mu\right)} \quad (37)$$

$$\leq \frac{2\tau\ln\frac{1}{\left(1-\frac{1}{2}\eta\mu\right)}}{\left(\frac{\eta\mu}{2\kappa} + (\frac{1}{2} - \frac{1}{4\kappa})\eta^2\mu^2\right)} \leq \frac{4\kappa\tau}{\eta\mu}\left(\frac{1}{1 - \frac{1}{2}\eta\mu} - 1\right) \quad (38)$$

$$\leq \frac{2\kappa\tau}{\eta\mu}\left(\frac{\eta\mu}{1 - \frac{1}{2}\eta\mu}\right) = \frac{2\kappa\tau}{1 - \frac{1}{2}\eta\mu}, \quad (39)$$

where from (36) to (37) we use the inequality $1 - a^x \leq x\ln\frac{1}{a}$, and from (38) to (39) we use the inequality $\ln x \leq x - 1$.

Similarly, for the second statement:

$$
\begin{aligned}
\sum_{t=0}^{T}(1-\frac{1}{4}\eta\mu)^t(1-\frac{1}{2\kappa})^{\lfloor\frac{t}{\tau}\rfloor} &= \sum_{s=0}^{S-1}\sum_{t=1}^{\tau}\left(1-\frac{1}{4}\eta\mu\right)^{s\tau+t}(1-\frac{1}{2\kappa})^s \\
&\leq \sum_{s=0}^{S-1}(1-\frac{1}{2\kappa})^s\sum_{t=1}^{\tau}\left(1-\frac{1}{4}\eta\mu\right)^{s\tau+t} \\
&\leq 2\sum_{s=0}^{S-1}(1-\frac{1}{2\kappa})^s\frac{\left(1-\frac{1}{4}\eta\mu\right)^{s\tau}\left(1-\left(1-\frac{1}{4}\eta\mu\right)^\tau\right)}{\eta\mu} \\
&= \frac{2(1-\left(1-\frac{1}{4}\eta\mu\right)^\tau)}{\eta\mu}\sum_{s=0}^{S-1}\left(1-\frac{1}{2\kappa}\right)^s\left(1-\frac{1}{4}\eta\mu\right)^{s\tau} \\
&\leq \frac{2(1-\left(1-\frac{1}{4}\eta\mu\right)^\tau)}{\eta\mu}\sum_{s=0}^{S-1}\left(1-\frac{1}{2\kappa}\right)^s\left(1-\frac{1}{4}\eta\mu\right)^s \\
&\leq \frac{2\tau\ln\frac{1}{\left(1-\frac{1}{4}\eta\mu\right)}}{\eta\mu}\frac{1}{1-\left(1-\frac{1}{2\kappa}\right)\left(1-\frac{1}{4}\eta\mu\right)} \\
&\leq \frac{2\tau\ln\frac{1}{\left(1-\frac{1}{4}\eta\mu\right)}}{\left(\frac{\eta\mu}{2\kappa}+(\frac{1}{4}-\frac{1}{8\kappa})\eta^2\mu^2\right)} \leq \frac{4\kappa\tau}{\eta\mu}\left(\frac{1}{1-\frac{1}{4}\eta\mu}-1\right) \\
&\leq \frac{2\kappa\tau}{\eta\mu}\left(\frac{\eta\mu}{1-\frac{1}{4}\eta\mu}\right)=\frac{2\kappa\tau}{1-\frac{1}{4}\eta\mu}.
\end{aligned}
$$

$\square$

## F.3  Proof of Theorem 5

Now we proceed to the proof of Theorem 5. According to Lemma 13 we have:

$$
\begin{aligned}
\mathbb{E}\left\|\boldsymbol{w}^{(t+1)}-\boldsymbol{w}^*\right\|^2 &\leq \left(1-\frac{\mu\eta}{2}\right)\mathbb{E}\left\|\boldsymbol{w}^{(t)}-\boldsymbol{w}^*\right\|^2-\eta\mathbb{E}\left[\Phi(\boldsymbol{w}^{(t)})-\Phi(\boldsymbol{w}^*)\right]+\eta^2\frac{2\sigma_w^2+4G_w^2}{m} \\
&\quad +4L^2\left(\eta^2+\frac{\eta}{\mu}\right)\mathbb{E}\left[\delta^{(t)}\right]+4\left(\frac{\eta}{\mu}+\eta^2\right)\mathbb{E}\left\|\nabla_{\boldsymbol{w}}F(\boldsymbol{w}^{(t)},\boldsymbol{\lambda}^{(\lfloor\frac{t}{\tau}\rfloor)})-\nabla\Phi(\boldsymbol{w}^{(t)})\right\|^2 \\
&\leq \left(1-\frac{\mu\eta}{2}\right)\mathbb{E}\left\|\boldsymbol{w}^{(t)}-\boldsymbol{w}^*\right\|^2-\eta\mathbb{E}\left[\Phi(\boldsymbol{w}^{(t)})-\Phi(\boldsymbol{w}^*)\right]+\eta^2\frac{2\sigma_w^2+4G_w^2}{m} \\
&\quad +4L^2\left(\eta^2+\frac{\eta}{\mu}\right)\mathbb{E}\left[\delta^{(t)}\right]+4\left(\frac{\eta}{\mu}+\eta^2\right)L^2\mathbb{E}\left\|\boldsymbol{\lambda}^*(\boldsymbol{w}^{(t)})-\boldsymbol{\lambda}^{(\lfloor\frac{t}{\tau}\rfloor)}\right\|^2,
\end{aligned}
$$

where we use the smoothness of $F$ at the last step to substitute $\|\nabla_{\boldsymbol{w}}F(\boldsymbol{w}^{(t)},\boldsymbol{\lambda}^*(\boldsymbol{w}^{(t)}))-\nabla_{\boldsymbol{w}}F(\boldsymbol{w}^{(t)},\boldsymbol{\lambda}^{(\lfloor\frac{t}{\tau}\rfloor)})\|^2$:

$$
\left\|\nabla_{\boldsymbol{w}}F(\boldsymbol{w}^{(t)},\boldsymbol{\lambda}^*(\boldsymbol{w}^{(t)}))-\nabla_{\boldsymbol{w}}F(\boldsymbol{w}^{(t)},\boldsymbol{\lambda}^{(\lfloor\frac{t}{\tau}\rfloor)})\right\|^2 \leq L^2\left\|\boldsymbol{\lambda}^*(\boldsymbol{w}^{(t)})-\boldsymbol{\lambda}^{(\lfloor\frac{t}{\tau}\rfloor)}\right\|^2.
$$

Then plugging in Lemma 14 yields:

$$
\begin{aligned}
\mathbb{E}\|\boldsymbol{w}^{(t+1)}-\boldsymbol{w}^*\|^2 &\leq \left(1-\frac{\mu\eta}{2}\right)\mathbb{E}\left\|\boldsymbol{w}^{(t)}-\boldsymbol{w}^*\right\|^2-\eta\mathbb{E}\left[\Phi(\boldsymbol{w}^{(t)})-\Phi(\boldsymbol{w}^*)\right]+\eta^2\frac{2\sigma_w^2+4G_w^2}{m} \\
&\quad +4L^2\left(\eta^2+\frac{\eta}{\mu}\right)\mathbb{E}\left[\delta^{(t)}\right] \\
&\quad +8\left(\frac{\eta}{\mu}+\eta^2\right)L^2\left(\left(1-\frac{1}{2\kappa}\right)^{\lfloor\frac{t}{\tau}\rfloor}\mathbb{E}\|\boldsymbol{\lambda}^{(0)}-\boldsymbol{\lambda}^*(\boldsymbol{w}^{(0)})\|^2+\kappa^2\tau^2\eta^2G_w^2\left(4\kappa^2+1\right)\right).
\end{aligned}
$$
(40)

Unrolling the recursion yields:

$$\mathbb{E}\|\boldsymbol{w}^{(T)} - \boldsymbol{w}^*\|^2$$

$$\leq \left(1 - \frac{1}{2}\mu\eta\right)^T \mathbb{E}\|\boldsymbol{w}^{(0)} - \boldsymbol{w}^*\|^2 + \sum_{t=1}^{T}\left(1 - \frac{1}{2}\mu\eta\right)^t \left[8L^2\kappa^2\tau^2\eta^2 G_w^2\left(\frac{\eta}{\mu} + \eta^2\right)(4\kappa^2 + 1)\right]$$

$$+ \sum_{t=1}^{T}\left(1 - \frac{1}{2}\mu\eta\right)^t\left[\eta^2\frac{2\sigma_w^2 + 4G_w^2}{m} + 4L^2\left(\eta^2 + \frac{\eta}{\mu}\right)\mathbb{E}\left[\delta^{(t)}\right]\right]$$

$$+ 8\left(\frac{\eta}{\mu} + \eta^2\right)L^2\mathbb{E}\|\boldsymbol{\lambda}^{(0)} - \boldsymbol{\lambda}^*(\boldsymbol{w}^{(0)})\|^2\sum_{t=1}^{T}\left(1 - \frac{1}{2}\mu\eta\right)^t\left(1 - \frac{1}{2\kappa}\right)^{\lfloor\frac{t}{\tau}\rfloor} \qquad (41)$$

$$\leq \exp\left(-\frac{1}{2}\mu\eta T\right)D_{\mathcal{W}}^2 + \eta\frac{4\sigma_w^2 + 8G_w^2}{\mu m} + 8L^2\left(\frac{\eta}{\mu} + \frac{1}{\mu^2}\right)\sum_{t=0}^{T}\mathbb{E}\left[\delta^{(t)}\right]$$

$$+ 16L^2\kappa^2\tau^2\eta^2 G_w^2\left(\frac{\eta}{\mu} + \frac{1}{\mu^2}\right)(4\kappa^2 + 1) + 16L^2\left(\frac{\kappa\tau}{1 - \frac{1}{2}\eta\mu}\right)\left(\frac{\eta}{\mu} + \eta^2\right)D_{\Lambda}^2, \qquad (42)$$

where we used the result from Lemma 15 from (41) to (42). Now, we simplify (40) by applying the telescoping sum on (40) for $t = \frac{T}{2}$ to $T$:

$$\frac{2}{T}\sum_{t=T/2}^{T}\mathbb{E}\left[\Phi(\boldsymbol{w}^{(t)}) - \Phi(\boldsymbol{w}^*)\right]$$

$$\leq \frac{2}{\eta T}\mathbb{E}\|\boldsymbol{w}^{(T/2)} - \boldsymbol{w}^*\|^2 + \eta\frac{2\sigma_w^2 + 4G_w^2}{m} + 4L^2\left(\eta + \frac{1}{\mu}\right)\frac{2}{T}\sum_{t=T/2}^{T}\mathbb{E}\left[\delta^{(t)}\right]$$

$$+ 8\left(\frac{1}{\mu} + \eta\right)L^2 D_{\Lambda}^2\frac{2}{T}\sum_{t=T/2}^{T}\left(1 - \frac{1}{2\kappa}\right)^{\lfloor\frac{t}{\tau}\rfloor} + 8\left(\frac{1}{\mu} + \eta\right)\kappa^2\tau^2\eta^2 L^2 G_w^2(4\kappa^2 + 1)$$

$$\leq \frac{2}{\eta T}\mathbb{E}\|\boldsymbol{w}^{(T/2)} - \boldsymbol{w}^*\|^2 + \eta\frac{2\sigma_w^2 + 4G_w^2}{m} + 80\eta^2\tau^2 L^2\left(\eta + \frac{1}{\mu}\right)\left(\sigma_w^2 + \frac{\sigma_w^2}{m} + \Gamma\right)$$

$$+ 16\left(\frac{1}{\mu} + \eta\right)L^2 O\left(\frac{\tau\exp(-\mu\eta T/4\tau)}{T}D_{\Lambda}^2\right) + 8\left(\frac{1}{\mu} + \eta\right)\kappa^2\tau^2\eta^2 L^2 G_w^2(4\kappa^2 + 1)$$

$$\leq \frac{2}{\eta T}\mathbb{E}\|\boldsymbol{w}^{(T/2)} - \boldsymbol{w}^*\|^2 + \eta\frac{2\sigma_w^2 + 4G_w^2}{m} + 80\eta^2\tau^2 L^2\left(\eta + \frac{1}{\mu}\right)\left(\sigma_w^2 + \frac{\sigma_w^2}{m} + \Gamma\right)$$

$$+ 16\left(\frac{1}{\mu} + \eta\right)L^2 O\left(\frac{\tau\exp(-\mu\eta T/4\tau)}{T}D_{\Lambda}^2\right) + 8\left(\frac{1}{\mu} + \eta\right)\kappa^2\tau^2\eta^2 L^2 G_w^2(4\kappa^2 + 1).$$

Plugging in (42) yields:

$$\frac{2}{T}\sum_{t=T/2}^{T}\mathbb{E}\left[\Phi(\boldsymbol{w}^{(t)}) - \Phi(\boldsymbol{w}^*)\right]$$

$$\leq \frac{2}{\eta T}\left(\exp\left(-\frac{1}{4}\mu\eta T\right)D_{\mathcal{W}}^2 + \eta\frac{4\sigma_w^2 + 8G_w^2}{\mu m} + 8L^2\left(\frac{\eta}{\mu} + \frac{1}{\mu^2}\right)\sum_{t=0}^{T}\mathbb{E}\left[\delta^{(t)}\right]\right)$$

$$+ \frac{2}{\eta T}\left(16L^2\kappa^2\tau^2\eta^2 G_w^2\left(\frac{\eta}{\mu} + \frac{1}{\mu^2}\right)(4\kappa^2 + 1) + 16L^2\left(\frac{\kappa\tau}{1 - \frac{1}{2}\eta\mu}\right)\left(\frac{\eta}{\mu} + \eta^2\right)D_{\Lambda}^2\right)$$

$$+ \eta\frac{2\sigma_w^2 + 4G_w^2}{m} + 80\eta^2\tau^2 L^2\left(\eta + \frac{1}{\mu}\right)\left(\sigma_w^2 + \frac{\sigma_w^2}{m} + \Gamma\right)$$

$$+ 16\left(\frac{1}{\mu} + \eta\right)L^2 O\left(\frac{\tau\exp(-\mu\eta T/4\tau)}{T}D_{\Lambda}^2\right) + 8\left(\frac{1}{\mu} + \eta\right)\kappa^2\tau^2\eta^2 L^2 G_w^2(4\kappa^2 + 1).$$

Combining the terms yields:

$$\frac{2}{T} \sum_{t=T/2}^{T} \mathbb{E}\left[\Phi(\boldsymbol{w}^{(t)}) - \Phi(\boldsymbol{w}^*)\right]$$

$$\leq \frac{2}{\eta T} \exp\left(-\frac{1}{4}\mu\eta T\right) D_{\mathcal{W}}^2 + 16\left(\frac{1}{\mu} + \eta\right) L^2 O\left(\frac{\tau \exp(-\mu\eta T/4\tau)}{T} D_{\Lambda}^2\right)$$

$$+ \left(\frac{4}{\mu T} + \eta\right) \frac{2\sigma_w^2 + 4G_w^2}{m} + \left(1 + \frac{2}{\mu\eta T}\right) 80\eta^2\tau^2 L^2 \left(\eta + \frac{1}{\mu}\right) \left(\sigma_w^2 + \frac{\sigma_w^2}{m} + \Gamma\right)$$

$$+ \left(\frac{4}{\mu\eta T} + 1\right) 8L^2\kappa^2\tau^2\eta^2 G_w^2 \left(4\kappa^2 + 1\right) \left(\eta + \frac{1}{\mu}\right)$$

$$+ \frac{32L^2}{T} \left(\frac{\kappa\tau}{1 - \frac{1}{2}\eta\mu}\right) \left(\frac{1}{\mu} + \eta\right) D_{\Lambda}^2.$$

And finally, plugging in $\eta = \frac{4\log T}{\mu T}$ and using the fact that $\Phi(\frac{2}{T}\sum_{t=T/2}^{T} \boldsymbol{w}^{(t)}) \leq \frac{2}{T}\sum_{t=T/2}^{T} \Phi(\boldsymbol{w}^{(t)})$ yields:

$$\mathbb{E}[\Phi(\hat{\boldsymbol{w}}) - \Phi(\boldsymbol{w}^*)]$$

$$\leq \frac{\mu D_{\mathcal{W}}^2}{2T\log T} + 16\left(\frac{1}{\mu} + \frac{4\log T}{\mu T}\right) L^2 O\left(\frac{\tau}{T^{(1+1/\tau)}} D_{\Lambda}^2\right)$$

$$+ \left(\frac{4}{\mu T} + \frac{4\log T}{\mu T}\right) \frac{2\sigma_w^2 + 4G_w^2}{m} + \left(1 + \frac{2}{\mu\eta T}\right) \frac{1280\kappa^2\tau^2\log^2 T}{T^2} \left(\eta + \frac{1}{\mu}\right) \left(\sigma_w^2 + \frac{\sigma_w^2}{m} + \Gamma\right)$$

$$+ \left(\frac{1}{\log T} + 1\right) \frac{8\kappa^4\tau^2\log^2 T}{T^2} G_w^2 \left(4\kappa^2 + 1\right) \left(\frac{4\log T}{\mu T} + \frac{1}{\mu}\right)$$

$$+ \frac{32L^2}{T} \left(\frac{\kappa\tau}{1 - \frac{2\log T}{T}}\right) \left(\frac{1}{\mu} + \frac{4\log T}{\mu T}\right) D_{\Lambda}^2$$

$$\leq \tilde{O}\left(\frac{\mu D_{\mathcal{W}}^2}{T}\right) + O\left(\frac{\kappa L\tau D_{\Lambda}^2}{T^{(1+1/\tau)}}\right) + \tilde{O}\left(\frac{\sigma_w^2 + G_w^2}{\mu m T}\right) + O\left(\frac{\kappa^2\tau^2(\sigma_w^2 + \Gamma)}{\mu T^2}\right)$$

$$+ \tilde{O}\left(\frac{\kappa^2 L\tau D_{\Lambda}^2}{T}\right) + \tilde{O}\left(\frac{\kappa^6\tau^2 G_w^2}{\mu T^2}\right).$$

$\square$

# G   Proof of Convergence of **DRFA-GA** in Nonconvex (PL Condition)-Strongly-Concave Setting

## G.1   Overview of Proofs

In this section we will present formal proofs in nonconvex (PL condition)-strongly-concave setting (Theorem 6). The main idea is similar to strongly-convex-strongly-concave case: we start from one iteration analysis, and plug in the upper bound of $\delta^{(t)}$ and $\|\nabla_{\boldsymbol{w}} F(\boldsymbol{w}^{(t)}, \boldsymbol{\lambda}^{(\lfloor \frac{t}{\tau}\rfloor)}) - \nabla\Phi(\boldsymbol{w}^{(t)})\|^2$.

However, a careful analysis need to be employed in order to deal with projected SGD in constrained nonconvex optimization problem. We employ the technique used in [10], where they advocate to study the following quantity:

$$P_{\mathcal{W}}(\boldsymbol{w}, \boldsymbol{g}, \eta) = \frac{1}{\eta}\left[\boldsymbol{w} - \prod_{\mathcal{W}}(\boldsymbol{w} - \eta\boldsymbol{g})\right].$$

If we plug in $\boldsymbol{w} = \boldsymbol{w}^{(t)}, \boldsymbol{g} = \boldsymbol{u}^{(t)} = \frac{1}{m}\sum_{i \in \mathcal{D}^{(\lfloor \frac{t}{\tau}\rfloor)}} \nabla f_i(\boldsymbol{w}_i^{(t)}; \xi_i^t)$, then

$$P_{\mathcal{W}}(\boldsymbol{w}^{(t)}, \boldsymbol{u}^{(t)}, \eta) = \frac{1}{\eta}\left[\boldsymbol{w}^{(t)} - \prod_{\mathcal{W}}\left(\boldsymbol{w}^{(t)} - \eta\boldsymbol{u}^{(t)}\right)\right].$$

characterize the difference between iterates $\boldsymbol{w}^{(t+1)}$ and $\boldsymbol{w}^{(t)}$. A trivial property of operator $P_{\mathcal{W}}$ is contraction mapping, which follows the property of projection:

$$\left\| P_{\mathcal{W}}(\boldsymbol{w}, \boldsymbol{g}_1, \eta) - P_{\mathcal{W}}(\boldsymbol{w}, \boldsymbol{g}_2, \eta) \right\|^2 \leq \left\| \boldsymbol{g}_1 - \boldsymbol{g}_2 \right\|^2 .$$

The significant property of operator $P_{\mathcal{W}}$ is given by the following lemma:

**Lemma 16** (Property of Projection, [10] Lemma 1). *For all $\boldsymbol{w} \in \mathcal{W} \subset \mathbb{R}^d$, $\boldsymbol{g} \in \mathbb{R}^d$ and $\eta > 0$, we have:*

$$\langle \boldsymbol{g}, P_{\mathcal{W}}(\boldsymbol{w}, \boldsymbol{g}, \eta) \rangle \geq \left\| P_{\mathcal{W}}(\boldsymbol{w}, \boldsymbol{g}, \eta) \right\|^2 .$$

The above lemma establishes a lower bound for the inner product $\langle \boldsymbol{g}, P_{\mathcal{W}}(\boldsymbol{y}, \boldsymbol{g}, \eta) \rangle$, and will play a significant role in our analysis.

## G.2 Proof of Technical Lemmas

**Lemma 17.** *If $F(\cdot, \boldsymbol{\lambda})$ satisfies $\mu$-generalized PL condition, then $\Phi(\cdot)$ also satisfies $\mu$-generalized PL condition.*

*Proof.* Let $\boldsymbol{w}^* \in \arg\min_{\boldsymbol{w} \in \mathcal{W}} \Phi(\boldsymbol{w})$. Since $F(\cdot, \boldsymbol{\lambda})$ satisfies $\mu$-generalized PL condition, we have for any $\boldsymbol{w} \in \mathcal{W}$:

$$\frac{1}{2\eta^2} \left\| \boldsymbol{w} - \prod_{\mathcal{W}} (\boldsymbol{w} - \eta \nabla_{\boldsymbol{w}} F(\boldsymbol{w}, \boldsymbol{\lambda}^*(\boldsymbol{w}))) \right\|^2 \geq \mu(F(\boldsymbol{w}, \boldsymbol{\lambda}^*(\boldsymbol{w})) - \min_{\boldsymbol{w}' \in \mathcal{W}} F(\boldsymbol{w}', \boldsymbol{\lambda}^*(\boldsymbol{w}))$$

$$\geq \mu(F(\boldsymbol{w}, \boldsymbol{\lambda}^*(\boldsymbol{w})) - F(\boldsymbol{w}^*, \boldsymbol{\lambda}^*(\boldsymbol{w}))$$

$$\geq \mu(F(\boldsymbol{w}, \boldsymbol{\lambda}^*(\boldsymbol{w})) - F(\boldsymbol{w}^*, \boldsymbol{\lambda}^*(\boldsymbol{w}^*))).$$

which immediately implies $\frac{1}{2\eta^2} \| \boldsymbol{w} - \prod_{\mathcal{W}} (\boldsymbol{w} - \eta \nabla \Phi(\boldsymbol{w})) \|^2 \geq \mu(\Phi(\boldsymbol{w}) - \Phi(\boldsymbol{w}^*))$ as desired. $\square$

**Lemma 18.** *For DRFA-GA, under Theorem 6's assumptions, we have:*

$$\mathbb{E}\left[\Phi(\boldsymbol{w}^{(t+1)}) - \Phi(\boldsymbol{w}^*)\right] \leq \left(1 - \frac{\mu\eta}{4}\right) \mathbb{E}\left[\Phi(\boldsymbol{w}^{(t)}) - \Phi(\boldsymbol{w}^*)\right]$$

$$+ \frac{3\eta}{2} \mathbb{E}\left\| \sum_{i=1}^{N} \lambda_i^{(\lfloor \frac{t}{\tau} \rfloor)} \nabla f_i(\boldsymbol{w}_i^{(t)}) - \nabla \Phi(\boldsymbol{w}^{(t)}) \right\|^2 + 3\eta \frac{2\sigma_w^2 + 4G_w^2}{2m},$$

(43)

*where $\alpha = L + \kappa L$*

*Proof.* Define the following quantities:

$$\boldsymbol{u}_t = \frac{1}{m} \sum_{i \in D^t} \nabla f_i(\boldsymbol{w}_i^{(t)}; \xi_i^t), \bar{\boldsymbol{u}}_t = \sum_{i=1}^{N} \lambda_i^{(\lfloor \frac{t}{\tau} \rfloor)} \nabla f_i(\boldsymbol{w}_i^{(t)}).$$

$$\tilde{R}^{(t)} = P_{\mathcal{W}}(\boldsymbol{w}^t, \boldsymbol{u}_t, \eta) = \boldsymbol{w}^{(t)} - \frac{1}{\eta} \prod_{\mathcal{W}} \left( \boldsymbol{w}^{(t)} - \eta \boldsymbol{u}_t \right)$$

$$R^{(t)} = P_{\mathcal{W}}(\boldsymbol{w}^t, \bar{\boldsymbol{u}}_t, \eta) = \boldsymbol{w}^{(t)} - \frac{1}{\eta} \prod_{\mathcal{W}} \left( \boldsymbol{w}^{(t)} - \eta \bar{\boldsymbol{u}}_t \right)$$

$$\hat{R}^{(t)} = P_{\mathcal{W}}(\boldsymbol{w}^t, \Phi(\boldsymbol{w}^{(t)}), \eta) = \boldsymbol{w}^{(t)} - \frac{1}{\eta} \prod_{\mathcal{W}} \left( \boldsymbol{w}^{(t)} - \eta \nabla \Phi(\boldsymbol{w}^{(t)}) \right).$$

By the $\alpha$-smoothness of $\Phi$ and the updating rule of $\boldsymbol{w}$ we have:

$$\mathbb{E}[\Phi(\boldsymbol{w}^{(t+1)})] - \mathbb{E}[\Phi(\boldsymbol{w}^{(t)})] \leq \frac{\alpha}{2}\mathbb{E}\left[\left\|\boldsymbol{w}^{(t+1)} - \boldsymbol{w}^{(t)}\right\|^2\right] + \left\langle \nabla\Phi(\boldsymbol{w}^{(t)}), \boldsymbol{w}^{(t+1)} - \boldsymbol{w}^{(t)}\right\rangle$$

$$\leq \frac{\eta^2\alpha}{2}\mathbb{E}\left[\left\|\tilde{R}^{(t)}\right\|^2\right] - \eta\mathbb{E}\left[\left\langle\nabla\Phi(\boldsymbol{w}^{(t)}), \tilde{R}^{(t)}\right\rangle\right]$$

$$\leq \frac{\eta^2\alpha}{2}\mathbb{E}\left[\left\|\tilde{R}^{(t)}\right\|^2\right] - \eta\mathbb{E}\left[\left\langle\boldsymbol{u}_t, P_{\mathcal{W}}(\boldsymbol{y}^t, \boldsymbol{u}_t, \eta)\right\rangle\right]$$

$$- \eta\mathbb{E}\left[\left\langle\nabla\Phi(\boldsymbol{w}^{(t)}) - \boldsymbol{u}_t, \tilde{R}^{(t)}\right\rangle\right].$$

According to Lemma 16, we can bound the first dot product term in the last inequality by $\|\tilde{R}^{(t)}\|^2$, so then we have:

$$\mathbb{E}[\Phi(\boldsymbol{w}^{(t+1)})] - \mathbb{E}[\Phi(\boldsymbol{w}^{(t)})]$$

$$\leq \frac{\eta^2\alpha}{2}\mathbb{E}\left[\left\|\tilde{R}^{(t)}\right\|^2\right] - \eta\mathbb{E}\left[\|\tilde{R}^{(t)}\|^2\right] - \eta\mathbb{E}\left[\left\langle\nabla\Phi(\boldsymbol{w}^{(t)}) - \boldsymbol{u}_t, \tilde{R}^{(t)}\right\rangle\right]$$

$$\leq -\left(\eta - \frac{\eta^2\alpha}{2}\right)\mathbb{E}\left[\left\|\tilde{R}^{(t)}\right\|^2\right] - \eta\mathbb{E}\left[\left\langle\nabla\Phi(\boldsymbol{w}^{(t)}) - \boldsymbol{u}_t, \tilde{R}^{(t)}\right\rangle\right]$$

$$\leq -\left(\eta - \frac{\eta^2\alpha}{2}\right)\mathbb{E}\left[\left\|\tilde{R}^{(t)}\right\|^2\right] + \frac{\eta}{2}\mathbb{E}\left[\left\|\nabla\Phi(\boldsymbol{w}^{(t)}) - \boldsymbol{u}_t\right\|^2 + \left\|\tilde{R}^{(t)}\right\|^2\right]$$

$$\leq \underbrace{-\left(\frac{\eta}{2} - \frac{\eta^2\alpha}{2}\right)}_{\leq -\frac{1}{4}\eta}\mathbb{E}\left[\left\|\tilde{R}^{(t)}\right\|^2\right] + \eta\mathbb{E}\left[\left\|\nabla\Phi(\boldsymbol{w}^{(t)}) - \bar{\boldsymbol{u}}_t\right\|^2 + \|\bar{\boldsymbol{u}}_t - \boldsymbol{u}_t\|^2\right]$$

$$\leq -\frac{1}{4}\eta\mathbb{E}\left[\left\|\tilde{R}^{(t)}\right\|^2\right] + \eta\mathbb{E}\left[\left\|\nabla\Phi(\boldsymbol{w}^{(t)}) - \bar{\boldsymbol{u}}_t\right\|^2\right] + \frac{\eta(2\sigma_w^2 + 4G_w^2)}{m}. \qquad (44)$$

Notice that:

$$\mathbb{E}\left[\left\|\hat{R}^{(t)}\right\|^2\right] \leq 2\mathbb{E}\left[\left\|\tilde{R}^{(t)}\right\|^2\right] + 2\mathbb{E}\left[\left\|\hat{R}^{(t)} - \tilde{R}^{(t)}\right\|^2\right]$$

$$\leq 2\mathbb{E}\left[\left\|\tilde{R}^{(t)}\right\|^2\right] + 4\mathbb{E}\left[\left\|\hat{R}^{(t)} - R^{(t)}\right\|^2\right] + 4\mathbb{E}\left[\left\|R^{(t)} - \tilde{R}^{(t)}\right\|^2\right]$$

$$\leq 2\mathbb{E}\left[\left\|\tilde{R}^{(t)}\right\|^2\right] + 4\mathbb{E}\left[\left\|\hat{R}^{(t)} - R^{(t)}\right\|^2\right] + 4\mathbb{E}\left[\left\|\boldsymbol{u}^{(t)} - \bar{\boldsymbol{u}}^{(t)}\right\|^2\right]$$

$$\leq 2\mathbb{E}\left[\left\|\tilde{R}^{(t)}\right\|^2\right] + 4\mathbb{E}\left[\left\|\nabla\Phi(\boldsymbol{w}^{(t)}) - \bar{\boldsymbol{u}}_t\right\|^2\right] + \frac{4\eta(2\sigma_w^2 + 4G_w^2)}{m}. \qquad (45)$$

Thus, plugging (45) into (44) to substitute $\mathbb{E}\left[\left\|\tilde{R}^{(t)}\right\|^2\right]$ yields:

$$\mathbb{E}[\Phi(\boldsymbol{w}^{(t+1)})] - \mathbb{E}[\Phi(\boldsymbol{w}^{(t)})]$$

$$\leq -\frac{1}{8}\eta\mathbb{E}\left[\left\|\hat{R}^{(t)}\right\|^2\right] + \frac{1}{2}\eta\mathbb{E}\left[\left\|\nabla\Phi(\boldsymbol{w}^{(t)}) - \bar{\boldsymbol{u}}_t\right\|^2\right] + \frac{\eta(2\sigma_w^2 + 4G_w^2)}{2m}$$

$$+ \eta\mathbb{E}\left[\left\|\nabla\Phi(\boldsymbol{w}^{(t)}) - \bar{\boldsymbol{u}}_t\right\|^2\right] + \frac{\eta(2\sigma_w^2 + 4G_w^2)}{m}$$

$$\leq -\frac{1}{8}\eta\mathbb{E}\left[\left\|\hat{R}^{(t)}\right\|^2\right] + \frac{3}{2}\eta\mathbb{E}\left[\left\|\nabla\Phi(\boldsymbol{w}^{(t)}) - \bar{\boldsymbol{u}}_t\right\|^2\right] + \frac{3\eta(2\sigma_w^2 + 4G_w^2)}{2m}. \qquad (46)$$

Plugging in the generalized PL-condition:

$$\frac{1}{\eta^2}\mathbb{E}\left[\left\|\prod_{\mathcal{W}}\left(\boldsymbol{w}^{(t)} - \eta\nabla\Phi(\boldsymbol{w}^{(t)})\right) - \boldsymbol{w}^{(t)}\right\|^2\right] = \mathbb{E}\left[\left\|\hat{R}^{(t)}\right\|^2\right] \geq 2\mu\left(\mathbb{E}[\Phi(\boldsymbol{w}^t)] - \mathbb{E}[\Phi(\boldsymbol{w}^*)]\right)$$

into (46) yields:

$$\mathbb{E}\left[\Phi(\boldsymbol{w}^{(t+1)}) - \Phi(\boldsymbol{w}^*)\right] \leq \left(1 - \frac{\mu\eta}{4}\right)\mathbb{E}\left[\Phi(\boldsymbol{w}^{(t)}) - \Phi(\boldsymbol{w}^*)\right]$$

$$+ \frac{3\eta}{2}\mathbb{E}\left\|\sum_{i=1}^N \lambda_i^{(\lfloor\frac{t}{\tau}\rfloor)}\nabla f_i(\boldsymbol{w}_i^{(t)}) - \nabla\Phi(\boldsymbol{w}^{(t)})\right\|^2 + 3\eta\frac{2\sigma_w^2 + 4G_w^2}{2m}.$$

$\square$

## G.3   Proof for Theorem 6

Now we proceed to the proof of Theorem 6. According to Lemma 18 we have:

$$\mathbb{E}\left[\Phi(\boldsymbol{w}^{(t+1)}) - \Phi(\boldsymbol{w}^*)\right] \leq \left(1 - \frac{\mu\eta}{4}\right)\mathbb{E}\left[\Phi(\boldsymbol{w}^{(t)}) - \Phi(\boldsymbol{w}^*)\right]$$

$$+ \frac{3\eta}{2}\underbrace{\mathbb{E}\left\|\sum_{i=1}^N \lambda_i^{(\lfloor\frac{t}{\tau}\rfloor)}\nabla f_i(\boldsymbol{w}_i^{(t)}) - \nabla\Phi(\boldsymbol{w}^{(t)})\right\|^2}_{T_1} + 3\eta\frac{2\sigma_w^2 + 4G_w^2}{2m}.$$

Now, we bound the term $T_1$ in above as:

$$T_1 \leq 2\mathbb{E}\left\|\nabla_{\boldsymbol{w}}\Phi(\boldsymbol{w}^{(t)}) - \sum_{i=1}^N \lambda_i^{(\lfloor\frac{t}{\tau}\rfloor)}\nabla_{\boldsymbol{w}} f_i(\boldsymbol{w}^{(t)})\right\|^2$$

$$+ 2\mathbb{E}\left\|\sum_{i=1}^N \lambda_i^{(\lfloor\frac{t}{\tau}\rfloor)}\nabla_{\boldsymbol{w}} f_i(\boldsymbol{w}^{(t)}) - \sum_{i=1}^N \lambda_i^{(\lfloor\frac{t}{\tau}\rfloor)}\nabla_{\boldsymbol{w}} f_i(\boldsymbol{w}_i^{(t)})\right\|^2$$

$$\leq 2\mathbb{E}\left\|\nabla_{\boldsymbol{w}} F(\boldsymbol{w}^{(t)}, \boldsymbol{\lambda}^*(\boldsymbol{w}^{(t)})) - \nabla_{\boldsymbol{w}} F(\boldsymbol{w}^{(t)}, \boldsymbol{\lambda}^{(\lfloor\frac{t}{\tau}\rfloor)})\right\|^2$$

$$+ 2\sum_{i=1}^N \lambda_i^{(\lfloor\frac{t}{\tau}\rfloor)}\mathbb{E}\left\|\nabla_{\boldsymbol{w}} f_i(\boldsymbol{w}^{(t)}) - \nabla f_i(\boldsymbol{w}_i^{(t)})\right\|^2$$

$$\leq 2L^2\mathbb{E}\left\|\boldsymbol{\lambda}^*(\boldsymbol{w}^{(t)}) - \boldsymbol{\lambda}^{(\lfloor\frac{t}{\tau}\rfloor)})\right\|^2 + 2L^2\mathbb{E}\left[\delta^{(t)}\right]$$

$$\leq 2L^2\left(2\left(1 - \frac{1}{2\kappa}\right)^{\lfloor\frac{t}{\tau}\rfloor}\mathbb{E}\|\boldsymbol{\lambda}^{(0)} - \boldsymbol{\lambda}^*(\boldsymbol{w}^{(0)})\|^2 + 2\kappa^2\tau^2\eta^2 G_w^2\left(4\kappa^2 + 1\right)\right) + 2L^2\mathbb{E}\left[\delta^{(t)}\right],$$

where we plug in the Lemma 14. Plugging $T_1$ back yields:

$$\mathbb{E}\left[\Phi(\boldsymbol{w}^{(t+1)}) - \Phi(\boldsymbol{w}^*)\right]$$

$$\leq \left(1 - \frac{1}{4}\mu\eta\right)\mathbb{E}\left[\Phi(\boldsymbol{w}^{(t)}) - \Phi(\boldsymbol{w}^*)\right] + 3\eta\frac{2\sigma_w^2 + 4G_w^2}{2m}$$

$$+ \frac{3\eta}{2}\left(4L^2\left(1 - \frac{1}{2\kappa}\right)^{\lfloor\frac{t}{\tau}\rfloor}\mathbb{E}\|\boldsymbol{\lambda}^*(\boldsymbol{w}^{(0)}) - \boldsymbol{\lambda}^{(0)}\|^2 + 4L^2\kappa^2\tau^2\eta^2 G_w^2\left(4\kappa^2 + 1\right) + 2L^2\mathbb{E}\left[\delta^{(t)}\right]\right)$$

$$\leq \left(1 - \frac{1}{4}\mu\eta\right)\mathbb{E}\left[\Phi(\boldsymbol{w}^{(t)}) - \Phi(\boldsymbol{w}^*)\right] + 3\eta\frac{2\sigma_w^2 + 4G_w^2}{2m}$$

$$+ 6\eta L^2\left(\left(1 - \frac{1}{2\kappa}\right)^{\lfloor\frac{t}{\tau}\rfloor}\mathbb{E}\|\boldsymbol{\lambda}^*(\boldsymbol{w}^{(0)}) - \boldsymbol{\lambda}^{(0)}\|^2\right)$$

$$+ \frac{3\eta}{2}\left(4L^2\kappa^2\tau^2\eta^2 G_w^2\left(4\kappa^2 + 1\right) + 2L^2\mathbb{E}\left[\delta^{(t)}\right]\right).$$

Unrolling the recursion yields

$$\mathbb{E}\left[\Phi(\boldsymbol{w}^{(T)}) - \Phi(\boldsymbol{w}^*)\right]$$

$$\leq \left(1 - \frac{1}{4}\mu\eta\right)^T \mathbb{E}\left[\Phi(\boldsymbol{w}^{(0)}) - \Phi(\boldsymbol{w}^*)\right] + \sum_{t=0}^{T}\left(1 - \frac{1}{4}\mu\eta\right)^t 3\eta\frac{2\sigma_w^2 + 4G_w^2}{2m}$$

$$+ 6\eta L^2 \mathbb{E}\|\boldsymbol{\lambda}^*(\boldsymbol{w}^{(0)}) - \boldsymbol{\lambda}_0\|^2 \sum_{t=0}^{T}\left[\left(1 - \frac{1}{2}\mu\eta\right)^t \left(1 - \frac{1}{2\kappa}\right)^{\lfloor\frac{t}{\tau}\rfloor}\right]$$

$$+ \frac{3}{2}\eta\left(\sum_{t=0}^{T}\left(1 - \frac{1}{4}\mu\eta\right)^t 4L^2\kappa^2\tau^2\eta^2 G_w^2\left(4\kappa^2 + 1\right) + 2L^2 \sum_{t=0}^{T}\left(1 - \frac{1}{4}\mu\eta\right)^t \mathbb{E}\left[\delta^{(t)}\right]\right)$$

$$\leq \exp\left(-\frac{\mu\eta T}{4}\right)\mathbb{E}\left[\Phi(\boldsymbol{w}^{(0)}) - \Phi(\boldsymbol{w}^*)\right] + 12\frac{2\sigma_w^2 + 4G_w^2}{2\mu m}$$

$$+ 6\eta L^2 \mathbb{E}\|\boldsymbol{\lambda}^*(\boldsymbol{w}^{(0)}) - \boldsymbol{\lambda}^{(0)}\|^2 \left(\frac{2\kappa\tau}{1 - \frac{1}{4}\eta\mu}\right)$$

$$+ \frac{6}{\mu}\left(4L^2\kappa^2\tau^2\eta^2 G_w^2\left(4\kappa^2 + 1\right)\right) + 3\eta L^2\left(10\eta^2\tau^2\left(\sigma_w^2 + \frac{\sigma_w^2}{m} + \Gamma\right)\right)T,$$

where we use the result of Lemmas 4 and 15. Plugging in $\eta = \frac{4\log T}{\mu T}$, and $m \geq T$, we have:

$$\Phi(\boldsymbol{w}^{(t)}) - \Phi(\boldsymbol{w}^*) \leq O\left(\frac{\Phi(\boldsymbol{w}^{(0)}) - \Phi(\boldsymbol{w}^*)}{T}\right) + \tilde{O}\left(\frac{\sigma_w^2 + G_w^2}{\mu T}\right) + \tilde{O}\left(\frac{\kappa^2 L\tau D_\Lambda^2}{T}\right)$$

$$+ \tilde{O}\left(\frac{\kappa^6\tau^2 G_w^2}{\mu T^2}\right) + \tilde{O}\left(\frac{\kappa^2\tau^2(\sigma_w^2 + \Gamma)}{\mu T^2}\right),$$

thus concluding the proof. □

## Footnotes

[2]This dependency is very heavy, and one open question is to see if we employ a variance reduction scheme to loosen this dependency.