[Reviews · NeurIPS 2020]

Review 1

Summary and Contributions: Focus on minimization of a convex combination of losses over N parties holding data, with uncertainty over what convex combination is desired, formulated as minimax optimization problem. This formulation was studied previously and this contribution is quite clear in what sense it extends the prior work. Appears to be valuable theoretical contributions (low confidence) but experimental evaluation is weak.

Strengths: Well grounded in prior work and clear in what is the novelty.

Weaknesses: Section 5

Correctness: Probably. I am not familiar with the theoretical results built upon so I don't have a good prior on what results to expect.

Clarity: Yes, with the exception of Sec 3 which reads harder.

Relation to Prior Work: Yes

Reproducibility: Yes

Additional Feedback: I have ready the response and other reviews. For concenrn about experimental evaluation, the response is mostly "[27] did it too", which is the concern I flag at the end of review. Not meeting a simple baseline is ignored. Moreover, [27] also tries to do experiment in a more realistic setup, where the simple baseline would not hold, which this work does not reproduce response does not mention. So I see this concern as not addressed. Based on opinions of other reviewers, I think the theoretical contribution is valuable but I can't comment on details. I feel the way this fits into the FL setup overall should be improved, or perhaps deemphasized - in which case my concerns would not be as valid. My overall rating stays the same, including the (low) confidence. Initial content: --- Minor personal opinion - It would be more beneficial if the title was a little more concrete What is in the introduction is very clear and well presented, but I would recommend adding details to it. In particular, the proposed algorithm would not be feasible for all scenarios where federated learning is relevant. For instance, this would not work for the system described by Bonawitz et al. "Towards Federated Learning at Scale: System Design" but would likely work for the system described in Ludwig et al. "IBM Federated Learning: An Enterprise Framework". See also Kairouz et al. "Advances and Open Problems in Federated Learning" for discussion on cross silo and cross device federated learning. I do not think this would make this work worse, quite the opposite. Scoping the work more clearly makes it easier to ask better questions of relevance to that setup, as well as highlights open problems for future work - I believe this notion of distribution robustness is equally important for both scenarios. L81: I do not think this can be called "real federated data". L86+ paragraph on convergence rates. I recommend reading Woodworth et al. "Is Local SGD Better than Minibatch SGD?" which just appeared in ICML and discusses how and when the existing comparisons are insufficient, especially when (not) compared with minibatch SGD. I think the discussion therein could also help better present the theoretical results in this work. L112 shwoing typo Section 3.1 I found the text is relatively hard to follow, but the Alg 1 is presented clearly. I suggest rethinking how exactly it is presented. Perhaps one can assume familiarity with the related work and ground the presentation in differences from such baseline. Sec 4 I cannot provide really confident assessment of the contribution here, as I am not very familiar with minimax optimization theory, and thus do not have a good idea what kind of rates should be achievable, and thus whether this contribution is significant or not. I hope other reviewers will comment on this. I did not look at the proofs. On the surface, it looks interesting to me, and difference from relevant [27] is quite clearly called out. Rather than being uniformly better, the contribution is presented as having worse convergence rate, but lower number of communication rounds to get there. As such, it is a different point on a possible pareto-frontier. Question: How do the rates compare with a basic algorithm such as gradient descent? Sec 5, Experiments. A major weakness of the paper. Only showing experiments on an artificially partitioned dataset is not meeting a bar for a persuasive evaluation today. For the specific setup presented here, comparison with a trivial baseline would be missing - predict most common label locally - It does not require any communication and achieves 100% accuracy. So I interpret the experimental result as showing poor performance on a very simple problem. The work [27] most closely related presents more complex experimental setup which is not even reproduced here. Other previous works such as [16] and also Caldas et al., "LEAF: A Benchmark for Federated Settings." or Reddi et al., "Adaptive Federated Optimization". present baselines based on dataset with more realistic partitioning. I recommend using those as starting point for design of a better experimental setup. If the other reviewers can confidently claim the theoretical results are interesting and bring significant novelty, I would be more willing to accept the work without experiments, compared to the current state. I am afraid that accepting the work in its current form would encourage others to submit works without proper evaluation and might thus be detrimental to the broader clarity in the field. ----- Potntial inspiration re Sec 4. "Technical challenge" Perhaps see recent Charles and Konecny, "On the Outsized Importance of Learning Rates in Local Update Methods" for inspiration on how the "changing minimizer" can be handled.


Review 2

Summary and Contributions: The paper studies distributionally robust (agnostic) federated learning as a minimax optimization problem for both strongly-convex-strongly-concave and nonconvex-strongly concave cases. The propose an algorithm with theoretical guarantees that is also communication efficient since it has local updates and partial participation of devices. Experimental results are also presented.

Strengths: Overall, I enjoyed reading this paper. It tackles an important problem in FL that is data heterogeneity and tries to approach it using distributionally robust or agnostic FL formulation. The proposed algorithm is clever. The theoretical guarantees are quite strong though they come with a weakness about gradient bounded assumption that I will mention.

Weaknesses: - For the theoretical results, the authors assume that the gradients are bounded. Many works in FL literature (See E.g. "FedPAQ: ..." by Reisizadeh et al. and "First Analysis of Local GD on Heterogeneous Data" by Khaled et al.) have tried to relax this assumption and it is well-known that assuming it makes the theory much simpler. Why do the authors need bounded gradients? Should be explained clearly. - A major promise of FL is preserving privacy. Local updates indeed increase privacy guarantees but the proposed algorithm requires sending a random w in the previous epoch to track a history of local gradients. This indeed compromises privacy and is in contrast with the key promise of FL. (From previous model, new model and an estimate of history of local gradients, information about data points will be leaked, right?) Can you comment on this? - As the authors mention, the communication cost will also be doubled. I believe that for theoretical purposes, this is fine but it would be good to add experiments as follows: (i) Add some random noise to model updates to increase differential privacy. How will the algorithm work then? (ii) Compress the models (quantize) to reduce communication load. This will add quantization noise. How will the algorithm work?

Correctness: I didn't check every detail but they look correct.

Clarity: Yes.

Relation to Prior Work: Yes, but it is good to add some references about quantization/compression and preserving privacy in federated learning.

Reproducibility: Yes

Additional Feedback: I have read the response and other reviews. I agree the numerical results can be improved, but I still find the theoretical contribution strong, so I would like to keep my score.


Review 3

Summary and Contributions: The paper generalizes the agnostic federated learning method of Mohri et al. [27] to allow communication between the nodes and central server to occur at given time intervals.

Strengths: The paper is clearly written, the proofs are detailed and the proposed method is a simple extension of previous algorithms. Convergence is proved in various settings, including convex, and under a PL condition.

Weaknesses: Motivation: One of the challenges mentioned in the introduction is low participation. This is not addressed, since the central server chooses which nodes participate at each round. This needs further discussion. Results: The convergence bounds assume a particular choice of the synchronization interval \tau. Since there is inherently a trade-off between communication cost and convergence rate, why was this particular trade-off chosen? The result would be more compelling if the trade-off could be adjusted, for example by giving the convergence rate (along with the optimal choice of parameters) when \tau is a given polynomial function of T. Another weakness is that these results assume a fixed horizon T.

Correctness: I only partially reviewed the appendix. The parts that I reviewed are technically correct.

Clarity: The paper is well-written overall, but suffers from frequent typos, undefined quantities, and a general lack of polish. Some examples below (apologies if I missed something): - The quantity \kappa (appears on line 78 and later in Section 4.3) is never defined. - The quantity f_i(x, \xi) was never formally defined, beyond saying (line 132) that \xi is randomly sampled from the i-th dataset. It is also not stated that f_i(x, \xi) is an unbiased estimate of f_i(x). This may be a common assumption, but it needs to be stated. - In Algorithm 1, there is confusion between \bar w and w. Should line 4, 12, 13 be \bar w instead of w? - Algorithm 2, line 2, should \xi be \xi_i? - line 163: "between local and global (*) at each iteration" missing noun at (*). - line 166: what is meant by a minimizer of a vector-valued function? - line 187: The bound involves a random variable \xi. Is it meant that the inequality holds almost surely? Is there a missing ^2 on the norm? - Missing ^2 in definition 3. - line 218: "centralized setting". Do the authors mean non-distributed? In distributed optimization one distinguishes between centralized (all nodes communicate with a central server) and decentralized (general communication graph). The setting of this paper is distributed centralized. - line 230: what is meant by "the choice of $\psi$ includes optimal transport"? Do you mean a Wasserstein distance? Please be more precise. - The bibliography needs to be more carefully reviewed. Refs [4] and [5] are the same. [9] should reference the original paper from the 40s. - Many typos in the appendix.

Relation to Prior Work: The literature review covers related work in federated learning. The problem studied here is a robust optimization problem, and there is a rich literature on the topic that should be reviewed, see for example Ben-Tal, El Ghaoui and Nemirovski, Robust Optimization.

Reproducibility: Yes

Additional Feedback: - Line 38: distribution drift is mentioned, which implies non-stationary distributions, but this is not the setting of the paper. Please rephrase. - If the experiments are simulated on a single machine, the communication cost should be inexistent or negligible. So how is the wall-clock time for DFRA worse than AFL? - In Figures 1 and 3, why was training stopped when accuracy reached 50%? It would be better to train the model until convergence, to verify experimentally whether the methods will indeed converge to a similar loss as \tau grows. - The conclusion can be improved, it currently has no meaningful discussion. - The broader impact section mentions that the method is designed to preserve the privacy of users, but there are no privacy guarantees provided here. The fact that the model is sent to the central server makes the method potentially vulnerable to privacy attacks, and this should be mentioned. Another issue, alluded to in the text but omitted from the broader impact section, is that when optimizing the worst-case distribution, the model could be more vulnerable to adversarial attacks from a single malicious node, and this limits the applicability of the method. ======================== Thank you for the response. One of my questions was why the authors made such specific choices of \tau, for example in Theorem 1. The response claims that this choice optimizes the rate (I am assuming this means the rate in T). I fail to see why this is the case. For example for Theorem 1, looking at the final bound in Section C.4 in the appendix (lines 540 and 542), wouldn't taking \tau = 1 (synchronize at each step), \gamma = \eta = 1/\sqrt T give an overall better rate of 1/\sqrt T? I think the result should be presented as a trade-off between communication cost and convergence rate: given a communication constraint \tau (for example a monomial function of T), what is the best choice of parameters \gamma, \eta that gives the best rate.


Review 4

Summary and Contributions: In this paper, the authors propose a new algorithm for robust FL with efficient communication.

Strengths: The strengths of the paper are as follows. 1. A nice robust FL with less cost communication compared to conventional FL is developed. 2. Nice theory is provided. 3. Experiments have been done to support the theory.

Weaknesses: The weaknesses of the paper are as follows. 1. To gain some advantage in communication, the convergence of algorithm is slower. 2. From the experiments, it seems that the proposed algorithm and conventional FL are not much different in terms of efficiency. (FIG2) 3. It seems that this framework may not be as efficient as asynchronous FL. In my opinion, it would be much better and convincing if the authors can compare their algorithm with asynchronous FL. 4. The same constant \tau should be used for all clients. It is not nice because some clients have many samples while other clients may have a small number of samples. 5. The assumption of bounded gradients and the assumption of strongly convex functions are used in the proof. However, these two assumptions cannot be used together as explained in https://arxiv.org/pdf/1802.03801.pdf. Therefore, the results for strongly convex/ convex cases seem pretty weak. 6. It would be very good for the reader if the authors can explain why the result in Theorem 1, i.e., max_\lambda E[...] - min_w E[...] <= O(1/sqrt(T)), tells us about the convergence rate of Algorithm 1. Actually, i do not see why w^ and lamda^ outputted by Algorithm 1 should be good solutions for the problem described in Equation 3?

Correctness: In my opinion, the claims are correct.

Clarity: Yes, it is.

Relation to Prior Work: Yes, it is.

Reproducibility: Yes

Additional Feedback:

[Author Response · NeurIPS 2020]

Many thanks to the reviewers for their deep, thoughtful reviews and constructive suggestions.

**Main Contributions**: We would firstly explain the theoretical contribution of our work. Our algorithm is the first to provide a provably communication-efficient and distributional robust FL algorithm, and both the design and the analysis of the algorithm is technically non-trivial.

(**R4**): Convergence analysis. To solve the minimax problem: $\min_{\boldsymbol{w}} \max_{\boldsymbol{\lambda}} F(\boldsymbol{w}, \boldsymbol{\lambda})$, our goal is to find the saddle point $(\boldsymbol{w}^*, \boldsymbol{\lambda}^*)$ such that $\forall \boldsymbol{w}, \boldsymbol{\lambda}, F(\boldsymbol{w}^*, \boldsymbol{\lambda}) \leq F(\boldsymbol{w}^*, \boldsymbol{\lambda}^*) \leq F(\boldsymbol{w}, \boldsymbol{\lambda}^*)$. We use the primal-dual gap: $\max_{\boldsymbol{\lambda}} F(\hat{\boldsymbol{w}}, \boldsymbol{\lambda}) - \min_{\boldsymbol{w}} F(\boldsymbol{w}, \hat{\boldsymbol{\lambda}})$ as convergence measure, a classic convergence measure used in convex-concave minimax optimization. It measures how much our solution $(\hat{\boldsymbol{w}}, \hat{\boldsymbol{\lambda}})$ is far from the saddle point $(\boldsymbol{w}^*, \boldsymbol{\lambda}^*)$ (if our solution is exactly the saddle point, then the primal-dual gap is zero). Our theoretical novelties are: (1) Our problem setting is novel. We study the minimax optimization problem where primal variable is trained locally, averaged periodically, and the dual variable can only be updated periodically. (2) To mitigate the issue that dual variable cannot get updated every iteration, we propose a novel gradient sampling method (as pointed by **R2** as a "clever algorithm" which we appreciate). By this gradient sampling update scheme, we have a unbiased approximation on the history gradients of dual variable. (3) We give the convergence analysis on convex-linear and nonconvex-linear settings. Nonconvex-concave (linear) is one of the hardest cases in minimax optimization, and in this work we even have more difficulties, due to the irregular updating scheme compared to a single machine case. In addition, we also give the analysis for regularized setting: strongly-convex-strongly-concave, and PL-strongly-concave cases. We totally agree with (**R4**) that we can not have both strong convexity and bounded gradients assumptions at the same time **unless** the problem is defined over a compact set. In our analysis we assume that parameter domain is bounded (Assumption 3), thus having both assumptions is sound. Currently we cannot remove bounded gradient assumption, because we need to characterize the variance of history gradient estimation of $\boldsymbol{\lambda}$. (**R3**) commented that "server choosing clients" mode cannot solve low participation issue. The solution can be that server sends request to more clients and waits until $K$ clients responded. The convergence analysis will be trickier though.

For the questions about convergence rate, we admit that to achieve communication efficiency, we sacrifice slightly on convergence rate (**R4**). However, that does NOT mean our algorithm is slower (**R4**). The communication cost is the major bottleneck that slows down the distributed training. As shown in the experiments, our algorithm converges faster than vanilla Agnostic Federated Learning (AFL). Compared to gradient descent (**R1**), in [27] they get $O(1/\sqrt{T})$ rate, slightly better than us, but not communication efficient. As for the choice of $\tau$ (**R3**), it is a very good question. We do optimize on $\eta$, $\gamma$ and $\tau$ to get the best convergence rate. The convergence rate is presented as polynomial of $\tau$ in the Appendix proofs. (**R4**) also had a comment that the same $\tau$ for all clients are not nice since clients will have different amounts of data. We respectfully disagree with this opinion, since each client will only compute gradient on a small fixed mini-batch of data, so it does not matter how many data they have in total.

For the privacy issue, (**R2**) proposed a very good question: since server asks client to evaluate its loss at $\tilde{\boldsymbol{w}}$ with data point $\xi = (\boldsymbol{x}, y)$, and return the loss $f_i(\tilde{\boldsymbol{w}}, \xi) = a$, does it mean server can infer the data point information equipped with $\tilde{\boldsymbol{w}}$ and $f_i(\tilde{\boldsymbol{w}}, \xi)$? Let us consider the simple linear regression case. If server wants to infer data point $\xi = (\boldsymbol{x}, y)$, it needs to solve the following problem: $\|\tilde{\boldsymbol{w}}^\top \boldsymbol{x} - y\|^2 = a$ to find out $\boldsymbol{x}$ and $y$. This problem does not admit a unique solution, or namely, server does not have enough information to determine $\boldsymbol{x}$ and $y$ at the same time. However, we should admit, there is still chance that information could be leaked due to some model inversion attack, but it is not just the problem in our work, but the problem in whole FL community. Thus the suggestions from (**R2**) about adding noise or model compression will be good future directions.

For experiment part, (**R1**) pointed out that our dataset partition is not practical and we should follow AFL's setup. We would argue that we exactly followed them: (1) We use the Fashion MNIST dataset, and adult dataset (in Appendix), the same as them; (2) We partition the dataset by giving one client only one class data, the same as them; (3) We use the full 10 classes Fashion MNIST dataset but AFL only use part of dataset, so our experiments are even more convincing; and (4) We compare the model trained in average domain (FedAvg) and trained by distributionally robust domain, the same as what they did with Fashion MNIST dataset, yet one difference is that we do not train the model fully on local data, but it is already widely observed (even in AFL paper) local model will have very poor out-of-distribution performance, let alone the worst distribution performance. The other question is that in Fig.2, the DRFA has the same performance with conventional FL (**R4**). This is because Fig.2 is showing the average distribution performance, hence, it is reasonable that in average our algorithm performs similar to FedAvg. We should mention that the main point is that our method is better than FedAvg in worst distribution performance, as shown in Fig.3. (**R4**) commented that our work may not be as efficient as asynchronous FL. We are confused in which aspect our work is worse, can asynchronous FL achieve more distributional robustness? or it can solve the DRO with less communication cost? If so, it would be great if providing related works and we are very happy to compare with them in subsequent version.

For some minor questions, Alg.1 L4, 12, 13 should be $\bar{\boldsymbol{w}}$ instead of $\boldsymbol{w}$, and in L12, 13 local models should not have bars. L166, it should be minimizer of loss functions not gradients. L187 we missed square sign. L218, we mean single machine case. We would also thank **R1** for different FL protocol papers, and we will discuss them.

[Meta-Review · NeurIPS 2020]

All reviewers agree (and I concur) that the paper contains solid theoretical results that improve in a non-trivial way upon previous work in FL. The overall impression was that the paper should be accepted based on its theoretical contributions. There were however some serious concerns regarding the experimental setup (that allows for trivial baselines that outperforms the evaluated methods); I urge the authors to seriously reconsider their methodology and ideally make a complete overhaul to the experimental part of the paper. (When doing so, please inspect carefully the comments of R1 for detailed guidelines.)